# Geostatistical upscaling of rain gauge data to support uncertainty analysis of lumped urban hydrological models

Manoranjan Muthusamy[1], Alma Schellart[1], Simon Tait[1], Gerard B.M. Heuvelink [2]

[1] Department of Civil and Structural Engineering, University of Sheffield, Sheffield, S1 3JD, UK
[2] Soil Geography and Landscape group, Wageningen University, Wageningen, 6700, The Netherlands

*Correspondence to*: Manoranjan Muthusamy (m.muthusamy@sheffield.ac.uk)

**Abstract.** In this study we develop a method to estimate the spatially averaged rainfall intensity together with associated level of uncertainty using geostatistical upscaling. Rainfall data collected from a cluster of eight paired rain gauges in a $400 \times 200$ m$^2$ urban catchment are used in combination with spatial stochastic simulation to obtain optimal predictions of the spatially averaged rainfall intensity at any point in time within the urban catchment. The uncertainty in the prediction of catchment average rainfall intensity is obtained for multiple combinations of intensity ranges and temporal averaging intervals. The two main challenges addressed in this study are scarcity of rainfall measurement locations and non-normality of rainfall data, both of which need to be considered when adopting a geostatistical approach. Scarcity of measurement points is dealt with by pooling sample variograms of repeated rainfall measurements with similar characteristics. Normality of rainfall data is achieved through the use of Normal Score Transformation. Geostatistical models in the form of variograms are derived for transformed rainfall intensity. Next spatial stochastic simulation which is robust to nonlinear data transformation is applied to produce realisations of rainfall fields. These realisations in transformed space are first back-transformed and next spatially aggregated to derive a random sample of the spatially averaged rainfall intensity. Results show that the prediction uncertainty comes mainly from two sources: spatial variability of rainfall and measurement error. At smaller temporal averaging intervals both these effects are high, resulting in a relatively high uncertainty in prediction. With longer temporal averaging intervals the uncertainty becomes lower due to stronger spatial correlation of rainfall data and relatively smaller measurement error. Results also show that the measurement error increases with decreasing rainfall intensity resulting in a higher uncertainty at lower intensities. Results from this study can be used for uncertainty analyses of hydrologic and hydrodynamic modelling of similar sized urban catchments as it provides information on uncertainty associated with rainfall estimation, which is arguably the most important input in these models. This will help to better interpret model results and avoid false calibration and force-fitting of model parameters.

Keywords: Geostatistical upscaling, spatial stochastic simulation, areal average rainfall intensity, hydrological modelling, uncertainty

## 1. Introduction

Being the process driving runoff, rainfall is arguably the most important input parameter in any hydrological modelling study. But it is a challenging task to accurately measure rainfall due to its highly variable nature over time and space, especially in small urban catchments. Despite recent advances in radar technologies rain gauge measurements are still considered to be the most accurate way of measuring rainfall, especially at short temporal averaging intervals (< 1 hour), which are of most interest in urban hydrology studies (Ochoa-Rodriguez et al., 2015). However, many commonly used urban hydrological models (e.g. SWMM, HBV) are lump catchment models (LCM) where time series of areal average rainfall intensity (AARI) are needed as model input. Therefore, point observations of rainfall need to be scaled up using spatial aggregation in order to be fed in to a LCM. There are a number of interpolation methods available for spatial aggregation and used in the various LCM to scale up point rainfall data. The simplest method is to take the arithmetic average (Chow, 1964) of the point observations within the catchment. But this method does not account for the spatial correlation structure of the rainfall and the spatial organisation of the rain gauge locations. Another commonly used method in hydrological modelling is the nearest neighbour interpolation (Chow, 1964; Nalder and Wein, 1998) which leads to Thiessen polygons. In this method the nearest observation is given a weight of one and other observations are given zero weights during interpolation, thereby ignoring spatial variability of rainfall to a certain extent. There are also other methods, with varying complexity levels, including inverse distance weighting (Dirks et al., 1998), polynomial interpolation (Tabios III and Salas, 1985) and moving window regression (Lloyd, 2005). The predictive performance of the above methods are found to be case-dependent and no single method has been shown to be optimal for all catchments and rainfall conditions (Ly et al., 2013). One common drawback with all the above methods is that they do not provide any information on the uncertainty of the predictions of AARI as all the methods are deterministic. The uncertainty in prediction of AARI mainly comes from two sources; uncertainty due to measurement errors and uncertainty associated with spatial variability of rainfall. The characteristics of measurement errors can vary depending on the rain gauge type. For example, errors associated with commonly used tipping bucket rain gauges range from errors due to wind, wetting, evaporation, and splashing (Fankhauser, 1998; Sevruk and Hamon, 1984) to errors due to its sampling mechanism (Habib et al., 2001). In addition to measurement errors and since rainfall can vary over space significantly, any spatial aggregation method for scaling up the point rainfall measurements incorporates more uncertainty (Villarini et al., 2008). The magnitude of the uncertainty depends on many factors including rain gauge density and location, rainfall variability, catchment size, topography and the spatial interpolation technique used. Quantification of the level of uncertainty is essential for robust interpretation of hydrological model outputs. For instance, the absence of information on uncertainty can lead to force fitting of hydrological model parameters to compensate for the uncertainty in rainfall input data (Schuurmans and Bierkens, 2006).

Geostatistical methods such as kriging present a solution to this problem by providing a measure of prediction error. In addition to this capability, these statistical methods also take into account the spatial dependence structure of the measured rainfall data (Ly et al., 2013; Mair and Fares, 2011). Although these features make geostatistical methods more attractive than deterministic methods, they are rarely used in LCM due to their inherent complexity and heavy data requirements. Since

they are statistical methods encompassing multiple parameters the amount of spatial data required for model inference is higher compared to deterministic methods. In addition the underlying assumption of geostatistical approaches typically requires data to be normally distributed (Isaaks and Srivastava, 1989). In general, catchments, especially those at small urban scales, do not contain as many measurement locations as required by geostatistical methods. Furthermore, rainfall intensity data are almost never normally distributed, especially at smaller averaging intervals (< hour) (Glasbey and Nevison, 1997). Despite these challenges geostatistical methods can provide information on uncertainty associated with predicted AARI. This capability can be utilised in uncertainty propagation analysis in hydrological models. In literature, geostatistical methods have been used to analyse the spatial correlation structure of rainfall at various spatial scales (Berne et al., 2004; Ciach and Krajewski, 2006; Emmanuel et al., 2012; Jaffrain and Berne, 2012), however its application to support uncertainty analyses of upscaling rainfall data has not been explored.

In this paper we present a geostatistical approach to derive AARI and the level of uncertainty associated with it from observations obtained from multiple "paired" rain gauges located in a small urban catchment. The proposed approach presents solutions to the above described challenges of geostatistical methods. First, it uses pooling of sample variograms of rainfall measurements at different times but with similar characteristics to increase the number of paired observations used to fit variogram models. Second, a data transformation method is employed to transform the rainfall data to obtain a normally distributed data set. The level of uncertainty in the prediction of AARI is then quantified for different combinations of temporal averaging intervals and intensity ranges for the studied urban catchment. We focused on a small urban catchment with a spatial extent of less than a kilometre given the recent research on the significance of unmeasured spatial rainfall variability at such spatial scales, especially for urban hydrological and hydrodynamic modelling applications (Gires et al., 2012, 2014; Ochoa-Rodriguez et al., 2015).

## 2. Data collection

### 2.1 Location and rain gauge network design

The study area is located in Bradford, a city in West Yorkshire, England. Bradford has a maritime climate, with an average yearly rainfall of 873 mm recorded from 1981-2010 (MetOffice, UK). The rain gauge network, used in this study was located at the premises of Bradford University (Fig. 1) and rainfall data were collected from paired tipping bucket rain gauges placed at eight locations covering an area of $400 \times 200$ m$^2$. Data used in this study were collected from April, 2012 to August, 2012 and from April, 2013 to August, 2013. These stations were located on selected roofs of the university buildings, thereby providing controlled, secure and obstruction-free measurement locations. Each station consists of two tipping bucket type rain gauges mounted 1 m apart. On each roof the paired gauges were placed such that the height of the nearest obstruction is less than two times the distance between the gauges and the obstruction The rim of each rain gauge was set up around 0.5 m above the surrounding ground level following UK standard practice (MetOffice, UK). An example of the measurement setup (Station 6) is also shown in Fig. 1. A histogram of the inter-station distances of the rain gauge

network is presented in Fig. 2. Lag distances covered in this network are distributed between 21 m (St. 4-St. 5) and 399 m (St. 1-St. 3).

All rain gauges are ARG100 tipping bucket type with an orifice diameter of 254 mm and a resolution of 0.2 mm. Dynamic calibration was carried out for each individual gauge before deployment and visual checks were carried out every 4-5 weeks during the measurement period to ensure that the instruments were free of dirt and debris. Data loggers were reset every 4-5 weeks during data collection to avoid any significant time drift. Measurements (number of tips) were taken every minute and recorded on TinyTag data loggers mounted in each rain gauge.

Quality control procedures were performed prior to statistical analysis, taking advantage of the paired gauge setup to detect gross measurement errors. The paired gauge design provides efficient quality control of the rain gauge data records as it helps to identify the instances when one of the gauges fails, and to clearly identify periods of missing or incorrect data (Ciach and Krajewski, 2006). During the dynamic calibration of all rain gauges in the laboratory before deployment, it was identified that the highest and lowest values of the calibration factors for the tipping bucket size are 0.196 mm and 0.204 mm. The gauges were recalibrated in the laboratory after the first period of measurement and it was found that the largest change in calibration factor for any gauge was a maximum of 4% of the original calibration factor. Therefore a maximum difference of 4% in volume per tip was assumed to be caused by inherent instrument error. It was therefore decided that this is the maximum acceptable difference between any pair of gauges. Sets of cumulative rainfall data corresponding to specific events from the paired gauges were checked against each other and if the (absolute) difference in cumulative rainfall was greater than 4%, that complete set was identified as unreliable and removed from further analysis.

## 2.2    Characteristics of the data

The total average network rainfall depth for the summer seasons of 2012 and 2013 are 538 mm and 207 mm, respectively. Figure 3 shows time series of daily rainfall averaged over the network for 2012 and 2013. There is a significant difference in cumulative rainfall between 2012 and 2013. This is because 2012 was the wettest year recorded in 100 years in the UK (MetOffice, UK) and 558 mm of rainfall during 2012 summer was unusually high. An average rainfall of only 360 mm was recorded during April to September over the 1981 - 2010 period at the nearest operational rain gauge station at Bingley, which is around 8 km from the study site with a similar ground elevation (MetOffice, UK).

The data set for 2012 and 2013 contains 13 events yielding more than 10 mm network average rainfall depth each and lasting for more than 20 min. A summary of these events is presented in Table 1. Note that this event separation is only used for the presentation of results in chapter 4. Hence it does not leave out any data from the development and calibration of the geostatistical model as presented in chapter 3. Table 1 shows that the total event duration ranges from 1.5 h to 11.4 h while the event network average rainfall intensity varies from 1.79 mm/h to 7.96 mm/h. Table 1 also includes summary statistics of peaks of events (temporal averaging interval of 5 min) for the eight stations within the network. Although the spatial extent of the area is only $400 \times 200$  $m^2$, it is clear that there is a considerable difference in rainfall intensity measurements

indicated by the standard deviation and range of peaks observed in the individual events. The maximum standard deviation between peaks of individual events is 9.27 mm/h for event 8, which is around 12.5% of the mean network peak intensity of 74.4 mm/h. This variation provides evidence of the potential importance of analysing uncertainty in the estimation of AARI even in such a small urban catchment.

## 3. Methodology

Figure 4 summarises the procedure of geostatistical upscaling of the rainfall data adapted in this study in a step-by-step instruction followed by the detail descriptions of each step. This complete procedure was repeated for temporal averaging intervals of 2 min, 5 min, 15 min and 30 min in order to investigate the effect of temporal aggregation on the prediction of AARI. The entire ten months of collected data were used for the development and calibration of the geostatistical model.

### 3.1 Step 1: Pooling of sample variograms

The rain gauge network contains eight measurement locations. These eight measurement locations give 28 spatial pairs at a given time instant which yields too few spatial lags than would normally be used in geostatistical modelling. For example, Webster and Oliver (2007) recommends around 100 measurement points to calibrate a geostatistical model. The procedure adapted in this study increases the number of pairs by pooling sample variograms for time instants with similar rainfall characteristics. With $n$ measurement locations and measurements taken at $t$ time instants, the pooling over $t$ time instants creates $t \times \frac{1}{2} \times n \times (n\text{-}1)$ spatial pairs. Although this procedure increases the number of spatial pairs by a factor $t$, the spatial separation distances for which information is available will be limited to the original configuration of the $n$ measurement locations.

The underlying assumption of this pooling procedure is that the spatial variability over the pooled time instants is the same. Therefore it is important to pool sample variograms of rainfall measurements with similar rainfall characteristics. Since the spatial rainfall variability is often intensity-dependent (Ciach and Krajewski, 2006), the characteristics of a less intense rainfall event may not be the same as that of a high intensity rainfall event. Hence to make the assumption of consistency of spatial variability, the range of rainfall intensity over the pooled time instants should be reasonably small. On the other hand, one should also make sure that there are enough time instants within a pooled subset to meet the data requirement to calibrate the geostatistical model. Based on the above two criteria, three rainfall intensity classes were selected. The maximum threshold value was limited to 10mm/hr to have enough time instants for the highest range (i.e. > 10 mm/hr) in order to produce stable variograms even at 30 min temporal averaging interval. It was then decided to divide the 0 – 10 mm/hr class to two equal subclasses (i.e. < 5mm/hr and 5-10 mm/hr). This resulted in three subclasses, which is a reasonable number given the size of the data set and computational demand. The number of time instants ($t$) within each rainfall intensity class is presented for three temporal averaging intervals in Fig. 5. The natural characteristic of rainfall data results in the dominance of lower intensity rainfall (0.1-5.0 mm/h) over the recording period. In addition, the number of time

instants $t$ obviously reduces with increasing temporal averaging intervals due to the aggregation process. As a consequence there are only seven time instants for the intensity range > 10 mm/h at the 30 min temporal averaging interval. This limits the maximum temporal averaging interval to 30 min for our analyses. For a catchment of this size ($400 \times 200$ m²) it is very unlikely to have a response time of more than 30 min. Hence, from a hydrological point of view consideration of temporal averaging intervals longer than 30 min would not be sensible. Note that although there are only seven time instants, the pooling procedure will produce 196 (=7×28) points to calculate and calibrate the geostatistical model for that intensity class.

## 3.2 Step 2: Standardisation of rainfall intensities

Having chosen the rainfall intensity classes to create pooled time instants, there can still be inconsistency in spatial variability between time instants within a class and therefore assuming a single geostatistical model for the whole subset may not be realistic. To reduce this effect to a certain extent, all observations within an intensity class were standardised using the mean and standard deviation of each time instant as follows:

$$\tilde{r}_{ix} = \frac{r_{ix} - m_i}{sd_i}$$
(1)

where $i=1\dots t$, $x=1\dots n$; $\tilde{r}_{ix}$ is standardised rainfall intensity at a time instant $i$ and location $x$; $r_{ix}$ is rainfall intensity at time instant $i$ and location $x$ and; $m_i, sd_i$ are mean and standard deviation of rainfall intensities at time instant $i$, respectively. Further steps were carried out on the standardised rainfall intensity.

## 3.3 Step 3: Normal transformation of data

The upper part of Fig. 6 shows the distribution of standardised rainfall intensity for a temporal averaging interval of 5 min derived using Eq. (1). From the figure it is clear that the data are not normally distributed. Distributions for other temporal averaging intervals (i.e. 2 min, 15 min and 30 min) show a similar behaviour. But the geostatistical upscaling method to be used is based on the normal distribution. This requires the rainfall data to be normally distributed prior to the calibration of the geostatistical model. The Normal Score Transformation (NST, also known as normal quantile transformation (Van der Waerden, 1953)) is a widely used method to transform a variable distribution to the Gaussian distribution. It has widely been applied in many hydrological applications (Bogner et al., 2012; Montanari andBrath, 2004; Todini, 2008; Weerts et al., 2011). The concept of NST is to match the $p$-quantile of the data distribution with the $p$-quantile of the standard normal distribution. Consider a standardised rainfall intensity $\tilde{r}$ with cumulative distribution $F_{\tilde{R}}(\tilde{r})$. It is transformed to a $r_N$ value with a Gaussian cumulative distribution $F_{R_N}(r_N)$ as follows:

$$r_N = F_{R_N}^{-1}\big(F_{\tilde{R}}(\tilde{r})\big)$$
(2)

Detailed description of NST including the steps involved can be found in Bogner et al. (2012), Van der Waerden (1953) and Weerts et al. (2011). The lower part of Fig. 6 shows the transformed standardised intensity for the temporal averaging interval of 5 min.

## 3.4    Step 4: Calibration of Geostatistical model

A geostatistical model of (normalised) rainfall intensity $r_N$ (derived from Section 3.3) at any location $x$ can be written as:

$$r_N(x) = p(x) + \varepsilon(x) \tag{3}$$

where $p(x)$ is the trend (explanatory part) and $\varepsilon(x)$ is the stochastic residual (unexplanatory part). Considering the availability of data, small catchment size and scope of this study, it was assumed that the trend is constant and does not depend on explanatory variables (e.g. topography of the area, wind direction). The stochastic term $\varepsilon$ is spatially correlated and characterised by a variogram model. A variogram model typically consists of three parameters; nugget, sill and range

(Isaaks and Srivastava, 1989). The nugget is the value of the semi-variance at near-zero distance. It is often greater than zero because of random measurement error and micro-scale spatial variation. The range is the distance beyond which the data are no longer spatially correlated. The sill is the maximum variogram value and equal to the variance of the variable of interest (Isaaks and Srivastava, 1989)

## 3.5    Step 5: Spatial stochastic simulation

The assumption of a constant trend makes that the spatial interpolation can be solved using an ordinary kriging system (Isaaks and Srivastava, 1989):

$$\sum_{x=1}^{n} w_x \, \gamma_{xy} - \mu = \gamma_{yz} \quad \forall \, y = 1, \dots, n \tag{4}$$

$$\sum_{y=1}^{n} w_y = 1 \tag{5}$$

where $w_y, y = 1, \dots, n$ are ordinary kriging weights, $\gamma_{xy}$ is the semivariance between rainfall intensities at locations $x$ and $y$ $\gamma_{yz}$ is the semivariance between rainfall intensities at location $y$ and prediction location $z$, and $\mu$ is a Lagrange parameter. Once the ordinary kriging weights are calculated using Eq. (4) and Eq. (5), point rainfall intensities can be predicted using

point kriging at any given point by taking the weighted average of the observed rainfall intensities, using the $w_l$ as weights. In this case we need a change of support from point to block as our intention is to predict the average rainfall intensity over the catchment. This is usually done by predicting at all points inside the catchment and integrating these over the catchment. This procedure is known as block kriging (Isaaks and Srivastava, 1989), which also has provisions for calculating the prediction error variance of the catchment average. But the procedure of NST as explained in Section 3.4 also involves back-

transformation of kriging predictions to the original domain at the end (Step 6). Since this transformation is typically non-linear, the back-transform of the spatial average of the transformed variable that is obtained from block kriging is not the same as the spatial average of the back-transformed variable; we need the latter and not the former. In principle, we could predict at all points within the block, back-transform all and next calculate the spatial average, but standard block kriging

software implementations do not support this and neither is it possible to compute the associated prediction error variance. Hence block kriging cannot be applied. The alternative used in this study is to apply a computationally more demanding spatial stochastic simulation approach, which involves generation of a larger number of realisations and spatial averaging of these realisations. Unlike kriging, spatial stochastic simulation does not aim to minimize the prediction error variance but focuses on the reproduction of the statistics such as the histogram and variogram model (Goovaerts, 2000). The output from

spatial stochastic simulation is a set of alternative rainfall realisations ('possible realities'). The mean of a large set of realisations approximates the kriging prediction, while their standard deviation approximates the kriging standard deviation. We used the sequential Gaussian simulation algorithm which involves the following steps (Goovaerts, 2000):

     i.      Define a prediction grid (a 25m × 25m regular grid in this case);

     ii.     Visit a randomly selected grid cell that has not been visited before and predict the transformed rainfall intensity

15           at the grid cell centre using ordinary kriging, this yields a kriging prediction and a kriging standard deviation;

     iii.    Use a pseudo-random number generator to sample from a normal distribution mean equal to the kriging prediction and standard deviation equal to the kriging standard deviation and assign this value to the grid cell centre;

     iv.    Add the simulated value to the conditioning data set, in other words treat the simulated value as if it were

20           another observation;

     v.      Go back to step ii and repeat the procedure until there are no more unvisited grid cells left.

The five steps above produce a single realisation. This must be repeated as many times as the number of realisations required (500 in this study). It must also be repeated for each time instant, which explains that the computational burden can be high. Implementation of these steps with the gstat package in R (Pebesma, 2004) is straightforward.

The grid size and number of simulations (i.e., the sample size) were selected considering the spatial resolution of available measurements and computational demand. It was observed that neither a finer grid nor more simulations improved the results significantly. Increasing the resolution to 10 m × 10 m only reduces the standard deviation of the prediction by less than 5% in most cases while making the computational time six times higher (a summary on computation power is presented as supplementary material).

**3.6     Step 6-9: Calculation of AARI and associated uncertainty**

Once the realisations have been prepared these are back-transformed by applying the inverse of Eq. (2) to all grid cells (step 6). Some values derived from spatial stochastic simulation were outside the transformed data range. Hence during back

transformation (step 6) of these values linear extrapolation was used. These linear models were derived using a selected number of head and tail portion of normal Q-Q plot. This is one of the simplest and most commonly used solutions for NST back-transformation (Bogner et al., 2012; Weerts et al., 2011). Considering the scope of this study and the relatively small number of data which had to be extrapolated, other extrapolation methods were not explored. After step 6, the back transformed realisations are spatially averaged one by one (step 7). This yields as many spatially averages as the number of realisations that had been generated in step 5. This set of values is a simple random sample from the probability distribution of the catchment average rainfall. Thus, the sample mean and standard deviation provide estimates of the mean and standard deviation of the distribution, respectively (step 8). Finally, by doing the inverse standardisation of the mean and standard deviation of the distribution to account for step 2, the AARI and associated uncertainty measure (standard deviation) were derived (Step 9).

## 4. Results and Discussion

### 4.1 Calibration of the geostatistical model of rainfall

As explained in Section 3.4, the geostatistical model of transformed rainfall data were calibrated using variograms for three different intensity ranges. This procedure was repeated for temporal averaging intervals of 2 min, 5 min, 15 min and 30 min. Exponential models were fitted to empirical variograms. The resulting variograms are presented in Fig. 7.

The variograms illustrate two properties of the collected rainfall measurements; spatial variability of rainfall, and measurement error. One of the main parameters which characterises these properties is the nugget. Theoretically at zero lag distance the variance should be zero. However most of the variograms exhibit a positive nugget effect (generally presented as nugget-to-sill ratio) at zero lag distance. This nugget effect can be due to two reasons; random measurement error and microscale spatial variability of rainfall. Unfortunately we cannot quantify these causes individually using the variograms. But there is a consistent pattern of nugget against both rainfall intensity class and temporal averaging interval which helps to interpret the variograms.

Considering the behaviour of nugget-to-sill ratio against rainfall intensity class, it can be observed that the smaller the intensity the higher the nugget-to-sill ratio, regardless of temporal averaging interval. For example, at 2 min averaging interval the nugget-to-sill ratio increases from zero to almost one (nugget variogram) as the rainfall intensity class changes from > 10 mm/h to < 5 mm/h. The pure nugget variogram at < 5 mm/h means that either there is no spatial correlation at the regarded distance, or the spatial correlation of the field cannot be detected by the measurements because of the measurement error. Looking at the behaviour of nugget-to-sill ratio against temporal averaging interval, Fig. 7 shows that the smaller the averaging interval the higher the nugget-to-sill ratio, regardless of rainfall intensity class. For example, for rainfall intensity class 5.0–10.0 mm/h the nugget-to-sill ratio decreases from almost one to zero as the temporal averaging interval increases from 2 min to 30 min. Overall these observations show that the combined effect of random measurement error and

microscale special variability of rainfall characterised by nugget-to-sill ratio decreases with increasing (a) rainfall intensity class and (b) averaging interval.

Regarding the behaviour of the nugget-to-sill ratio against averaging interval, it is expected that with the averaging interval the (microscale) spatial correlation of rainfall would increase, which partly explains the observed pattern. The increase in spatial correlation of rainfall intensity with increasing temporal averaging interval agrees with other similar studies (e.g. Ciach and Krajewski, 2006; Fiener and Auerswald, 2009; Krajewski et al., 2003; Peleg et al., 2013; Villarini et al., 2008). For example, Krajewski et al. (2003) observed in their study on analysis of spatial correlation structure of small-scale rainfall in central Oklahoma a similar behaviour using correlogram functions for different temporal averaging intervals. But commenting on the decreasing trend of the nugget-to-sill ratio against intensity class, it cannot be attributed to improvement in microscale spatial correlation as it is neither natural nor proven. In fact, in Fig. 7 the behaviour of spatial correlation against rainfall intensity class does not show a distinctive trend except at the origin, i.e. the nugget effect. The absence of any consistent trend of spatial variability against intensity class was also observed in Ciach and Krajewski, (2006). Meanwhile this decreasing trend of nugget-to-sill ratio against rainfall intensity corresponds well with measurement errors of tipping bucket type rain gauges caused by its sampling mechanism (hereafter referred as TB error). This is due to the rain gauges' inability to capture small temporal variability of the rainfall time series. The behaviour of TB error against rainfall intensity as seen from Fig. 7 complements results from previous studies (Habib et al., 2001; Villarini et al., 2008). These studies also show that the TB error decreases with temporal averaging interval. Habib et al. (2001) found similar behaviour of TB error with increasing intensity (0-100 mm/h) and also with increasing averaging interval (1 min, 5 min and 15 min). Although the bucket size used in their study (0.254 mm) is slightly different from our rain gauge bucket size of 0.2 mm, the characteristic of the TB error against rainfall intensity for different averaging interval is consistent in both cases. In summary, the behaviour of nugget-to-sill ratio of the variograms against temporal averaging interval can be explained by the combined effect of microscale spatial variability of rainfall and TB error, while the behaviour of nugget-to-sill ratio against intensity range can mainly be attributed to the latter.

In addition to the nugget-to-sill ratio, another parameter that characterises the variograms is the range, i.e. the distance up to which there is spatial correlation. At lower temporal averaging intervals ($\leq$ 5 min) the variograms for all rainfall intensity classes reach the variogram range very quickly (< 100 m). But at averaging intervals $\geq$ 15 min, the range has not been reached even at a maximum separation distance, showing the improvement in spatial correlation. High spatial variability of rainfall at shorter temporal averaging interval ($\leq$ 5 min) is an important observation in the context of urban drainage runoff modelling, as the time step used in such models is generally around 2 min for small catchments.

The fact that the data set covers only 10 months of data from two years with varying climatology is something that need to be acknowledged. However, for previous studies using such a dense network the duration of data collection is similar (e.g.: 15 months - Ciach and Krajewski, 2006; 16 months - Jaffrain and Berne, 2012). These time periods are reflection of the practical and funding issues to maintain such dense networks operating accurately for extended periods. The characteristics of our data are comparable with (Ciach and Krajewski, 2006; Fiener and Auerswald, 2009) as these studies also used rainfall

data from warm months to investigate the spatial correlation structure. Despite the fact that the data cover only 10 months all derived variogram models are stable and reliable. Webster and Oliver (2007) suggested around 100 samples to reliably estimate a variogram model. Even in the case of 30 min temporal averaging interval and > 10 mm/hr (where we had the fewest number of observations) we had a total of 196 spatial lags to calculate the variogram. Furthermore, we demonstrated

that all derived variogram models are stable and reliable by examining sub-sets of the data. We randomly selected 80% of the data from each intensity class and reproduced the variograms to compare them with the variograms presented in Fig. 7. We had to limit the subclass percentage to 80% to give enough time instants to reproduce variograms for all subclasses. We repeated this procedure a few times. Comparing these variograms with Fig. 7 shows that these variograms are very similar. One set of the variograms computed from 80% of the data are presented as supplementary material. This analysis supports

our claim that the variograms shown in Fig. 7 are stable and an adequate representation of the rainfall spatial variation for each intensity class and temporal averaging interval.

One of the assumptions we made during the pooling procedure is that the spatial variability is reasonably consistent within a pooled intensity class. We acknowledge that with narrower intervals the assumption of consistency in spatial variability would be more realistic. But with the available data we had to find a compromise with the number of time instants. We

believe that using three intensity subclasses is a reasonable compromise. Further we also introduced step 2 (section 3.2) that standardises the rainfall for each time instant within a subset. Although variograms are derived only for the whole subset, step 2 (before geostatistical upscaling) and step 9 (after geostatistical upscaling) ensure that the probabilistic model is adjusted for each time instant separately. Effectively, we assume the same correlogram for time instants of the same subclass, not the same variogram. Although this does not justify the assumption of similar spatial correlation structure within

the pooled classes, it at least relaxes the assumption of the same variogram within subclasses. To compare the behaviour of variogram models for a narrower intensity interval, we produced variograms for narrower intensity classes ranging from 0 to 14 mm/hr for the 5 min averaging interval. The highest intensity class is limited to ≥12 - <14 mm/hr as for further narrower ranges (i.e ≥14 - <16 mm/hr and so on) there are not enough sample points to produce a meaningful variogram. Narrower intensity classes means that the assumption of similar spatial variability within a pooled subset is more realistic. Comparing

Fig. 8 with Fig. 7, we conclude that the variograms shown in Fig. 7 are accurate representations of the average spatial variability conditions for corresponding intensity classes.

## 4.2    Geostatistical upscaling of rainfall data

Having calculated all variograms, the next step is to apply spatial stochastic simulation for the time instants of interest followed by steps 6 to 9 in Fig. 4 to calculate the AARI together with associated uncertainty. This procedure was carried out

for all events presented in Table 1. The following sections present and discuss the predicted AARI and associated uncertainty levels derived from step 9.

#### 4.2.1    Prediction error vs AARI

The scatter plot in Fig. 9 shows the coefficient of variation of the prediction error (CV, refer Eq. (6)) plotted against predicted AARI at 5 min averaging interval for all time instants of all events presented in Table 1.

$$CV = \frac{AARI\ prediction\ error\ standard\ deviation}{Predicted\ AARI} \times 100\% \qquad (6)$$

The uncertainty level in the prediction of AARI represented by the CV is due to the combined effect of both spatial variability of rainfall and TB error in the rainfall data. It can be seen here that there is a clear trend of increasing CV with increasing AARI. The CV values are as high as 80% when the AARI is smaller than 1 mm/hr and they get reduced to less than 10% when AARI is larger than 10 mm/hr. In a previous study by Pedersen et al. (2010) using rainfall measurements from similar tipping bucket type rain gauges, they also found  that the uncertainty in prediction of mean rainfall depth decreases with increasing mean rainfall depth, but due to the limited information in their results they could not analyse this observation in detail. But here it is clear that this observation corresponds well with what we already observed in variograms in Fig. 7. These variograms show higher nugget-to-sill ratio at lower intensity due to high TB error consequently causing higher uncertainty in the prediction of AARI. At intensity class 0-5 mm/hr the nugget-to-sill ratio was almost one (nugget variogram) and as a result the derived CV values are significantly higher than other two intensity classes. It is interesting to note that, in the range of 1- 10 mm/hr, there are few points that are separated from the larger cluster with almost zero CV. It shows a consistent rainfall measurement over the area at these time instants, which results in a very small CV in the predicted AARI.

The above discussion is based on results from 5 min temporal averaging interval. The following section discusses the effect of temporal averaging interval on prediction error. Further, although CV in Fig. 9 gets as high as 80%, the corresponding AARI is less than 1 mm/hr, thus the prediction error has a very less significance in urban hydrology. Hence we also analysed the prediction error associated with rainfall events' peaks in the last section.

#### 4.2.2    Prediction error vs temporal averaging interval

Having analysed the behaviour of the prediction error CV against predicted AARI, this section presents the effect of temporal averaging interval on the prediction error of AARI. Figure 10 shows the kriging predictions with 95 % prediction intervals derived from the prediction standard deviation for temporal averaging intervals of 2 min, 5 min, 15 min and 30 min for event 11.  Event 11 has average conditions in terms of event duration and peak intensity. Prediction errors of other events against the temporal averaging interval follow the same pattern of behaviour.

While short time intervals are of greater interest in urban hydrology, they also lead to large uncertainties. Figure 10 shows the smaller the temporal averaging interval, the larger the prediction interval and the larger the level of uncertainty.. This is due to the combined effect of higher spatial variability and larger TB error at lower temporal averaging interval as seen from Fig. 7.  When the averaging interval is larger than 15 min the prediction interval width becomes negligible. But temporal scales of interest in urban hydrology of similar sized catchment can be as low as 2 min where there is still considerable

uncertainty. The 95 % prediction interval shows around ± 13 % of error in rainfall intensity corresponding to a prediction of peak rainfall of 47 mm/h at 2 min averaging interval. While temporal aggregation decreases uncertainty, it obviously leads to a significant reduction of the predicted peaks of AARI. For example, the peak of event 11 gets reduced to around 20 mm/h from around 50 mm/h when averaging interval increases from 2 min to 30 min. Hence a careful trade-off between temporal

resolution and accuracy in rainfall prediction is needed to decide the most appropriate time step for averaging point rainfall data for urban hydrologic applications.

The decreasing trend of uncertainty in the prediction of AARI with increasing temporal averaging interval agrees with a previous study by Villarini et al. (2008). Although the spatial extent of their study is much larger (360 km$^2$), their results also show that the spatial sampling uncertainties tend to decrease with increasing temporal averaging interval due to improvement

in measurement accuracy and improved  spatial correlation.

### 4.2.3    Prediction error Vs peak rainfall intensity

In addition to rainfall event durations, rainfall event peaks are also of significant interest in urban hydrology as most of the hydraulic structures in urban drainage systems are designed based on peak discharge which is often derived from peak rainfall. Hence it is important to consider the uncertainty in prediction of peaks of AARI. Figure 11 presents predicted peaks

of AARI for all 13 events presented in Table 1, together with labels indicating corresponding CV (%) values. The peak intensities range from 6 mm/h to 92 mm/h at 2 min averaging interval and this range narrows down to 3 mm/h – 21 mm/h at averaging interval of 30 min as a result of temporal aggregation. As expected, temporal aggregation from 2 min to 30 min also results in the reduction of CV. The highest CV at 2 min averaging intervals is 13% for event 4 and reduces to 1.7% at 30 min averaging interval. But it can also be noted that events 5, 6, 8 and 11 show their highest CV at 5 min averaging interval

and not at 2 min averaging interval. Tracking back these events, they indeed show more spatial variation over 5 min period compared to 2 min period around the peak.

As discussed in section 4.2.1, CV decreases with increasing predicted rainfall peaks and this effect is dominant when the averaging interval is at the lowest, i.e. 2 min. This is when the TB error is at its highest. When the temporal averaging interval is 30 min where the TB error is at its lowest, the difference between CV for lower (< 10 mm/h) and higher (> 10

mm/h) intensity becomes smaller. At 30 min averaging interval the mean CV below and above 10 mm/h are 1.7 % and 1.2 % respectively, but they increase to 6.6 % and 3.5 % at 2 min averaging interval. The maximum CV at 2 min averaging interval are 13 % and 6.8 % for lower (< 10 mm/h) and higher (> 10 mm/h) rainfall intensity respectively. Even though these values are significantly less than what we observed from Fig. 9 when the rainfall intensity is less than 1 mm/hr, they are still high considering the required accuracy defined in standard guidelines of urban hydrological modelling practice. For example, the

current urban drainage verification guideline (WaPUG, 2012) in the UK defines a maximum allowable deviation of 25 % to - 15 % in peak runoff demanding more accurate prediction of rainfall which is the main driver of the runoff process in urban areas. A 13% uncertainty in rainfall will result in a similar level of uncertainty in runoff prediction for a completely

impervious surface according to the well-established rational formula (Viessman and Lewis, 1995) which is still widely used for estimating design discharge in small urban catchments.

## 5. Conclusions

Geostatistical methods have been used to analyse the spatial correlation structure of rainfall at various spatial scales, but its application to estimate the level of uncertainty in rainfall upscaling has not been fully explored mainly due to its inherent complexity and demanding data requirements. In this study we presented a method to overcome these challenges and predict AARI together with associated uncertainty using geostatistical upscaling. We used a spatial stochastic simulation approach to address the combination of change of support (from point to catchment) and non-normality of rainfall observations for prediction of AARI and the associated uncertainty. We addressed the issue of scarcity in measurement points by using repetitive rainfall measurements (pooling) to increase the number of spatial samples used for variogram estimation. The methods were illustrated with rainfall data collected from a cluster of eight paired rain gauges in a $400 \times 200$ m$^2$ urban catchment in Bradford, UK. The spatial lag ranges from 21 m to 399 m. As far we are aware these are the smallest lag ranges in which spatial variability in rainfall is examined in an urban area using point rainfall measurements. We defined intensity classes and derived different geostatistical models (variograms) for each intensity class separately. We also used different temporal averaging intervals, ranging from 2 to 30 min, which are of interest in urban hydrology. To the best of our knowledge this is the first such attempt to assign geostatistical models for a combination of intensity class and temporal averaging interval. Finally, we quantified the level of uncertainty in the prediction of AARI for these different combinations of temporal averaging intervals and rainfall intensity ranges.

A summary of the significant findings are listed below:

- Several studies (e.g. Berne et al., 2004; Gebremichael and Krajewski, 2004; Krajewski et al., 2003) used a single geostatistical model in the form of variogram/correlogram for the entire range of rainfall intensity. The current study shows that for small time and space scales the use of a single geostatistical model based on a single variogram is not appropriate and a distinction between rainfall intensity classes and length of temporal averaging intervals should be made.

- The level of uncertainty in the prediction of AARI using point measurement data essentially comes from two sources; spatial variability of the rainfall and measurement error. The significance and characteristics of the measurement error observed here mainly corresponds to sampling related error of tipping bucket type rain gauges (TB error) and may vary for other types of rain gauges.

- TB error decreases with increasing rainfall intensity. As a result of that, the prediction error decreases with increasing AARI. At 5 min averaging interval the CV values are as high as 80% when the AARI is smaller than 1 mm/hr and they get reduced to less than 10% when AARI is larger than 10 mm/hr

- At smaller temporal averaging intervals, the effect of both spatial variability and TB error is high, resulting in higher uncertainty levels in the prediction of AARI. With increasing temporal averaging interval the uncertainty

becomes smaller as the spatial correlation increases and the TB error reduces. At 2 min temporal averaging interval the average CV in the prediction of peak AARI is 6.6 % and the maximum CV is 13 % and they are reduced to 1.5 % and 3.6 % respectively at 30 min averaging interval.

- TB error at averaging intervals of less than 5 min, especially at low intensity rainfall measurements, is as significant as spatial variability. Hence proper attention to TB error should be given in any application of these measurements, especially in urban hydrology where averaging intervals are often as small as 2 min.

Although the spatial stochastic simulation method used in this study needs more computational power (a summary on computation power is presented as supplementary material) than block kriging, it is a robust approach and allows data transformation during spatial interpolation and aggregation. Such data transformation is important because rainfall data are not normally distributed for small temporal averaging intervals. The pooling procedure used in this study helps provide a solution to meet the data requirements for geostatistical methods as it extends the available information for variogram estimation. Commenting on the minimum number of measurement points needed to employ this method is difficult, because like any other geostatistical interpolation method, the efficiency of this method also heavily depends on reliable estimation of the geostatistical model (variogram). Hence, it basically comes down to the question of whether or not a given measurement network can produce a meaningful variogram. As mentioned, Webster and Oliver (2007) advised that around 100 measurement points are needed to adequately estimate a geostatistical model. But there is no single universal rule to define the minimum number of bins and the number of samples for each bin to produce a reliable variogram. Further, since pooling sample variograms of repeated measurements would produce a multiplication of spatial lags, the size of the available data set would also play a role in deciding the minimum number of measurement points.

An urban catchment of this size needs rainfall data at a temporal and spatial resolution which is higher than the resolution of most commonly available radar data (1000 m, 5 min).   In addition the level of uncertainty in radar measurements would be much higher than that of point measurements, especially at a small averaging interval (< 5 min, Seo and Krajewski, 2010; Villarini et al., 2008), which are often of interest in urban hydrology. Hence, experimental rain gauge data similar to the ones used in this study are crucial for similar studies focused on small urban catchments.

Results from this study can be used for uncertainty analyses of hydrologic and hydrodynamic modelling of similar sized urban catchments in similar climates as it provides information on uncertainty associated with rainfall estimation which is arguably the most important input in these models. This information will help to differentiate input uncertainty from total uncertainty thereby helping to understand other sources of uncertainty due to model parameter and model structure. This estimate of the relative importance of uncertainty sources can help to avoid false calibration and force fitting of model parameters (Vrugt et al., 2008).  This study can also help to judge optimal temporal averaging interval for rainfall estimation of hydrologic and hydrodynamic modelling especially for small urban catchments.

**Acknowledgements**

This research was done as part of the Marie Curie ITN - Quantifying Uncertainty in Integrated Catchment Studies project (QUICS). This project has received funding from the European Union's Seventh Framework Programme for research, technological development and demonstration under Grant Agreement no. 607000.

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

**Table 1: Summary of events which yielded more than 10 mm rainfall and lasted for more than 20 min with summary statistics of event peaks (derived at 5 min temporal averaging interval) from all stations.**

| Event ID. | Date | Network average duration (h) | Network average intensity (mm/h) | Network average rainfall (mm) | Summary statistics of peaks between different stations (mm/h) | | | |
|---|---|---|---|---|---|---|---|---|
| | | | | | Mean | Std. Dev | Max | Min |
| 1 | 18/04/2012 | 6.33 | 2.20 | 13.9 | 5.10 | 0.550 | 6.02 | 4.74 |
| 2 | 25/04/2012 | 6.42 | 2.55 | 16.3 | 7.05 | 0.751 | 8.32 | 5.92 |
| 3 | 09/05/2012 | 8.92 | 1.79 | 16.0 | 5.10 | 0.537 | 5.97 | 4.74 |
| 4 | 14/06/2012 | 6.83 | 1.99 | 13.6 | 5.25 | 0.636 | 6.04 | 4.74 |
| 5 | 22/06/2012 | 11.4 | 2.39 | 27.3 | 12.7 | 1.72 | 15.4 | 9.67 |
| 6 | 06/07/2012 | 4.42 | 5.31 | 23.4 | 38.5 | 4.52 | 42.9 | 30.5 |
| 7 | 06/07/2012 | 3.25 | 3.23 | 10.5 | 7.20 | 0.679 | 8.46 | 5.93 |
| 8 | 07/07/2012 | 1.50 | 7.84 | 11.8 | 74.4 | 9.27 | 86.5 | 61.9 |
| 9 | 19/07/2012 | 3.08 | 3.35 | 10.3 | 12.7 | 2.01 | 14.5 | 9.74 |
| 10 | 15/08/2012 | 2.00 | 7.96 | 15.9 | 43.0 | 3.69 | 47.8 | 37.5 |
| 11 | 14/05/2013 | 7.92 | 2.14 | 17.0 | 8.08 | 1.20 | 9.55 | 6.09 |
| 12 | 23/07/2013 | 1.75 | 6.51 | 11.4 | 37.7 | 2.09 | 42.6 | 35.7 |
| 13 | 27/07/2013 | 8.17 | 4.34 | 35.5 | 26.6 | 1.23 | 27.5 | 23.8 |

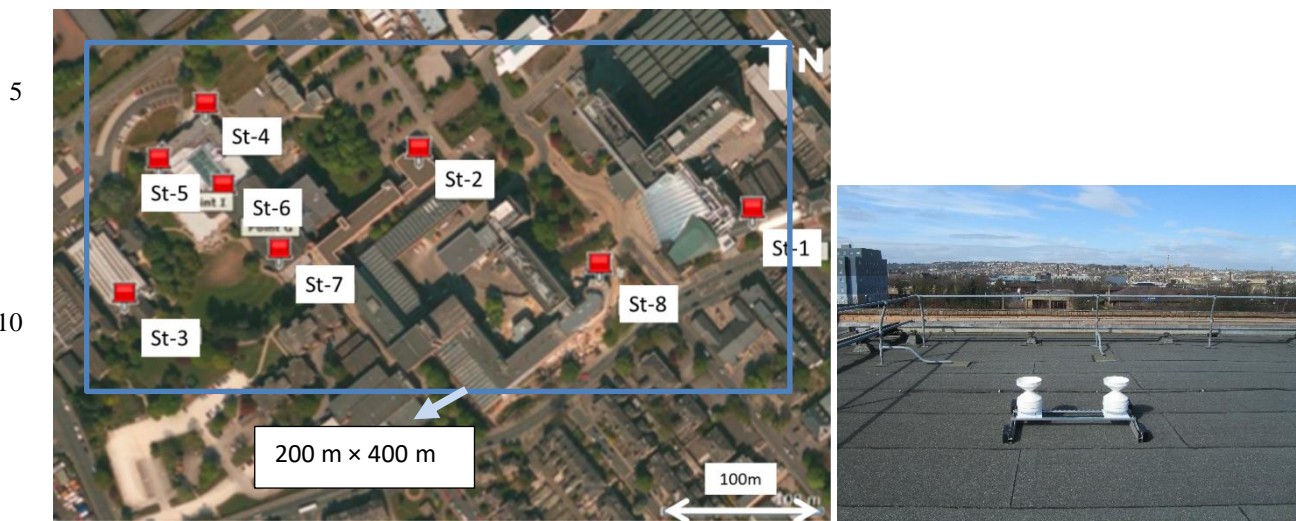

**Figure 1: (Left) A aerial view of - rain gauge network covering an area of $200 \times 400$ m$^2$ at Bradford University, UK. (Right) A photograph of paired rain gauges at station 6.**

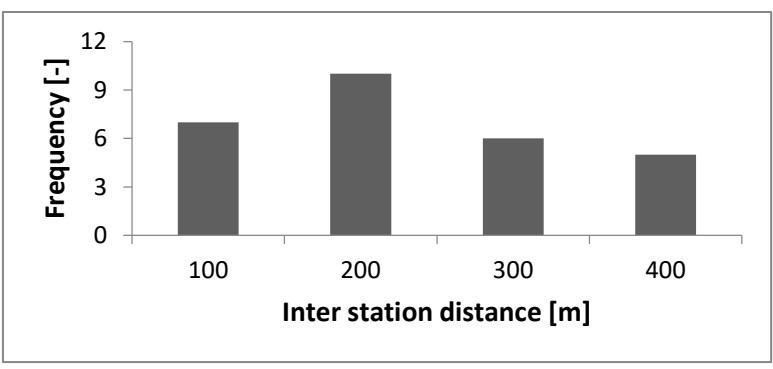

**Figure 2: Histogram with class interval width of 100 m showing frequency distribution of inter-station distances (m)**

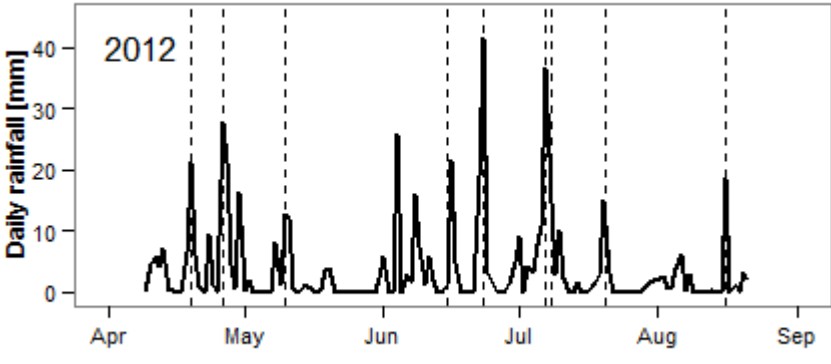

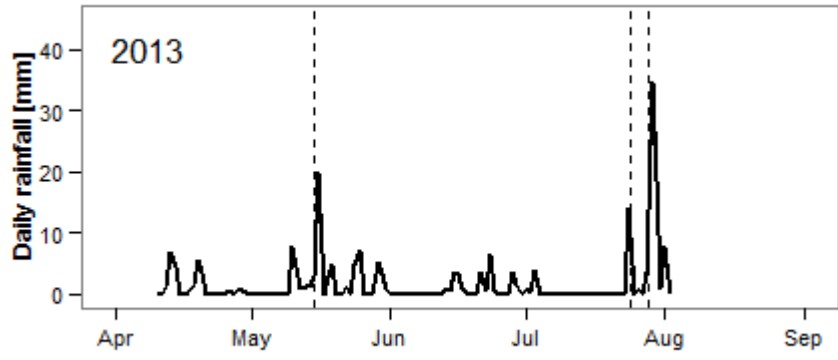

**Figure 3 : Time series of network average daily rainfall in the two seasons of 2012 and 2013 with vertical dashed lines indicating the events presented in Table 1**

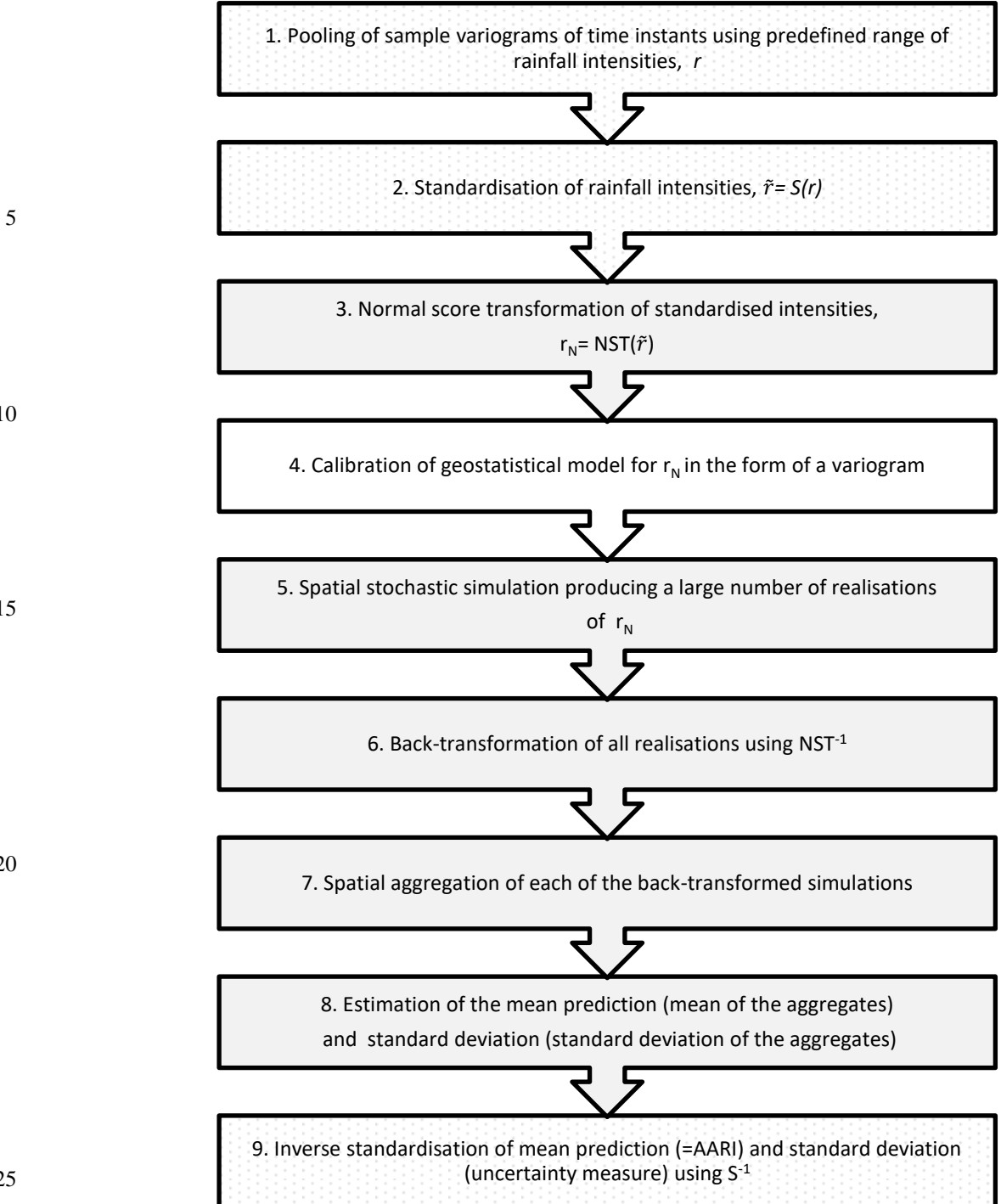

**Figure 4: Step by step procedure developed in this study to predict AARI and associated level of uncertainty. Boxes highlighted in dots indicate the steps to resolve the problem of scarcity in measurement locations, grey boxes show the steps introduced to address non-normality of rainfall data.**

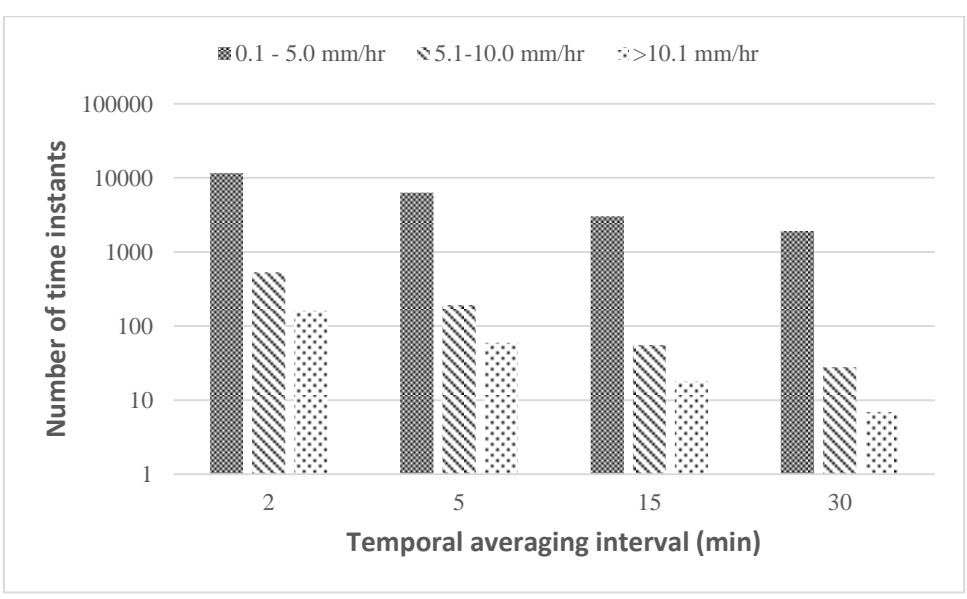

**Figure 5 : Number of time instants for each temporal averaging interval and rainfall intensity class combination.**

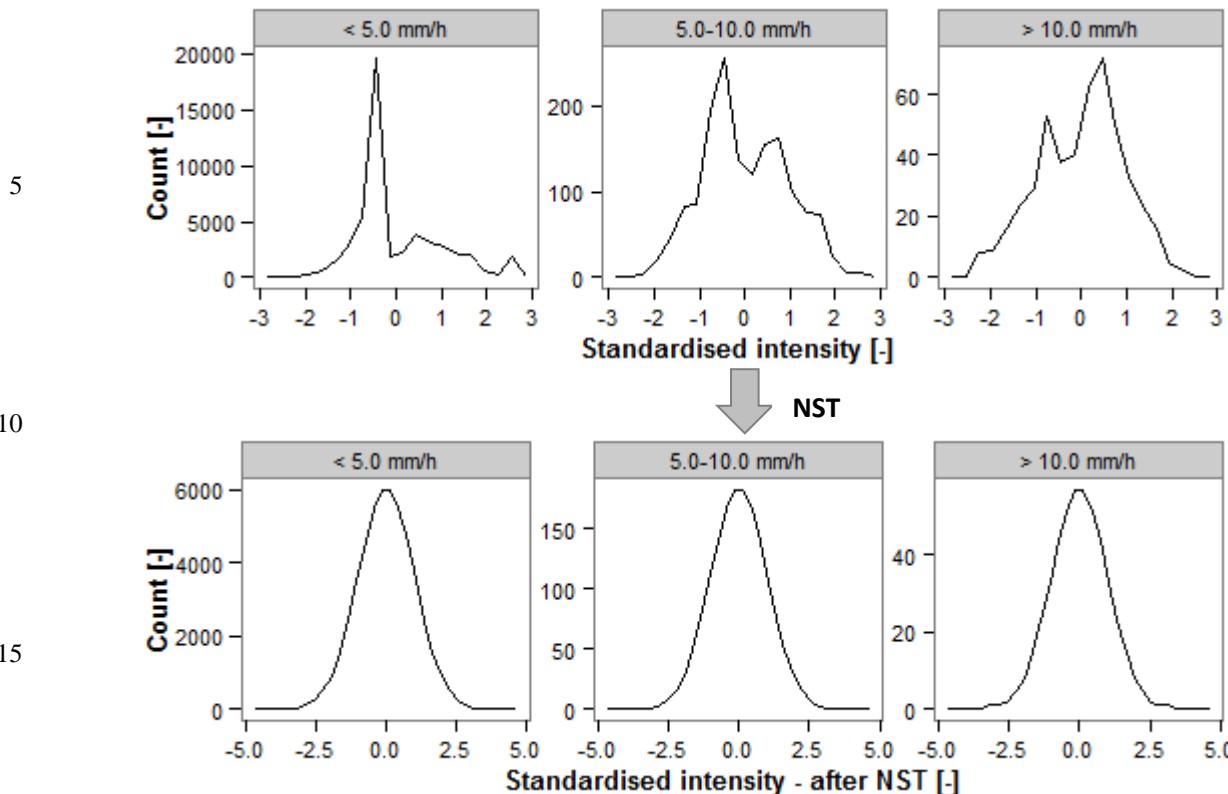

**Figure 6: Distribution of standardised rainfall intensity for different rainfall intensity classes at a temporal averaging interval of 5 min before (upper part) and after (lower part) normal score transformation (NST).**

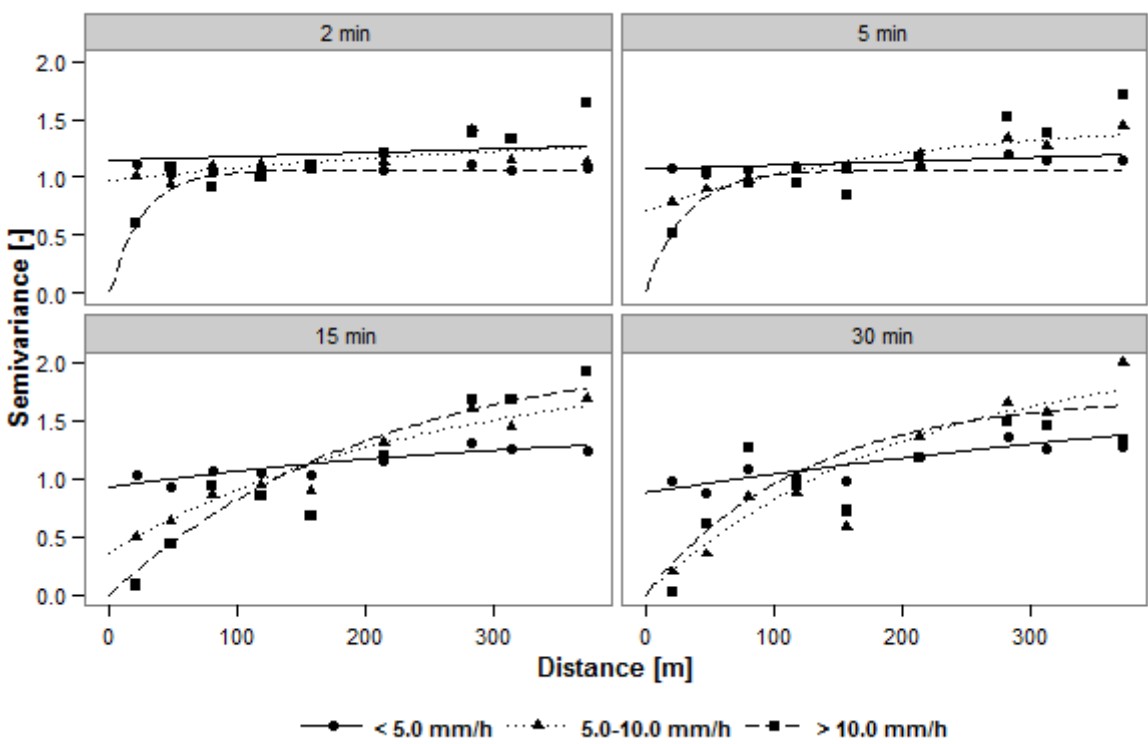

**Figure 7: Calculated variograms for each intensity class within each temporal averaging interval**

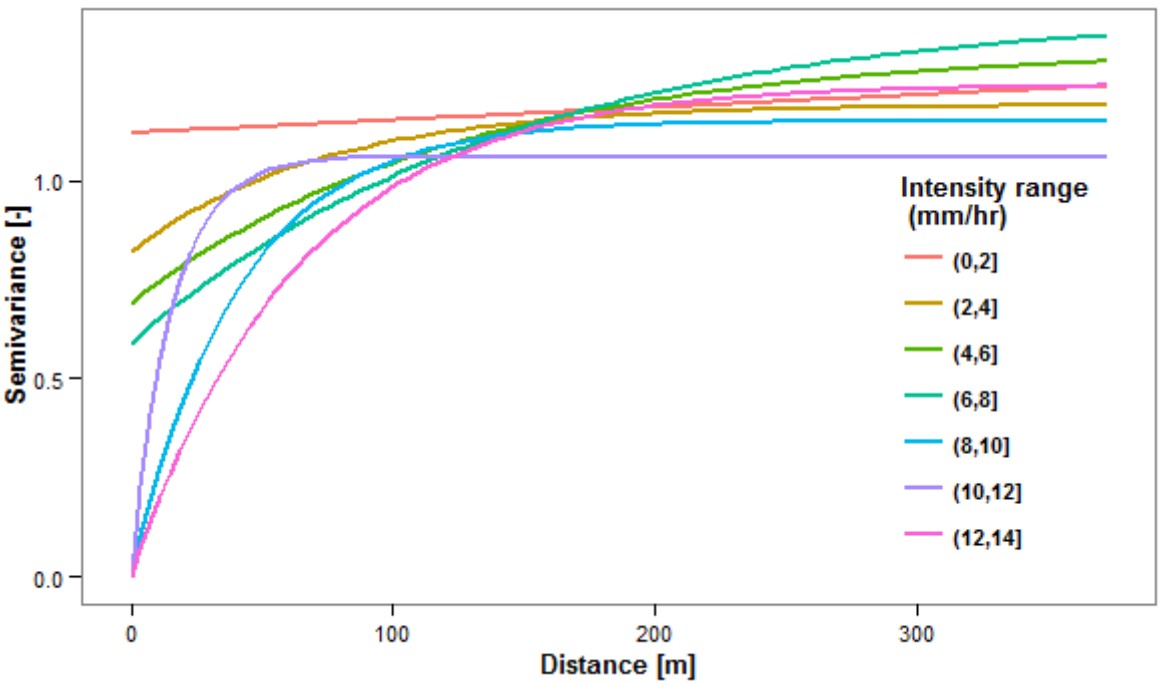

**Figure 8: Calculated variograms for a narrower range of intensity at 5 min averaging interval.**

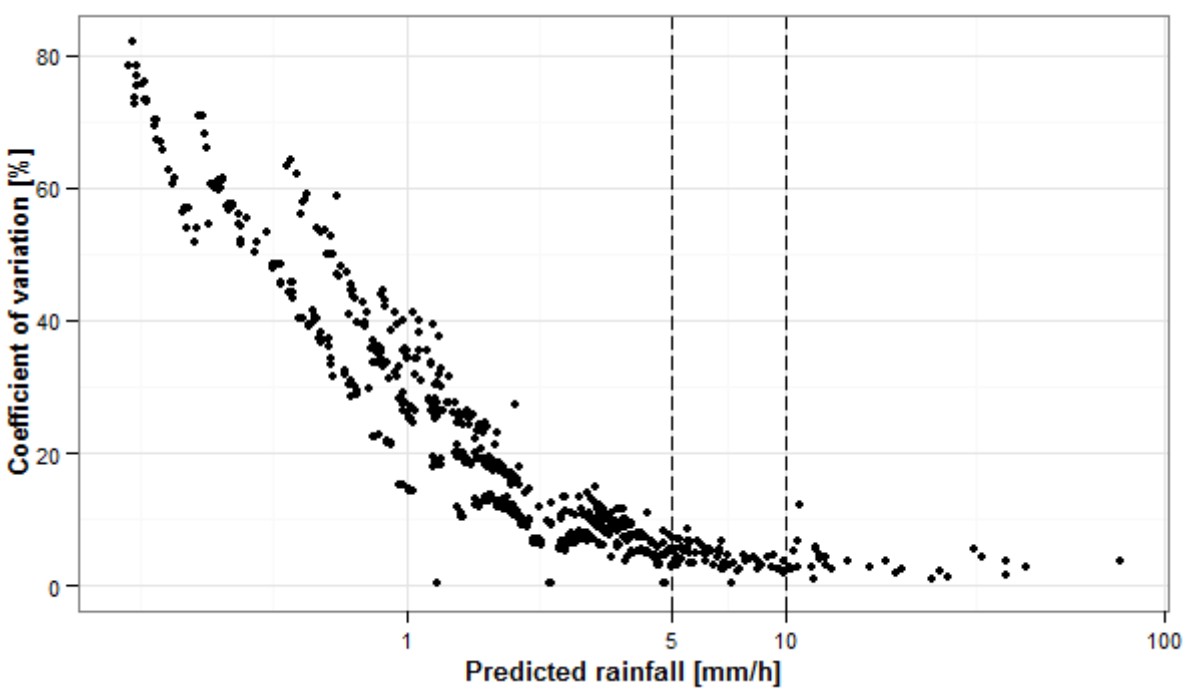

**Figure 9: AARI prediction error CV (%) values against predicted AARI for averaging interval of 5 min.**

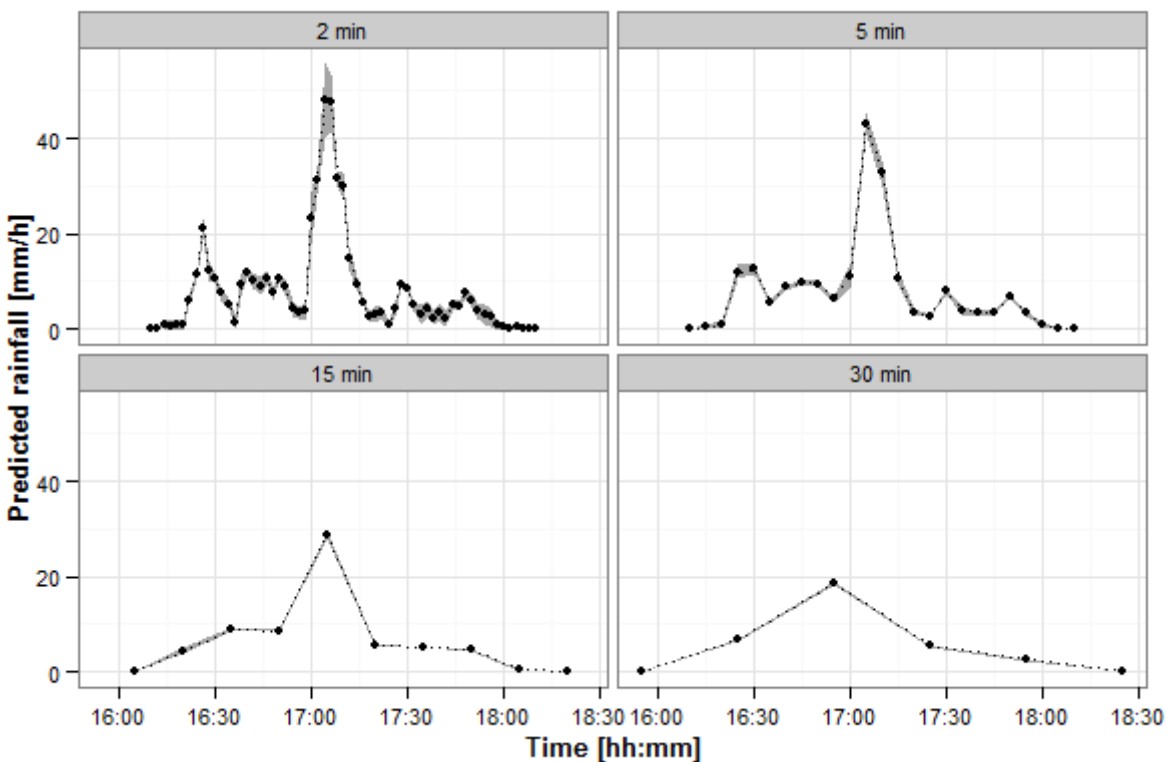

**Figure 10: Predictions of AARI (indicated by points) together with 95 % prediction intervals (indicated by grey ribbon) for rainfall event 11 for different averaging intervals.**

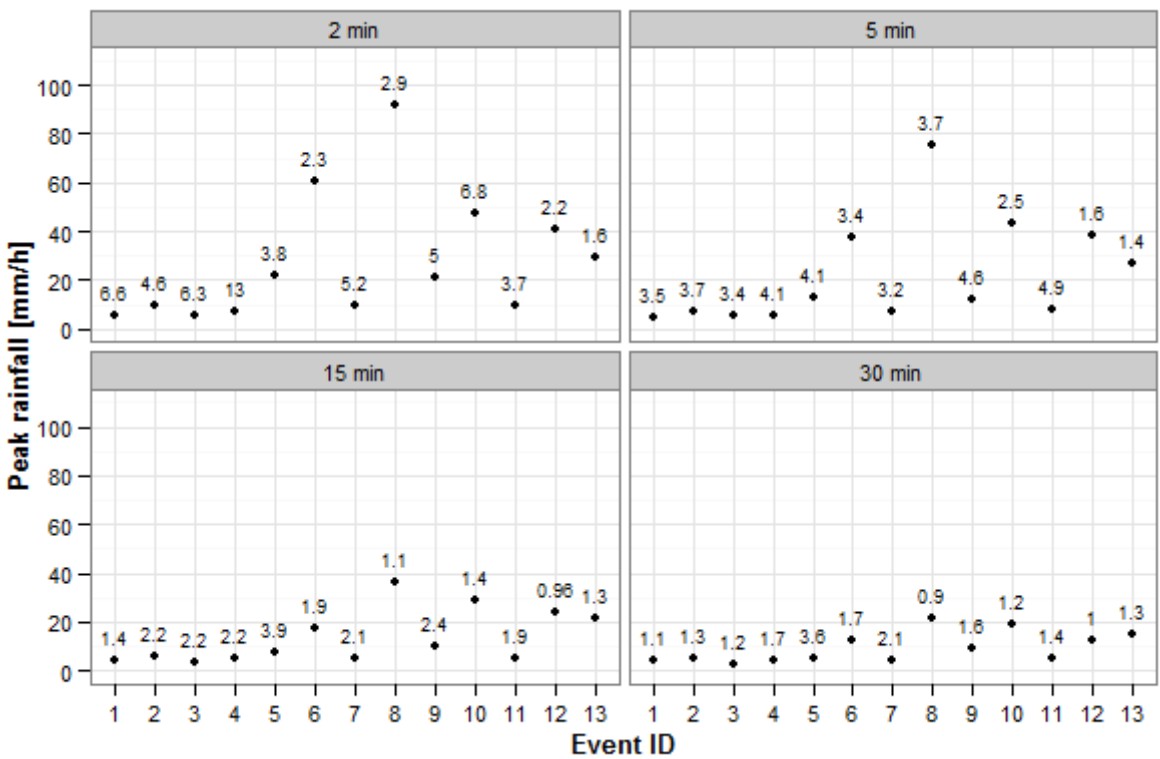

**Figure 11 : Predictions of event peaks of AARI (indicated by points) together with labels indicating corresponding CV (%) values.**