# Peer review of "Geostatistical upscaling of rain gauge data to support uncertainty analysis of lumped urban hydrological models"

_Hydrology and Earth System Sciences, 2016_

## Referee Comment (RC1) · Anonymous Referee #1 · 5 Jul 2016

General comments

The goal of the paper is the analysis of the uncertainty of the areal interpolation of precipitation data from a dense gauge network with a high temporal resolution. The topic itself is relevant and currently discussed, in particular, in the context of radar measurements. From this point of view it is very interesting to analyze such a very dense gauge network with respect to the influence of the spatial variability/measurement errors on the areal interpolation. The paper is well structured and the methods used are (in most cases) clearly presented using a step wise explanation. However, a major problem of the study is the sample size of only 13 events (of 2 years). As these events are highly variable as shown in table 1, the uncertainty of the entire study is rather high. Although

there is a pooling method presented that tries to overcome this issue, the underlying assumption might still be problematic (see comment [5: 5]). Furthermore, many descriptive results are presented, but their consequences are rarely discussed. There are also a lot of trivial results in the paper. It should become much clearer, what is a logical consequence of the precipitation structure and what is an actually new result of this study. For example, temporal averaging will always reduce precipitation peaks. Finally, the language could be more precise because several sentences are too fuzzy.

Specific comments

[Page: Line]

[2: 19] "Since rainfall can vary over space significantly, any method for scaling up the point rainfall measurements adds uncertainty on top of existing measurement error." The measurement error is not explained so far. It should be introduced as it is a major aspect in the rest of the paper.

[Fig 5] For a better comparison, the classes of the histogram should be the same as the classes used later for the variogram.

[2: 32] "…not always…" Rainfall intensity values are almost never normally distributed

[3: 32] "During the dynamic calibration…" How did you identify the volume error per tip using a dynamic calibration? The entire section is not clear: "Every long set of data…if the differences…" What is the long set and what kind of differences?

[3: 8] "… to obtain a more normally …" After the transformation they are perfectly normally distributed.

[4: 14]: How are the events defined? What are the criteria of the end of the event; if all stations show zero precipitation? Is there a minimum separation time of two events? Or will a few minutes without rainfall already separate the events? Why does the event need to be at least 20 min?

[5: 5] "The underlying assumption..." This is your major assumption for the pooling based on the trade-off between independence and number of available time instants for the three defined classes. However, you did not show any analysis to validate the independence assumption. Instead, literature is cited, which shows the dependence of the spatial correlation on the intensity. How can you be sure that the influence is small enough to be disregarded, and that the pooling procedure does not mess up your further analysis?

[5: 11] "As expected, with increasing temporal averaging..." This is not only expected, this is obvious for an aggregation process of rainfall.

[5: 12-13] This is also not surprising due to the skewed intensity distribution of precipitation. It should become clearer in this part [5:11-13], that the results are just natural characteristics of precipitation.

[5: 14] "...only eight..." Where does this number come from? In figure 5 these are way more than eight for the 30 min average.

[6: 26] "It is negligible small..." Isn't that contradicting to the later parts where the uncertainty of the tipping bucket error is analyzed (for example in [12:15]).

[6:29-7:22] In the entire chapter it becomes not clear what kind of stochastic simulation was performed (conditional/unconditional) and which method was used to obtain the 500 simulation results.

[7: 10] Kriging would be possible, if the back-transformation of the individual points was performed before the averaging. (No block kriging, but ordinary kriging of single points (25 x 25m grid)

[8: 1]"...is spatial aggregation of each and every simulation..." Should it be of "each time step"?

[8: 16] How can you be sure that this effect is caused by measurement errors? Couldn't the nugget effect be also caused by the pooling technique, that is, by mixing different

time steps; or by a high variability in the natural precipitation process? (see Remark [9: 7-21])

[9: 2] Is there a reason why the content of Habib et al. is mentioned explicitly and not of Villarini et al?

[9: 7-21] The important point of the interpretation of the variograms is not clearly stated. The variograms actually show, that for short time periods < 5 min (except for high intensities), there is almost no spatial correlation, that is the field is just random. If the nugget (explained here as tipping bucket error) is almost as high as the sill, there are two options. First, there is just no spatial correlation at the regarded distance, or the spatial correlation of the field cannot be detected by the tipping buckets because of the measurement error. There is also a very weak correlation (even for high aggregations) for the intensities smaller 5.0 mm/h. How can you be sure, that the nugget comes from the tipping bucket error, and does not represent a very high spatial variability of the natural precipitation at very short distances?

[10: 16] "Here it can be noted..." As the precipitation intensities are never uniformly distributed, this effect is a logic consequence.

[10: 30] This chapter should be rewritten, could be shortened and included in the next one. As explained in the last sentence of the paragraph, it is difficult to compare the standard deviation of different absolute values. Figure 10 is rather useless, as the intervals (uncertainty) cannot be read. A table including the standard deviations in addition with the CVs for the single events could help.

[11: 12-26] What is the actual goal of that chapter? Has there any kind of significance testing be performed? As there is one very large CV value <10 mm (2 min) out of six, the comparison between the means might be biased. If this value was an outlier, would the result still be that considerable? There seem to be a tendency, but the sample size is very small and thus, there is a lot of uncertainty in this result.

[11: 22] "This is fairly high. . ." How did you judge that? 25% to -15 % of the peak runoff is your reference. But what is the expected influence of the uncertainty of the rainfall when estimating the runoff from the precipitation? This is an important question for the judgement as one should now the necessary accuracy of the input precipitation and not the total uncertainty of the runoff.

[11: 26] "Hence a better trade-off . . ." What does this actually mean? How could this be achieved? There is a link missing between the shown problems and the actual application. Later, in the conclusion, it becomes clearer, but in this chapter this sentence is kind of fuzzy.

[12: 6-10] It should be mentioned, that the result of "peak" intensities are explained here.

[12: 24] ". . . radar measurements . . . would be much higher. . ." Please, give some references here.

[12: 27] You did not show explicitly the advantage of the paired rain gauges. Unless the improvements are shown in the main chapters, they should not be part of the conclusions.

Technical corrections

[2: 7] . . .where time series of areal . . . "are" needed.

[2: 27] . . . [8,9] . . . citations missing

[4: 9] "That" is because. . .

[8:5/8] Index Error: px should be pi

[9: 21] . . . "look" similar. . .

---

## Referee Comment (RC2) · Anonymous Referee #2 · 15 Jul 2016

General

This paper suggest a method to upscale rainfall intensity from point scale to catchment scale. The authors suggest a Kriging-based stochastic method for this upscaling; a method that allows an uncertainty estimation of the areal rainfall. I found the method suggested by the authors very interesting. I think that it will be of interest mainly for the hydro-meteorologist community (dealing with weather radar estimations) rather than for the urban hydrologist community. Main problem in the paper is the short period of observation (two seasons) that is expressed in a low confidence in the presented results. I found that some key papers dealing with dense rain-gauge networks and rainfall variability in the past were not mentioned and that some trivial aspects discussed in the

past are repeated in here. I would suggest the authors to thoroughly revise the paper as follow: make the upscaling method as the main focus of the paper, explain it with much further details and with a much clearer language. Use the data you have from the dense rain-gauge network as a case study to demonstrate how you can upscale rainfall for the catchment / weather radar scale and show the advantages of estimating uncertainties with the method you are suggesting. Please find below my specific comments, following by some general comments.

Specific comments

[Page:Lines]

[2:4-6] Please support this statement with a reference.

[2:19-21] Since you are dealing with rainfall uncertainty for small domains, taking into consideration the rainfall spatial and temporal variability, I am strongly recommend you to check also the following papers that were published in the recent years, which I think you will find them all relevant to your study:

Peleg N, Ben-Asher M, Morin E. Radar subpixel-scale rainfall variability and uncertainty: lessons learned from observations of a dense rain-gauge network. Hydrol. Earth Syst. Sci. 2013 Jun 14;17(6):2195-208.

Krajewski, W. F., Kruger, A., and Nespor, V.: Experimental and numerical studies of small-scale rainfall measurements and variability, Water Sci. Technol., 37, 131–138, doi:10.1016/s0273-1223(98)00325-4, 1998.

Pedersen, L., Jensen, N. E., Christensen, L. E., and Madsen, H.: Quantification of the spatial variability of rainfall based on a dense network of rain gauges, Atmos. Res., 95, 441–454, doi:10.1016/j.atmosres.2009.11.007, 2010.

Fiener, P. and Auerswald, K.: Spatial variability of rainfall on a sub-kilometre scale, Earth Surf. roc. Land., 34, 848–859, doi:10.1002/esp.1779, 2009.

Seo, B. C. and Krajewski, W. F.: Investigation of the scale dependent variability of radar-rainfall and rain gauge error covariance, Adv. Water Resour., 34, 152–163, doi:10.1016/j.advwatres.2010.10.006, 2011.

Gebremichael, M. and Krajewski, W. F.: Assessment of the statistical characterization of small-scale rainfall variability from radar: Analysis of TRMM ground validation datasets, J. Appl. Meteorol., 43, 1180–1199, doi:10.1175/1520-0450(2004)043<1180:aotsco>2.0.co;2, 2004.

Ciach, G. J. and Krajewski, W. F.: On the estimation of radar rainfall error variance, Adv. Water Resour., 22, 585–595, doi:10.1016/s0309-1708(98)00043-8, 1999.

[2:27] Reference typo.

[2:32] I would even claim that it is rare to find locations where the rainfall is normally distributed.

[3:3] word "often" can be deleted.

[3:19-20] This is a very short period of observation, and winter rainfall is not represented at all. How does it affect your results? Moreover, in [4:7-12] you mention the large difference between the two years of observation. This imply that the climatology was different between the two years and therefore I would expect that it will somehow influence on the rainfall spatial correlation. With only two years, the variability expected for the spatial rainfall structure cannot be represented and this should be discussed.

[Fig. 1] The recommendation is to mount rain-gauges elevated at 1.2 m above ground, where here the gauges seem to be placed directly on ground level (roof top). I wonder how this affects rainfall intensity estimations.

[3:29] What about time drift? Did you reset the loggers every 4-5 weeks to avoid this problem?

[4:1-5] This is not clear to me. If I got it right, you are comparing paired gauges for

each rain event by accumulating the rainfall over the gauges and comparing them and if the difference exceed the 4Peleg, N., Marra, F., Fatichi, S., Paschalis, A., Molnar, P. and Burlando, P., 2016. Spatial variability of extreme rainfall at radar subpixel scale. Journal of Hydrology.

[5:10] three rainfall intensity classes were SUBJECTIVELY selected? What was the criterion?

[6:8] n.d.?

[6:26] "It is negligibly small in the case of rainfall intensity data". Not necessarily, Peleg et al. (2013) reported a 0.92 nugget for 1-min time resolution. I am not sure that this can be neglected.

[7:13-22] spatial stochastic simulation – Please provide more information about how the actual stochastic engine works. How was the variogram reproduced?

[7:22] I would except that a finer grid would improve your predictions. Especially when very high rainfall intensity is recorded over the domain, as a rapid (exponential) decrease in rainfall intensity from the center away is expected (for convective rainfall at least).

[Equation 6 and 7] Equation 6 – doesn't it also need to be divided by m? I think the readers are aware to the statistics of mean and standard deviation thus you can probably delete these equations.

[8:17] "nugget effect . . . at zero lag distance due to measurement error" – are you sure it is just because of a measurement error? Rainfall variability exists between pair gauges, even for a 1 m distance, at least for temporal resolution of 1-5 min. Please check over the paper I have mentioned at comment [2:19-21] above. Your statement is repeated several times again during the text. I would at least discuss the possibility of having the nugget effect as more than a simple representation of rain-gauges measurement error.

[9:18-21] I would argue that reason why "the behaviour of spatial correlation against

rainfall intensity class is not very distinctive" in your study is due to the short period of data you have used.

[Equation 8] CV equation is also commonly known, I suggest you to delete this equation as well.

[12:23] X-band radar can reach 250 m and 3 min resolution. I think it is good enough for small urban catchments.

[12:29-31] for a similar climate.

[General comment 1] I think your method suggested for rainfall upscaling is really interesting and can be very useful to some of the reader. However I, as a reader, would like to have more information, such as: What is the minimum number of rain-gauges required for a given catchment is order to apply your suggested upscaling (e.g. are 3 gauges over 1 km2 are enough?)? What should be the spatial configuration of these rain-gauges over the domain? Another question- if you would leave one of the gauges out of your analysis, how it would affect the results (what is the sensitivity of the network design?)?

[General comment 2] You stated that the stochastic model require some "computational demands". Can you give some details? How much time is needed to run the stochastic model per time step? What kind of machine do you need to use? It can be given as supplementary information but some readers might be interested to know.

[General comment 3] The paper is oriented for the urban hydrology community, but if fact who will benefits the most from your method are hydro-meteorologists that are often looking for different methods to upscale rainfall observation from point scale to weather radar scale. Consider changing the title and addressing this as well. I think that due to the lack of sufficient length of observation, you should focus more on the method and who can benefit from it and less on your results.

---

## Short Comment (SC1) · 12 Aug 2016

Complete author list: Müller, Hannes; Callau Poduje, Ana; Fangmann, Anne; Plötner, Stefan; Shehu, Bora; Uniyal, Bhumika

(all from the Institute of Water Resources Management, Hydrology and Agricultural Hydraulic Engineering, Leibniz Universität Hannover, Hanover, 30167, Germany)

This review results from six reviewers, all interested in the topic of the manuscript. Due to the number of six additional reviewers, the review is organized in major comments, suggestions and technical notes.

Brief summary: The manuscript deals with uncertainties resulting from upscaling of
rainfall data. Upscaling hereby includes temporal aggregation as well as the determination of areal rainfall from point measurements. The topic is highly interesting and the investigation can be a good contribution to this field. However, we think that the manuscript can be improved significantly in the methods and the results part.

Major Comments:

Data:

p4 l7-21 The measuring period is quite short with two summer periods in 2012 and 2013. However, for such a dense network this is often the case. The two periods differ clearly and hence it is difficult to draw general conclusions from results. Open questions to the data set:

- From the two periods events are selected using a certain threshold. Why are events selected and why is the investigation not carried out for the whole observation periods? The results shown later are not based on/related to events. Is there a need for the event separation?

- A threshold of 10 mm network average rainfall depth and a minimum of 20 min rainfall duration were chosen for the event selection. How have these thresholds been chosen? The chosen thresholds can lead to exclusion of convective events with high rainfall amounts at one station, but no rainfall at the other stations. This is also indicated by the durations of the resulting events, ranging from 1.5 h to 11.4 h, which are more typical for stratiform events and not convective ones. Indeed these convective ones are crucial for urban hydrology and the resulting uncertainty in spatial upscaling is very high. Have convective events be excluded from the investigation by the chosen thresholds?

- How is the network average rainfall depth calculated for the event selection? In the introduction several methods are discussed. Is ordinary kriging applied here?

Methodology:

p5 l12 What is pooled – events or single time steps? In the text before, time steps (p5 l1) and events (p5 l6-7) are mentioned. If time steps are pooled (and not events), later for one event different variograms may be used due to different intensities of the single time steps in the event, right?

p7 l14 What is spatial stochastic simulation? All results are based on this method, so an explanation in the text is necessary (not only a reference). Is it applied as a subsequent step to the ordinary kriging or instead of the ordinary kriging? What is the stochastic simulation based on?

Results:

p8 l25 The nugget-to-sill ratio is interpreted as measurement error, decreasing with an increasing temporal aggregation. The movement of events is ignored, which could significantly contribute to this ratio. With 2 min time steps, the event has reached one (pair) of the gauges, after 30 min all gauges are influenced by the event. This explanation should be implemented. Is it possible with other measurements (wind velocity,...) to exclude / quantify this effect? Also, can the whole nugget effect be described as measurement error from the author's point of view?

Conclusions:

General comment: Some conclusions are trivial (e.g. the intensity becomes less with increasing averaging interval), and there could be more conclusions out of the investigation. What is the message to the urban hydrologic modelers? How can this uncertainty be involved in the calibration process/result discussion? Is the uncertainty greater/smaller than other uncertainties in urban hydrological modeling? Is it useful to take this uncertainty into account, if others are higher? What are results of other investigations concerning areal rainfall uncertainties? Is it assumed, that the uncertainty increases with increasing area sizes in the lumped model? What is the recommendation for rain gauges number per square kilometer from this investigation? How sensitive are the results, if the station density/combination of stations is changed in the investigation? The measurement set-up is quite dense. Can general conclusions be drawn to less dense networks (and how)? Can the results be validated with an urban hydrologic model?

Suggestions:

Title: The title doesn't fit to the content of the manuscript. There are no urban hydrological models applied. Also, if no kriging is applied (not sure about that, see major comment p7 l14), it's not an geostatistical upscaling. The title "Estimation of uncertainties from spatial and temporal upscaling on an urban scale" is therefore misleading.

Introduction:

p2 l19-25 There exist other methods for the estimation of uncertainties (bootstrapping,...), which should be mentioned in this context. Indeed, a focus should put on these methods, their comparisons and a reasonable decision for the applied method should be given at the end.

p2 l8-19 In the introduction a number of interpolation methods are mentioned, which are not used afterwards in the investigation. They could be left out.

Methodology:

p5 l11 How have the thresholds for the pooling been chosen?

p6 l2 Methods and results (Fig. 6) are mixed.

p6 l10 The method of NST could be explained briefly.

p6 l17 Step 4 is not a step, only a description, and can be moved to step 5.

p7 l4 explanation for q is missing

p7 l11-13 Even if block kriging cannot be applied, an ordinary kriging with subsequent back-transformation and averaging could. What is the disadvantage in comparison to the spatial stochastic simulation? Since block-kriging is not applied, it could be left out.

[Figure]

p8 l9 With the standard deviation and the mean of areal rainfall intensities the re-standardisation is carried out. For the former standardisation the standard deviation and the mean of point values have been used (since it is not clear, what has been pooled (see comment p5 l12), we assume time steps). Shouldn't be standard deviation and mean for standardization and re-standardisation be from the same type, so either from area or point values?

Results:

p8 16-17 The nugget is interpreted as spatial rainfall variability or measurement error. The network offers the great possibility to have rain gauge pairs with distances of 1 m. Measurement errors have been excluded before by the paired measured time series. So the spatial variability can be shown for these small distances, or why should this not be possible?

p8 l28 Since all errors have been excluded under the usage of the paired time series, the word TB error is somehow misleading. "Sampling error" could be more appropriate.

p10 l3 Showing the CV would be more effective then showing the standard deviation in Fig. 8. An increasing of the standard deviation with an increasing intensity is trivial (which is even stated on p11 l7-10). Also, a logarithmic plot would be useful.

p10 l31-32 Design on peak rainfall intensity: The intensity AND the duration are impor-tant and both are used for the dimensioning of e.g. a sewer system.

p11 l1 Fig. 10 Maybe it would be useful to use violin plots instead of only the standard deviation to show the uncertainty.

p11 l12 Fig. 11 The readability of the figure could be increased by colors and/or drawing only contours, not filling them. Showing the means as functions, not as fixed values, would give better conclusions. The high mean for 2 min, <10 mm/h is caused by only one extreme CV ($\sim$13 %) and is not representable.

Conclusion:

p12 l10-13 Decreasing peaks due to aggregation in time is trivial and not a conclusion.

p12 l32-33 "This information can help to avoid false calibration and force fitting of model parameters" It remains unclear, how the result of the investigation can be used for the avoidance of the before mentioned issues.

Technical notes:

p7 l5 "locationsxl" to "locations xl" and "locationsx0" to "locations x0"

p8 l5 Eq. (6) "pi" and not "px", also the division by "m" is missing – Since this is a simple equation, it could left out, also Eq. (7)

p8 l6 Eq. (7) The term under the root has to be squared.

p8 l25 "nugget-to-still ratio" to "nugget-to-sill ratio" (several times)

p9 l3 "in their study found" to "found in their study"

p9 l15 "(2003) in their" to "(2003) found in their"

p10 l11-13 In Fig. 9 event 10 is shown, not event 11 (regarding to Fig. 10).

p12 l22 "methods to a certain extent" – fuzzy phrase

––––––––––––––––––––––

---

## Author Comment (AC1) · 27 Sep 2016

**General comments**
The goal of the paper is the analysis of the uncertainty of the areal interpolation of precipitation data from a dense gauge network with a high temporal resolution. The topic itself is relevant and currently discussed, in particular, in the context of radar measurements.

From this point of view it is very interesting to analyse such a very dense gauge network with respect to the influence of the spatial variability/measurement errors on the areal interpolation. The paper is well structured and the methods used are (in most cases) clearly presented using a step wise explanation. However, a major problem of the study is the sample size of only 13 events (of 2 years). As these events are highly variable as shown in table 1, the uncertainty of the entire study is rather high. Although there is a pooling method presented that tries to overcome this issue, the underlying assumption might still be problematic (see comment [5: 5]). Furthermore, many descriptive results are presented, but their consequences are rarely discussed. There are also a lot of trivial results in the paper. It should become much clearer, what is a logical consequence of the precipitation structure and what is an actually new result of this study. For example, temporal averaging will always reduce precipitation peaks. Finally, the language could be more precise because several sentences are too fuzzy.

*Authors:* We thank the reviewer for the professional and thorough review of this paper. We have attempted to address all the comments and our response is listed below.

….However, a major problem of the study is the sample size of only 13 events (of 2 years). As these events are highly variable as shown in table 1, the uncertainty of the entire study is rather high.

*Reply:* The major technical comment from the reviewer is that our analyses are based on a small sample size of only 13 events. We would like to clarify that the entire ten months of rainfall data from 8 locations were used for the development and calibration of the geostatistical model. Webster and Oliver (2007) suggested around 100 samples to reliably estimate a variogram model. Even in the case of 30 min temporal averaging interval and > 10 mm/hr (where we had the least observations) we had 196 sampling to calculate the variogram which is sufficiently larger than 100. Hence all our variogram models (based on which further results are derived) are stable and reliable. The event separation (ref Table 1) is used only for the analyses presented in Sections 4.2.3 and 4.2.4. We apologise if it was not clear in the manuscript that the entire 10 months of data was used and we will add additional text in the revised manuscript to clarify this.

*Furthermore, to test the stability of variogram estimation we carried out the following experiment. We randomly selected 80% of the data from each intensity class and produced the variograms again to compare these with the variograms presented in Figure 7 of the manuscript. The variograms computed from 80% of the data are presented below in Figure C1.*

[Figure]

**Figure C1: Calculated variograms for each temporal averaging interval and for each range of intensity within a temporal averaging interval using randomly selected 80% of the data from each subclasses**

[Figure]

**Figure 1: Calculated variograms for each temporal averaging interval and for each range of intensity within a temporal averaging interval**

*Comparison of Figure C1 with Figure 7 shows that these are very similar. This analysis supports our claim that the variograms shown in Figure 7 are stable and an adequate representation of the rainfall spatial variation for each intensity class and temporal averaging interval.*

*We acknowledge that the data cover only 10 months i.e. two summer periods in 2012 and 2013. However, for previous studies using such a dense network the duration of data collection is similar (e.g.: 15 months - Ciach and Krajewski, 2006; 16 months - Jaffrain and Berne, 2012). These durations are a reflection of the practical and funding issues to maintain such networks operating accurately for extended periods. The characteristics of our data are comparable with (Ciach and Krajewski, 2006; Fiener and Auerswald, 2008) as these studies also used rainfall data from warm months to investigate the spatial correlation structure.*

Although there is a pooling method presented that tries to overcome this issue, the underlying assumption might still be problematic (see comment [5: 5]).

Reply: *The underlying assumption of pooling is discussed in detail in our reply to specific comment 5: 5.*

Furthermore, many descriptive results are presented, but their consequences are rarely discussed. There are also a lot of trivial results in the paper. It should become much clearer, what is a logical consequence of the precipitation structure and what is an actually new result of this study. For example, temporal averaging will always reduce precipitation peaks. Finally, the language could be more precise because several sentences are too fuzzy.

Reply: *We will attempt to deal with the concerns stated by the referee. Superfluous sentences will be removed and some will be rewritten to reduce the attention paid to trivial results, to improve clarity and discuss the consequence of the analysis carried out in the paper. Details of what is proposed are described below.*

*Trivial results removed, or reduced*
*Several sentences will be removed and some will be rewritten to try and reduce the level of trivial results, including the ones mentioned by the reviewer in comments. For more details please refer to the specific comments [5: 11], [5: 12-13] and [10: 16].*

*Improvement of clarity and language*
*The following parts will be modified to give a clearer explanation, as indicated in specific comments of the reviewer:*
*Quality control of rainfall data: See our reply to comment [3:32]*
*Event separation: See our reply to comment [4: 14]*
*Stochastic simulation: See our reply to comment [6:29-7:22]*
*Interpretation of the variograms: See our reply to comment [8: 16]*

*New results/novelty*
*Because of the nature of this study more emphasis is given to the methodology. We believe that this methodology is novel for the following reasons:*

- *In literature, geostatistics has been used to analyse the spatial correlation structure of rainfall at various spatial scales, but its application to estimate the level of uncertainty in rainfall upscaling has not been fully explored mainly due to its inherent complexity and demanding data requirements. In this study we attempted to address these challenges which include the use of repetitive rainfall measurements (pooling) to increase the spatial samples artificially.*

- *We used the spatial stochastic simulation to address the combination of change of support (from point to catchment) and non-normality for prediction of rainfall and associated uncertainty. To the best of our knowledge this has not been done previously.*

- *We classified intensity classes and derived different geostatistical models (variograms) for each class separately. On top of that we also used different temporal averaging intervals. To the best of our knowledge no previous study attempted to assign geostatistical models for a combination of intensity class and temporal averaging interval.*

*In addition we presented and discussed the following new results:*

- *Spatial correlation structure of rainfall at a spatial extent of 200m × 400m has been presented. To the best of our knowledge no previous studies have analysed spatial correlation structure of rainfall at such a small spatial extend.*

- *The current study shows that for this spatial scale the use of a single geostatistical model based on a single variogram is not appropriate. Instead, different variograms for different rainfall intensity classes should be used. This is a key finding of the study.*

- *In addition to visual interpretation we also presented quantification of uncertainty in upscaled rainfall predictions in terms of CV for peaks of rainfall intensity. Previous studies (e.g. Villarini et al. 2008) have provided such quantification of uncertainty only using simpler error measures (normalized root mean squared error and normalized mean absolute error) while we use geostatistical approaches that take the effect of spatial correlation into account.*

**Specific comments**
**[Page: Line]**

[2: 19] "Since rainfall can vary over space significantly, any method for scaling up the point rainfall measurements adds uncertainty on top of existing measurement error."
The measurement error is not explained so far. It should be introduced as it is a major aspect in the rest of the paper.

*Reply: A short introduction (given below) on measurement error mainly focusing on tipping bucket errors will be added in our revised manuscript.*

*"....The uncertainty mainly comes from two sources; uncertainty due to measurement errors and uncertainty associated with spatiotemporal variability of rainfall. The characteristics of measurement errors can vary depending on the rain gauge type. For example, errors associated with tipping bucket rain gauges which is one of the widely used rain gauge type range from errors due to wind, wetting, evaporation, and splashing (Fankhauser, 1998; Sevruk and Hamon, 1984) to errors due to its sampling mechanism (Habib et al., 2001)....."*

[Fig 5] For a better comparison, the classes of the histogram should be the same as the classes used later for the variogram.

*Reply: The rainfall intensity class intervals were the same for histograms (Fig. 5) and variograms (Fig. 7).*

[2: 32] ": : :not always: : :" Rainfall intensity values are almost never normally distributed

*Reply: Thank you for pointing this out. This will be corrected in the revised manuscript.*

[3: 32] "During the dynamic calibration: : :" How did you identify the volume error per tip using a dynamic calibration? The entire section is not clear: "Every long set of data: : :if the differences: : :" What is the long set and what kind of differences?

*Reply: This section will be modified in the revised manuscript as given below to give more precise information on how the quality control was carried out using paired setup.*

*Quality control procedures were performed prior to statistical analysis, taking advantage of the paired gauge setup. The paired gauge design provides efficient quality control of the rain gauge data records as it helps to identify the instances when one of the gauges fails, and to flag the periods of missing or incorrect data (Ciach and Krajewski, 2006). During the dynamic calibration of all rain gauges in the laboratory before the deployment, it was identified that the highest and lowest values of the calibration factors for the tipping bucket size are 0.196mm and 0.204mm, respectively, resulting in a maximum potential error of ± 4 % due to the observed variation per tip for any two of the rain gauges used in this study. It was therefore decided that this is the maximum acceptable difference in cumulative rainfall that could be accepted between any pair of gauges. Sets of cumulative rainfall data corresponding to specific events from the paired gauges were checked against each other and if the (absolute) difference in cumulative rainfall was greater than 4 %, that complete set was identified as unreliable (e.g. due to partial clogging caused by debris) and the data from both gauges were removed from further analysis.*

[3: 8] ": : : to obtain a more normally : : :" After the transformation they are perfectly normally distributed.

*Reply: Thank you for pointing out this out. This will be corrected in the revised manuscript*

[4: 14]: How are the events defined? What are the criteria of the end of the event; if all stations show zero precipitation? Is there a minimum separation time of two events?
Or will a few minutes without rainfall already separate the events? Why does the event need to be at least 20 min?

*Reply: There are two conditions for defining an event: the yield should be more than 10 mm and the event duration should be larger than 20 minutes. If all stations show zero precipitation based on 5 min averaging interval it is considered as the end of an event and there is no minimum separation time between events. Figure 3 and Table 1 show that except for events 6 and 7 there are no events close to each other.*

*We chose a 20 min window in order to get a sufficient number of data values when temporal averaging intervals of more than 2 min are used in the analysis. For example, a 10 min event gives two data values when using a 5 min averaging interval and gives only 1 data value for 15 min and 30 min averaging intervals. Hence in order to have at least two data values for all temporal averaging intervals examined, a minimum event duration of 20 minutes is needed. Table 1 shows that the lowest event duration in the collected data was 1.5 hours. Hence all events had at least 45 data values for a 2 minute averaging interval and at least 3 data values for a 30 minutes averaging interval.*

*Also note that, as we mentioned already, the event separation (ref Table 1) is used only for the analyses presented in Sections 4.2.3 and 4.2.4. Hence these criteria don't leave out any data in the development and calibration of the geostatistical models.*

[5: 5] "The underlying assumption: : :" This is your major assumption for the pooling based on the trade-off between independence and number of available time instants for the three defined classes. However, you did not show any analysis to validate the independence assumption. Instead, literature is cited, which shows the dependence of the spatial correlation on the intensity. How can you be sure

that the influence is small enough to be disregarded, and that the pooling procedure does not mess up your further analysis?

*Reply: We agree with the concerns expressed by the reviewer. This is the reason why we introduced step 2 (6: 1-5) to treat each and every time instant within a subset, by individually calculating their mean and standard deviation. Although variograms are derived only for the whole subset, step 2 (before geostatistical upscaling) and step 9 (after geostatistical upscaling) ensure that the probabilistic model is adjusted for each time instant separately, based on the mean and standard deviation for that particular time instant. Effectively, we assume the same correlogram for time instants of the same subclass, not the same variogram. Although this does not justify the assumption of similar spatial correlation structure with in the pooled classes, it at least relaxes the assumption of the same variogram within subclasses.*

*The similarity in Figures C1 and 7 also shows that the characteristics of the data from subsets are consistent with those of the entire pooled class. Moreover, several studies (e.g. Dirks et al., 1998; Ly et al., 2011; Tao et al., 2009) used only a single geostatistical model in the form of single variograms/correlograms for the entire range of rainfall intensity. The current study shows that for small time and space scales the use of a single geostatistical model based on a single variogram is not appropriate. This is a key finding of this study. We agree that with narrower intervals the assumption of consistency in spatial variability would be more realistic. But with the available data we had to find a compromise with the number of time instants. We believe that using three intensity subclasses is a workable compromise. Based on Figure C3 (in our reply to comment 8:16), where variograms are produced for narrower subclasses, we conclude that the variograms shown in Figure 7 are good representations of the average spatial variability conditions for corresponding intensity classes.*

[5: 11] "As expected, with increasing temporal averaging: : :" This is not only expected, this is obvious for an aggregation process of rainfall.

*Reply: We agree with this comment.*
*"As expected, with increasing temporal averaging the number of time instants t reduces." will be corrected as "Number of time instants t reduces with increasing temporal averaging intervals due to the aggregation process."*

[5: 12-13] This is also not surprising due to the skewed intensity distribution of precipitation. It should become clearer in this part [5:11-13], that the results are just natural characteristics of precipitation.

*Reply: We agree that the dominance of low intensity rainfall is a natural phenomenon. It will be corrected in the manuscript.*
*"Figure 5 also shows that the higher the intensity, the smaller the t. There is a large difference between t for lower and higher intensity ranges which shows the dominance of lower intensity (0.1-5.0 mm/h) rainfall over the recording periods" will be corrected as*
*"The natural characteristic of rainfall data results in the dominance of lower intensity rainfall (0.1-5.0 mm/h) over the recording period."*

[5: 14] ": : :only eight: : :" Where does this number come from? In figure 5 these are way more than eight for the 30 min average.

*Reply: Thank you for pointing out this. We apologise for the mismatch between text and figure. This histogram shows the number of time instants (t) × number of stations (eight). It will be corrected in the revised manuscript to show the number of time instants (t) to be consistent with the text. Actually there are seven (not eight) time instants where intensity exceeds 10 mm/hr at 30 min temporal averaging interval. The new figure is presented below. This figure will replace Figure 5 in the revised manuscript.*

[Figure]

**Figure C2: Calculated variograms for 5 min averaging interval and for a narrower range of intensity.**

[6: 26] "It is negligible small: : :" Isn't that contradicting to the later parts where the uncertainty of the tipping bucket error is analysed (for example in [12:15]).

*Reply: We wanted to argue that any physical micro-scale spatial variation between rain gauges, which theoretically may be one of the causes of the nugget effect, is negligibly small for this case. We did not mean to claim that the nugget itself is negligibly small. We accept that the particular sentence was poorly constructed. We also think that this sentence can be removed as the nugget effect is discussed in detail at a later stage in the manuscript.*

[6:29-7:22] In the entire chapter it becomes not clear what kind of stochastic simulation was performed (conditional/unconditional) and which method was used to obtain the 500 simulation results.

*Reply: We accept that this section requires more detail. We will modify this section in the revised manuscript as given below.*

*The output from spatial stochastic simulation is a set of alternative realisations ('possible realities') of rainfall at user-defined grid points. The differences among these realisations are used as a measure of uncertainty. Hence it involves the following two steps*
*1.        Definition of grid cells (25m × 25m in this case)*
*2.        Production of substantial number of possible realisations (500 in this case) for each time instant using stochastic simulation based on the corresponding geo statistical model (variograms in this case).*

*We used the stochastic simulation conditioned on the available rainfall data at measured stations. The grid size and number of simulations were selected considering the spatial resolution of available measurements and computational demand. It was observed that neither a finer grid nor more simulations improves the results much.  Increasing the resolution to 10 m × 10 m only improves the standard deviation of the prediction by less than 5% in most cases while almost doubles the computational time.*

[7: 10] Kriging would be possible, if the back-transformation of the individual points was performed before the averaging. (No block kriging, but ordinary kriging of single points (25 x 25m grid)

*Reply: This might work for the prediction but not for the prediction uncertainty of spatially aggregated rainfall, which we also needed to quantify in this study. That is why we turned to spatial stochastic simulation.*

[8: 1]": : :is spatial aggregation of each and every simulation: : :" Should it be of "each time step"?

*Reply: It should be 'spatial aggregation of each and every simulation (realisation)' (i.e. 500 per time instant). This yields 500 values of the areal mean per time instant. We will modify Section 3.5 to give a clearer explanation of spatial stochastic simulation (refer comment [6:29-7:22]) which we believe that it will help to understand the subsequent steps better.*

[8: 16] How can you be sure that this effect is caused by measurement errors? Couldn't the nugget effect be also caused by the pooling technique, that is, by mixing different time steps; or by a high variability in the natural precipitation process? (see Remark [9: 7-21])

*Reply: The two other possible reasons for the nugget effect as mentioned by the reviewer are discussed in detail separately below.*

1. *Pooling of time instants*
   *We do not think pooling is the reason for the nugget effect. We pooled sample variograms from different time instants. But that cannot cause a nugget effect; it just gives back the average nugget of those for all time instants. If none of the variograms from different time instants have a large nugget, then the pooled one also will not have a large nugget.*

[Figure]

**Figure C3: Calculated variograms for 5 min averaging interval and for a narrower range of intensity.**

*Furthermore, Figure C3 shows the behaviour of variogram models for narrower intensity classes ranging from 0 to 14 mm/hr for the 5 min averaging interval. The highest intensity class is limited to ≥12 - <14 mm/hr as for further narrower ranges (i.e ≥14 - <16 mm/hr and so on) there are not enough sample points to produce a meaningful variogram. The variograms for the narrower classes show that the assumption of similar spatial variability*

*with in a pooled subset is stronger. Hence the effect of variation caused by pooling is reduced. Comparing this figure with Figure 7 not only shows the behaviour of the nugget effect against the intensity as seen from Figure 7, it also makes clear that there is a clear decreasing trend in the nugget as intensity increases. This also indicates that pooling is not responsible for the pattern of the nugget effect.*

2. *High variability in the natural precipitation process*
   *Please refer to our response to the comment [9: 7-21]*

[9: 2] Is there a reason why the content of Habib et al. is mentioned explicitly and not of Villarini et al?

*Reply: The main aim of (Habib et al., 2001) was to investigate the sampling error of tipping bucket measurements, whereas (Villarini et al., 2008) was a later study and derived a similar conclusion as a part of their findings. Among the two only (Habib et al., 2001) discuss the sampling errors of tipping bucket rain gauges extensively. Hence their work is explicitly mentioned.*

[9: 7-21] The important point of the interpretation of the variograms is not clearly stated. The variograms actually show, that for short time periods < 5 min (except for high intensities), there is almost no spatial correlation, that is the field is just random. If the nugget (explained here as tipping bucket error) is almost as high as the sill, there are two options. First, there is just no spatial correlation at the regarded distance, or the spatial correlation of the field cannot be detected by the tipping buckets because of the measurement error. There is also a very weak correlation (even for high aggregations) for the intensities smaller 5.0 mm/h. How can you be sure, that the nugget comes from the tipping bucket error, and does not represent a very high spatial variability of the natural precipitation at very short distances?

*Reply: We accept that the nugget effect could be due to a combination of micro-scale spatial variability and measurement error. Since we cannot quantify the nugget effect caused by measurement error we cannot prove quantitatively that the nugget is only due to the measurement error.*

*But with regarding the clear increasing trend of nugget against (a) decreasing temporal averaging interval and (b) increasing intensity range, we believe this trend is as a result of similar trend in sampling related error of tipping bucket (TB error). If it is due to spatial variability of rainfall then such trends expected to be consistent at greater distance too. But the variograms show no consistent trend against intensity range at greater distance. In a previous similar study (Ciach and Krajewski, 2006), where the behaviour of spatial correlation against rainfall intensity was analysed, they also could not find a consistent trend and concluded that such trends are not consistent.*

*In summary,*
1. *Nuggets corresponds to each variogram could be a combined effect of measurement error and spatial variability. We cannot comment on individual contribution as we cannot quantify them.*
2. *The trends in the nugget against a. rainfall intensity and b. temporal averaging interval correspond well with TB error. Hence these trends in the nugget can be attributed to the TB error.*

*We will include the above discussion in the revised manuscript.*

[10: 16] "Here it can be noted: : :" As the precipitation intensities are never uniformly distributed, this effect is a logic consequence.
*Reply: Thank you pointing this out, this will be corrected in the revised manuscript.*

*"Here it can be noted that in this event this effect decreases the event peak AARI from around 50 mm/h to around 20 mm/h as the temporal averaging interval increases from 2 min to 30 min." will be corrected as "As a result of temporal aggregation the event peak gets reduced from around 50 mm/h to around 20 mm/h as the temporal averaging interval increases from 2 min to 30 min"*

[10: 30] This chapter should be rewritten, could be shortened and included in the next one. As explained in the last sentence of the paragraph, it is difficult to compare the standard deviation of different absolute values. Figure 10 is rather useless, as the intervals (uncertainty) cannot be read. A table including the standard deviations in addition with the CVs for the single events could help.

*Reply: Sections 4.2.3 and 4.2.4 will be combined into one section in the revised manuscript as suggested by the reviewer. We agree that the standard deviation in Figure 10 is too small to be read. Figure 10 will be modified to include CV instead of standard deviation. The modified figure is given below (Figure C4) followed by the discussion that we plan to include (please refer to our response to comment 11: 12-26 regarding the sample size)*

[Figure]

**Figure C4 : Predictions of event peaks of AARI (indicated by points) together with labels indicating corresponding CV (%) values.**

*Rainfall event peaks are of significant interest in urban hydrology as most of the hydraulic structures in urban drainage systems are designed based on peak discharge which is often derived from peak rainfall. Hence it is important to consider the uncertainty in prediction of peaks of AARI. Figure 10 presents predicted peaks of AARI for all 13 events presented in Table 1, together with labels indicating corresponding CV (%) values. Since CV is a normalised measure it gives a clearer picture of behaviour of error against prediction. The peak intensities range from 6 mm/h to 92 mm/h at 2 min averaging interval and this range narrows down to 3 mm/h – 21 mm/h at averaging interval of 30 min as a result of temporal aggregation. Temporal aggregation from 2 min to 30 min also results in the*

*reduction of CV as expected. The highest CV at 2 min averaging intervals is 13% for event 4 and it gets reduced to 1.7% at 30 min averaging interval. But it can also be noted that events 5, 6, 8 and 11 show their highest CV at 5 min averaging interval and not at 2 min averaging interval. Tracking back these events, they indeed show more spatial variation over 5 min period compared to 2 min period around the peak.*

*Although the sample size is not large enough to derive a firm conclusion about the behaviour of CV against the intensity ranges, there is a tendency that CV is higher when the predicted intensity is lower than 10 mm/h.  This compliments what is observed from the variograms in Fig. 7, where below 10 mm/h the TB error becomes high. Hence it is expected to see a higher uncertainty characterised by CV at lower rainfall intensity (< 10 mm/h) especially at a temporal averaging interval of 2 min where the TB error is at its highest. But when the temporal averaging interval is 30 min where the TB error is at its lowest, the difference between CV for lower (< 10 mm/h) and higher (> 10 mm/h) intensity becomes smaller. At 30 min averaging interval the mean CV below and above 10 mm/h are 1.7 % and 1.2 % respectively, but they increase to 6.6 % and 3.5 % at 2 min averaging interval. The maximum CV at 2 min averaging interval are 13 % and 6.8 % for lower (< 10 mm/h) and higher (> 10 mm/h) rainfall intensity respectively. This is fairly high considering the required accuracy defined in standard guidelines of urban hydrological modelling practises. For example urban drainage verification guidelines (WaPUG, 2012) in UK set a maximum allowable deviation of 25 % to -15 % in peak runoff demanding more accurate prediction of rainfall which is the main driver of the runoff process.*

[11: 12-26] What is the actual goal of that chapter? Has there any kind of significance testing be performed? As there is one very large CV value <10 mm (2 min) out of six, the comparison between the means might be biased. If this value was an outlier, would the result still be that considerable? There seem to be a tendency, but the sample size is very small and thus, there is a lot of uncertainty in this result.

*Reply: We plan to remove Figure 11 from the manuscript and discussions will then be based on revised Figure 10 (please refer our response to comment 10: 30). Although we do not think peak prediction at event 8 is an outlier, we accept the fact that the sample size is too small to derive a firm conclusion. This will also be mentioned clearly in the revised manuscript. We also performed the test without event 8, which results in the same trend. But the average CV for the range < 10 mm/hr is reduced to 5.3 % from 6.6% at 2 min averaging interval.*

*We also tried to relax the condition of minimum rainfall yield from 10 mm to 5 mm to derive more events. This gave 19 events instead of 13. Similar analysis was carried out on these events and the results are presented in Figure C5 below,*

[Figure]

**Figure C5: Coefficient of variation plotted against predicted peaks of AARI for the 19 events**

*The figure shows similar behaviour to Figure 11 where a higher uncertainty (characterised by CV) is seen for lower rainfall intensity (< 10 mm/h) at a temporal averaging interval of 2 min, where the TB error is at its highest. The trend is clearer this time as the highest three CV values at 2 min averaging interval correspond with lower intensity rainfall. The differences between average CV corresponding to lower (<10 mm/hr) and higher (>10 mm/hr) intensity rainfall are 7.2% and 4.0%, respectively. This is mainly due to the dominance of TB error corresponding to lower intensity rainfall at 2 min averaging interval. But when the temporal averaging interval is 30 min and the TB error is minimal this effect is not there anymore. In fact the average CV is slightly higher (by 0.8%) for lower intensity rainfall than for higher intensity rainfall. But the difference is too small to derive a conclusion for the 30 min averaging interval.*

[11: 22] "This is fairly high: : :" How did you judge that? 25% to -15 % of the peak runoff is your reference. But what is the expected influence of the uncertainty of the rainfall when estimating the runoff from the precipitation? This is an important question for the judgement as one should know the necessary accuracy of the input precipitation and not the total uncertainty of the runoff.

*Reply: It will need a hydrologic modelling set up for this catchment to find out the exact influence of rainfall and rainfall uncertainty in urban runoff. This is beyond the scope of this study. Hence a simpler approach using Rational Formula to calculate the surface runoff is used here to form an idea of the influence. According to the rational formula (Q=kCIA) runoff has a linear relationship with rainfall intensity given a constant area and a constant runoff coefficient. Hence a 13% deviation in rainfall intensity would result in 13% deviation in surface runoff, which is high given the allowed limit of -15% to 25%, also not to forget other sources of uncertainty due to parameter and model structure. Furthermore, in several recent studies (Gires et al., 2012; Schellart et al., 2012) the effect*

*of this small scale (< 1km) variability of rainfall on urban runoff peak has been proven to be significant.*

[11: 26] "Hence a better trade-off : : :" What does this actually mean? How could this be achieved? There is a link missing between the shown problems and the actual application. Later, in the conclusion, it becomes clearer, but in this chapter this sentence
is kind of fuzzy.

*Reply: We accept that this sentence is not clear. This sentence will be removed from this section to avoid repetitive discussion of the same.*

[12: 6-10] It should be mentioned, that the result of "peak" intensities are explained here.

*Reply: Thank you for pointing out this. We will do so in the revised manuscript.*

[12: 24] ": : : radar measurements : : : would be much higher: : :" Please, give some references here.

*Reply: The following references will be added in the revised manuscript:*
*(Seo and Krajewski, 2010; Villarini et al., 2008).*

[12: 27] You did not show explicitly the advantage of the paired rain gauges. Unless the improvements are shown in the main chapters, they should not be part of the conclusions.

*Reply: Thank you for this comment. This will be removed from the conclusion in the revised manuscript.*

**Technical corrections**
[2: 7] : : :where time series of areal : : : "are" needed.
[2: 27] : : : [8,9] : : : citations missing
[4: 9] "That" is because: : :
[8:5/8] Index Error: px should be pi
[9: 21] : : : "look" similar: : :

*Reply: Thank you for pointing out these errors. All will be corrected in the revised manuscript.*

**References**

Ciach, G. J. and Krajewski, W. F.: Analysis and modeling of spatial correlation structure in small-scale rainfall in Central Oklahoma, Adv. Water Resour., 29, 1450–1463, doi:10.1016/j.advwatres.2005.11.003, 2006.

Dirks, K. N., Hay, J. E., Stow, C. D. and Harris, D.: High-resolution studies of rainfall on Norfolk Island Part II: Interpolation of rainfall data, J. Hydrol., 208(3-4), 187–193, doi:10.1016/S0022-1694(98)00155-3, 1998.

Fankhauser, R.: Influence of systematic errors from tipping bucket rain gauges on recorded rainfall data, Water Sci. Technol., 37(11), 121–129, doi:10.1016/S0273-1223(98)00324-2, 1998.

Fiener, P. and Auerswald, K.: Spatial variability of rainfall on a sub-kilometre scale, , 859(February), 848–859, doi:10.1002/esp, 2008.

Gires, A., Onof, C., Maksimovic, C., Schertzer, D., Tchiguirinskaia, I. and Simoes, N.: Quantifying the impact of small scale unmeasured rainfall variability on urban runoff through multifractal downscaling: A case study, J. Hydrol., 442-443, 117–128, doi:10.1016/j.jhydrol.2012.04.005, 2012.

Habib, E., Krajewski, W. F. and Kruger, A.: Sampling Errors of Tipping-Bucket Rain Gauge

Measurements, J. Hydrol. Eng., 6(2), 159–166, doi:10.1061/(ASCE)1084-0699(2001)6:2(159), 2001.

Jaffrain, J. and Berne, A.: Quantification of the small-scale spatial structure of the raindrop size distribution from a network of disdrometers, J. Appl. Meteorol. Climatol., 51(5), 941–953, doi:10.1175/JAMC-D-11-0136.1, 2012.

Ly, S., Charles, C. and Degré, a.: Geostatistical interpolation of daily rainfall at catchment scale: the use of several variogram models in the Ourthe and Ambleve catchments, Belgium, Hydrol. Earth Syst. Sci., 15, 2259–2274, doi:10.5194/hess-15-2259-2011, 2011.

Schellart, A. N. A., Shepherd, W. J. and Saul, A. J.: Influence of rainfall estimation error and spatial variability on sewer flow prediction at a small urban scale, Adv. Water Resour., 45, 65–75, doi:10.1016/j.advwatres.2011.10.012, 2012.

Seo, B.-C. and Krajewski, W. F.: Scale Dependence of Radar Rainfall Uncertainty: Initial Evaluation of NEXRAD's New Super-Resolution Data for Hydrologic Applications, J. Hydrometeorol., 11(5), 1191–1198, doi:10.1175/2010JHM1265.1, 2010.

Sevruk, B. and Hamon, W. R.: International comparison of national precipitation gauges with a reference pit gauge, instruments and observing methods, Geneva., 1984.

Tao, T. A. O., Chocat, B., Liu, S. and Xin, K.: Uncertainty analysis of interpolation methods in rainfall spatial distribution–a case of small catchment in Lyon, J. Environ. Prot. (Irvine,. Calif)., 1(01), 50, doi:10.4236/jwarp.2009.12018, 2009.

Villarini, G., Mandapaka, P. V., Krajewski, W. F. and Moore, R. J.: Rainfall and sampling uncertainties: A rain gauge perspective, J. Geophys. Res. Atmos., 113(11), 1–12, doi:10.1029/2007JD009214, 2008.

Webster, R. and Oliver, M. a: Geostatistics for environmental scientists, Second edi., John Wiley & Sons, Ltd, West Sussex, England., 2007.

---

## Author Comment (AC2) · 27 Sep 2016

Anonymous referee #2 Received and published: 15 Jul 2016

**General**

This paper suggest a method to upscale rainfall intensity from point scale to catchment scale. The authors suggest a Kriging-based stochastic method for this upscaling; a method that allows an uncertainty estimation of the areal rainfall. I found the method suggested by the authors very interesting. I think that it will be of interest mainly for the hydro-meteorologist community (dealing with weather radar estimations) rather than for the urban hydrologist community. Main problem in the paper is the short period of observation (two seasons) that is expressed in a low confidence in the presented results.

I found that some key papers dealing with dense rain-gauge networks and rainfall variability in the past were not mentioned and that some trivial aspects discussed in the past are repeated in here. I would suggest the authors to thoroughly revise the paper as follow: make the upscaling method as the main focus of the paper, explain it with much further details and with a much clearer language. Use the data you have from the dense rain-gauge network as a case study to demonstrate how you can upscale rainfall for the catchment / weather radar scale and show the advantages of estimating uncertainties with the method you are suggesting. Please find below my specific comments, following by some general comments.

Authors: We thank the reviewer for the professional and thorough revision of our paper. We have attempted to address all comments listed below.

...Main problem in the paper is the short period of observation (two seasons) that is expressed in a low confidence in the presented results...

*Reply:* Please refer to our detailed response to the reviewer's specific comment on data used in the study [3:19-20]

...I found that some key papers dealing with dense rain-gauge networks and rainfall variability in the past were not mentioned

*Reply:* Please refer to our detailed response to reviewer's specific comment on literature used in the study [2:19-21]

....and that some trivial aspects discussed in the past are repeated in here.

*Reply:* We attempt to deal with the concerns stated by the reviewer. Few sentences will be removed and some will be rewritten in the revised manuscript to try and reduce the level of trivial results. They are listed below

[5: 11] "As expected, with increasing temporal averaging the number of time instants t reduces." will be corrected as "Number of time instants t reduces with increasing temporal averaging intervals due to the aggregation process."

[5: 12-13] "Figure 5 also shows that the higher the intensity, the smaller the t. There is a large difference between t for lower and higher intensity ranges which shows the dominance of lower intensity (0.1-5.0 mm/h) rainfall over the recording periods" will be corrected as "The natural characteristic of rainfall data results in the dominance of lower intensity rainfall (0.1-5.0 mm/h) over the recording period."

[10: 16] "It is expected that with increasing temporal averaging interval the local minima and maxima of AARI get smoothed out. Here it can be noted that in this event this effect decreases the event peak AARI from around 50 mm/h to around 20 mm/h as the temporal averaging interval

increases from 2 min to 30 min." will be corrected as "It is obvious that with increasing temporal averaging interval the local minima and maxima of AARI get smoothed out. As a result the event peak gets reduced from around 50 mm/h to around 20 mm/h as the temporal averaging interval increases from 2 min to 30 min"

... I would suggest the authors to thoroughly revise the paper as follow: make the upscaling method as the main focus of the paper, explain it with much further details and with a much clearer language. Use the data you have from the dense rain-gauge network as a case study to demonstrate how you can upscale rainfall for the catchment / weather radar scale and show the advantages of estimating uncertainties with the method you are suggesting.

Reply: Thank you for the suggestion. With all due respect to the reviewer's suggestion, we think that the manuscript is already heavily focused on methodology with a dedicated section which covers around 35% of the manuscript (page wise). Further, to enable the reader to follow the methodology more easily, we explained it with a step by step procedure for a general case of estimating uncertainty in upscaling of point rainfall data. We tried to keep the methodology as general as possible while also providing enough detail on how each step is applied in the case of the Bradford case study.. Furthermore, we believe that introducing the data before the methodology enables the reader to follow the methodology more easily as some of the steps require a pre-introduction to the data to explain why such step is needed. Hence we feel that the structure of the manuscript follows a logical work flow. Changing the structure to follow the reviewer's advice will lead to repetition.

Nevertheless we accept that some part of the manuscript needs more explanation, especially the spatial stochastic simulation methodology. We also agree that the language could be clearer throughout the manuscript. Hence we are planning to modify several sections/ parts of sections including the following based on reviewer's specific comments:

Quality control of rainfall data using paired gauge set up: please refer to our response to reviewer's specific comment [4:1-5]

Spatial stochastic simulation: please refer to our response to reviewer's specific comment [7:13-22]

**Specific comments**

**[Page:Lines]**

[2:4-6] Please support this statement with a reference.

*Reply:* The following references will be added in the revised manuscript: (Seo and Krajewski, 2010; Villarini et al., 2008)

[2:19-21] Since you are dealing with rainfall uncertainty for small domains, taking into consideration the rainfall spatial and temporal variability, I am strongly recommend you to check also the following papers that were published in the recent years, which I think you will find them all relevant to your study:

Peleg N, Ben-Asher M, Morin E. Radar subpixel-scale rainfall variability and uncertainty: lessons learned from observations of a dense rain-gauge network. Hydrol. Earth Syst. Sci. 2013 Jun 14;17(6):2195-208.

Krajewski, W. F., Kruger, A., and Nespor, V.: Experimental and numerical studies of small-scale rainfall measurements and variability, Water Sci. Technol., 37, 131–138, doi: 10.1016/s0273-1223(98)00325-4, 1998.

Pedersen, L., Jensen, N. E., Christensen, L. E., and Madsen, H.: Quantification of the spatial variability of rainfall based on a dense network of rain gauges, Atmos. Res., 95, 441–454, doi:10.1016/j.atmosres.2009.11.007, 2010.

Fiener, P. and Auerswald, K.: Spatial variability of rainfall on a sub-kilometre scale, Earth Surf. roc. Land., 34, 848–859, doi:10.1002/esp.1779, 2009.

Seo, B. C. and Krajewski, W. F.: Investigation of the scale dependent variability of radar-rainfall and rain gauge error covariance, Adv. Water Resour., 34, 152–163, doi:10.1016/j.advwatres.2010.10.006, 2011.

Gebremichael, M. and Krajewski, W. F.: Assessment of the statistical characterization of small-scale rainfall variability from radar: Analysis of TRMM ground validation datasets, J. Appl. Meteorol., 43, 1180–1199, doi:10.1175/1520-0450(2004)043<1180:aotsco>2.0.co;2, 2004.

Ciach, G. J. and Krajewski, W. F.: On the estimation of radar rainfall error variance, Adv. Water Resour., 22, 585–595, doi:10.1016/s0309-1708(98)00043-8, 1999.

*Reply:* Thank you for suggesting the above papers. We wanted to keep our introduction mainly focused on the methodology that we adapted for this study. Hence we followed the below order in our introduction

- Lumped hydrological models need spatial average rainfall over catchments
- Focus is on rainfall observations at points
- Thus point observations need to be scaled up
- *Review of existing methods that can do this*
- Disadvantages of these methods
- Solution that does not have these disadvantages is to take geostatistical approach
- The main challenges with geostatistical approach and how these can be dealt with

We quoted the most relevant studies wherever necessary. We accept that there are many other studies like the ones mentioned by the reviewer which can be relevant because of the similar spatial extent of the rainfall data. But most of these publications are based or partly based on radar data (areal rainfall data) and outside the main scope of our study, which is upscaling of point rainfall data and estimating associated uncertainty. Discussion of the literature where both radar data and point rainfall are used together to compare and/or analyse spatial correlation is slightly out of the context and might lead to a very lengthy Introduction and might also confuse the readers even if the spatial extend of interest is same. Therefore to keep the Introduction to the point and precise on what we wanted to introduce (upscaling of point rainfall data and associated uncertainty) we have not included literature which are completely/partly based on radar data.

However from the reviewer's suggestion we found the following papers directly relevant to our study. We thank the reviewer for suggesting these papers. We will discuss these papers in appropriate sections of the revised manuscript. We summarised how these papers are related to our study below.

Pedersen, L., Jensen, N. E., Christensen, L. E., and Madsen, H.: Quantification of the spatial variability of rainfall based on a dense network of rain gauges, Atmos. Res., 95, 441–454, doi:10.1016/j.atmosres.2009.11.007, 2010.

The aim of this paper is to quantify the uncertainties of using a single rain gauge to represent the rainfall over a  $500 \times 500$  m area. A field experiment placing nine rain gauges within an area of  $500 \times 500$  m, each representing one-ninth of the area, was used to address the issue. The variability of rainfall is studied and uncertainty in areal rainfall is estimated for different time scales. Although a

simpler approach is used in this study to estimate the uncertainty, results are still comparable to our study.

Fiener, P. and Auerswald, K.: Spatial variability of rainfall on a sub-kilometre scale, Earth Surf. roc. Land., 34, 848–859, doi:10.1002/esp.1779, 2009

In this study, a network of 13 tipping-bucket rain gauges was operated on a 1.4 km2 test site in southern Germany for four years to quantify spatial trends in rainfall depth, intensity, erosivity, and predicted runoff. Their data is comparable to ours as they also used summer half-year data for their analyses. Although they did not calculate any uncertainty in areal rainfall estimation, their analyses on spatial trend against temporal averaging interval could be of interest to our study. One of their conclusions suggests that in the longer term there is no difference in rainfall depth within the test site, but in short-time periods or for single events the assumption of spatially uniform rainfall is invalid on the sub-kilometre scale, This complements one of the findings from our study.

[2:27] Reference typo.

*Reply:* The following references will be added in the revised manuscript. (Ly et al., 2013; Mair and Fares, 2011)

[2:32] I would even claim that it is rare to find locations where the rainfall is normally distributed.

Reply: Thank you for pointing this error out. This will be corrected in the revised manuscript.

[3:3] word "often" can be deleted.

Reply: Thank you for pointing this error out. This will be corrected in the revised manuscript.

[3:19-20] This is a very short period of observation, and winter rainfall is not represented at all. How does it affect your results? Moreover, in [4:7-12] you mention the large difference between the two years of observation. This imply that the climatology was different between the two years and therefore I would expect that it will somehow influence on the rainfall spatial correlation. With only two years, the variability expected for the spatial rainfall structure cannot be represented and this should be discussed.

Reply: We acknowledge that the data cover only 10 months, i.e. two summer periods in 2012 and 2013. However, for such a dense network this is often the case (Eg: 15 months - Ciach and Krajewski, 2006; 16 months - Jaffrain and Berne, 2012) due to practical difficulties and funding issues to maintain such networks for longer periods. The characteristics of our data are comparable with those studied in Ciach and Krajewski (2006) and Fiener and Auerswald (2008) as they also used rainfall data only from warm months to investigate spatial correlation structure of rainfall data.

We would also like to point out that the entire ten months of rainfall data, from 8 locations were used for the development of the geostatistical model in the form of variograms. Webster and Oliver (2007) suggested around 100 samples to reliably estimate a variogram model. Even in the case of 30 min temporal averaging interval and > 10 mm/hr (where we had the least observations) we had 196 sampling to calculate the variogram which is sufficiently larger than 100. Hence all our variogram models (based on which further results are derived) are stable and reliable.

Furthermore, to test the stability of variogram estimation we carried out the following experiment. We randomly selected 80% of the data from each intensity class and produced the variograms again to compare these with the variograms presented in Figure 7 of the manuscript. The variograms computed from 80% of the data are presented below in Figure C1.

----- < 5.0 mm/h ···· 4···· 5.0-10.0 mm/h -- 4-- > 10.0 mm/h

Figure C1: Calculated variograms for each temporal averaging interval and for each range of intensity within a temporal averaging interval using randomly selected 80% of the data from each subclasses

Figure 1: Calculated variograms for each temporal averaging interval and for each range of intensity within a temporal averaging interval

Comparison of Figure C1 with Figure 7 shows that these are very similar. This analysis supports our claim that the variograms shown in Figure 7 are stable and an adequate representation of the rainfall spatial variation for each intensity class and temporal averaging interval.

Nevertheless we agree that different climatology of two years might influence the results and we will discuss this in the revised manuscript.

[Fig. 1] The recommendation is to mount rain-gauges elevated at 1.2 m above ground, where here the gauges seem to be placed directly on ground level (roof top). I wonder how this affects rainfall intensity estimations.

Reply: We understand that different guidelines suggest different elevations when it comes to height of a rain gauge from the surrounding ground level. In our case we followed the standard UK practice (http://www.metoffice.gov.uk/guide/weather/observations-guide/how-we-measure-rainfall) which suggests the rim of the tipping bucket to be around 0.5 m above the surrounding ground level. This clarification will be added to the manuscript.

[3:29] what about time drift? Did you reset the loggers every 4-5 weeks to avoid this problem?

Reply: Thank you for pointing this out. The data loggers were reset every 4-5 weeks during data collection to avoid any significant time drift. We will include this information in the revised manuscript.

[4:1-5] This is not clear to me. If I got it right, you are comparing paired gauges for each rain event by accumulating the rainfall over the gauges and comparing them and if the difference exceed the 4%

Peleg, N., Marra, F., Fatichi, S., Paschalis, A., Molnar, P. and Burlando, P., 2016. Spatial variability of extreme rainfall at radar subpixel scale. Journal of Hydrology.

*Reply:* This section will be modified in the revised manuscript as given below to give more precise information on how the quality control was carried out using paired setup.

Quality control procedures were performed prior to statistical analysis, taking advantage of the paired gauge setup. The paired gauge design provides efficient quality control of the rain gauge data records as it helps to identify the instances when one of the gauges fails, and to flag the periods of missing or incorrect data (Ciach and Krajewski, 2006). During the dynamic calibration of all rain gauges in the laboratory before the deployment, it was identified that the highest and lowest values of the calibration factors for the tipping bucket size are 0.196mm and 0.204mm, respectively, resulting in a maximum potential error of  $\pm 4$  % due to the observed variation per tip for any two of the rain gauges used in this study. It was therefore decided that this is the maximum acceptable difference in cumulative rainfall that could be accepted between any pair of gauges. Sets of cumulative rainfall data corresponding to specific events from the paired gauges were checked against each other and if the (absolute) difference in cumulative rainfall was greater than 4 %, that complete set was identified as unreliable (e.g. due to partial clogging caused by debris) and the data from both gauges were removed from further analysis.

[5:10] three rainfall intensity classes were SUBJECTIVELY selected? What was the criterion?

Reply: The maximum threshold value was limited to 10mm/hr to have enough time instants for the highest range (i.e. > 10 mm/hr) to produce stable variograms even at 30 min temporal averaging interval. We then decided to divide 0 - 10 mm/hr in to two equal subclasses (i.e.

---

## Author Comment (AC3) · 27 Sep 2016

**H. Müller (**mueller@iww.uni-hannover.de)

Complete author list: Müller, Hannes; Callau Poduje, Ana; Fangmann, Anne; Plötner, Stefan; Shehu, Bora; Uniyal, Bhumika
(all from the Institute of Water Resources Management, Hydrology and Agricultural Hydraulic Engineering, Leibniz Universität Hannover, Hanover, 30167, Germany)

This review results from six reviewers, all interested in the topic of the manuscript. Due to the number of six additional reviewers, the review is organized in major comments, suggestions and technical notes.

Brief summary: The manuscript deals with uncertainties resulting from upscaling of rainfall data. Upscaling hereby includes temporal aggregation as well as the determination of areal rainfall from point measurements. The topic is highly interesting and the investigation can be a good contribution to this field. However, we think that the manuscript can be improved significantly in the methods and the results part.

*Authors: We thank the reviewers for the professional and thorough revision of this paper. We have attempted to address all the comments listed below*

**Major Comments:**
Data:

p4 l7-21 The measuring period is quite short with two summer periods in 2012 and 2013. However, for such a dense network this is often the case. The two periods differ clearly and hence it is difficult to draw general conclusions from results.

*Reply: We acknowledge that the data cover only 10 months, i.e. two summer periods in 2012 and 2013. However, for such a dense network this is often the case (Eg: 15 months - Ciach and Krajewski, 2006; 16 months - Jaffrain and Berne, 2012) due to practical difficulties and funding issues to maintain such networks for longer periods. The characteristics of our data are comparable with those studied in Ciach and Krajewski (2006) and Fiener and Auerswald (2008) as they also used rainfall data only from warm months to investigate spatial correlation structure of rainfall data.*

*We would also like to point out that the entire ten months of rainfall data, from 8 locations were used for the development of the geostatistical model in the form of variograms. Webster and Oliver (2007) suggested around 100 samples to reliably estimate a variogram model. Even in the case of 30 min temporal averaging interval and > 10 mm/hr (where we had the least observations) we had 196 sampling to calculate the variogram which is sufficiently larger than 100. Hence all our variogram models (based on which further results are derived) are stable and reliable.*

*Furthermore, to test the stability of variogram estimation we carried out the following experiment. We randomly selected 80% of the data from each intensity class and produced the variograms again to compare these with the variograms presented in Figure 7 of the manuscript. The variograms computed from 80% of the data are presented below in Figure C1.*

[Figure]

**Figure C1: Calculated variograms for each temporal averaging interval and for each range of intensity within a temporal averaging interval using randomly selected 80% of the data from each subclasses**

[Figure]

**Figure 1: Calculated variograms for each temporal averaging interval and for each range of intensity within a temporal averaging interval**

*Comparison of Figure C1 with Figure 7 shows that these are very similar. This analysis supports our claim that the variograms shown in Figure 7 are stable and an adequate representation of the rainfall spatial variation for each intensity class and temporal averaging interval.*

*Nevertheless we agree that different climatology of two years might influence the results and we will discuss this in the revised manuscript.*

- From the two periods events are selected using a certain threshold. Why are events selected and why is the investigation not carried out for the whole observation periods?
The results shown later are not based on/related to events. Is there a need for the event separation?

*Reply: The event separation (ref Table 1) is used only for the analyses presented in Sections 4.2.3 and 4.2.4. We apologies if this was not clear in the manuscript and we will add some text in the revised manuscript to clarify this position.*

- A threshold of 10 mm network average rainfall depth and a minimum of 20 min rainfall duration were chosen for the event selection. How have these thresholds been chosen? The chosen thresholds can lead to exclusion of convective events with high rainfall amounts at one station, but no rainfall at the other stations. This is also indicated by the durations of the resulting events, ranging from 1.5 h to 11.4 h, which are more typical for stratiform events and not convective ones. Indeed these convective ones are crucial for urban hydrology and the resulting uncertainty in spatial upscaling is very high. Have convective events be excluded from the investigation by the chosen thresholds?

*Reply: We chose a 20 min window in order to get sufficient data when temporal averaging intervals of more than 2 min are used in the analysis. For example, a 10 min event could give two data values for a 5 min averaging interval and would give only 1 data value for 15 min and also for 30 min averaging intervals. Hence in order to have at least two data values for all the temporal averaging intervals examined minimum event duration of 20 minutes was needed. Table 1 show that the lowest event duration in the collected data was 1.5 hours. Hence all events had at least 45 data vales for 2 minute averaging interval and at least 3 data values for 30 minutes averaging interval.*

*As we mentioned in the above response the entire ten months of rainfall data from 8 locations were used for the development of the geostatistical model in the form of variograms. Hence no data are excluded from the investigation. The event separation (ref Table 1) is used only for the analyses presented in Sections 4.2.3 and 4.2.4.*

- How is the network average rainfall depth calculated for the event selection? In the introduction several methods are discussed. Is ordinary kriging applied here?

*Reply: The network average rainfall depth is calculated using arithmetic mean of the rainfall depths of 8 stations over the network. Ordinary kriging is not applied here.*

Methodology:
p5 l12 What is pooled – events or single time steps? In the text before, time steps (p5 l1) and events (p5 l6-7) are mentioned. If time steps are pooled (and not events), later for one event different variograms may be used due to different intensities of the single time steps in the event, right?

*Reply: Time steps, not events, were pooled to increase the number of spatial pairs. Please refer to p4 | 26 – p5 | 3 in the manuscript for a detailed explanation. P5 l 6-7 is just a part of an explanation on how the intensity classes are chosen.*

*Different variograms corresponding to different intensity classes can be used for a single event as a single event can contain a range of intensity values which fall into different intensity classes.*

p7 l14 What is spatial stochastic simulation? All results are based on this method, so an explanation in the text is necessary (not only a reference). Is it applied as a subsequent step to the ordinary kriging or instead of the ordinary kriging? What is the stochastic simulation based on?

*Reply: Since spatial stochastic simulation is robust to nonlinear data transformation it was used as an alternative to ordinary (block) kriging. We accept that this section requires more detail. We will modify this section in the revised manuscript as given below.*

*The output from spatial stochastic simulation is a set of alternative realisations ('possible realities') of rainfall at user-defined grid points. The differences among these realisations are used as a measure of uncertainty. Hence it involves the following two steps*
*1.        Definition of grid cells (25m × 25m in this case)*
*2.        Production of substantial number of possible realisations (500 in this case) for each time instant using stochastic simulation based on the corresponding geo statistical model (variograms in this case).*

*We used the stochastic simulation conditioned on the available rainfall data at measured stations. The grid size and number of simulations were selected considering the spatial resolution of available measurements and computational demand. It was observed that neither a finer grid nor more simulations improves the results much. Increasing the resolution to 10 m × 10 m only improves the standard deviation of the prediction by less than 5% in most cases while almost doubles the computational time.*

Results:
p8 l25 The nugget-to-sill ratio is interpreted as measurement error, decreasing with an increasing temporal aggregation. The movement of events is ignored, which could significantly contribute to this ratio. With 2 min time steps, the event has reached one (pair) of the gauges, after 30 min all gauges are influenced by the event. This explanation should be implemented. Is it possible with other measurements (wind velocity,: : :) to exclude / quantify this effect? Also, can the whole nugget effect be described as measurement error from the author's point of view?

*Reply: We accept that the nugget effect could be due to a combination of micro-scale spatial variability and measurement error. Since we cannot quantify the nugget effect caused by measurement error we cannot prove quantitatively that the nugget is only due to the measurement error.*

*But with regarding the clear increasing trend of nugget against (a) decreasing temporal averaging interval and (b) increasing intensity range, we believe this trend is as a result of similar trend in sampling related error of tipping bucket (TB error). If it is due to spatial variability of rainfall then such trends expected to be consistent at greater distance too. But the variograms show no consistent trend against intensity range at greater distance. In a previous similar study (Ciach and Krajewski, 2006), where the behaviour of spatial correlation against rainfall intensity was analysed, they also could not find a consistent trend and concluded that such trends are not consistent.*

*In summary,*
   *1. Nuggets corresponds to each variogram could be a combined effect of measurement error and spatial variability. We cannot comment on individual contribution as we cannot quantify them.*
   *2. The trends in the nugget against a. rainfall intensity and b. temporal averaging interval correspond well with TB error. Hence these trends in the nugget can be attributed to the TB error.*

*We will include the above discussion in the revised manuscript.*

*Since TB error is sampling related, other measurements (wind velocity, etc) cannot help quantifying or reducing this error.*

Conclusions:
General comment: Some conclusions are trivial (e.g. the intensity becomes less with increasing averaging interval), and there could be more conclusions out of the investigation.
What is the message to the urban hydrologic modelers? How can this uncertainty be involved in the calibration process/result discussion? Is the uncertainty greater/smaller than other uncertainties in urban hydrological modeling? Is it useful to take this uncertainty into account, if others are higher? What are results of other investigations concerning areal rainfall uncertainties? Is it assumed, that the uncertainty increases with increasing area sizes in the lumped model? What is the recommendation for rain gauges number per square kilometer from this investigation? How sensitive are the results, if the station density/combination of stations is changed in the investigation? The measurement set-up is quite dense. Can general conclusions be drawn to less dense networks (and how)? Can the results be validated with an urban hydrologic model?

_Reply:_ _Please refer to our response on specific comment about trivial conclusion p12 l10-13._

Thank you for suggesting more conclusions. Please find below our response.

….What is the message to the urban hydrologic modelers? How can this uncertainty be involved in the calibration process/result discussion? Can the results be validated with an urban hydrologic model?

_We believe the summary of our finding (p12/1-18) are all in interest to urban hydrologic modellers. In addition, results from this study can be used for uncertainty analyses of hydrologic and hydrodynamic modelling of similar sized urban catchments as it provides information on uncertainty associated with rainfall estimation. This estimate of uncertainty in combination of estimates of uncertainty due to model structure and model parameter will help to indicate the significance of rainfall uncertainty. This estimate of the relative importance of uncertainty sources can help to avoid false calibration and force fitting of model parameters (Vrugt et al., 2008). We already discussed this briefly in our manuscript (p12/29-35. it is a challenging task to validate these results using hydrological modelling as such validation also needs estimations of other sources of uncertainty (structural and parameter) as well as overall uncertainty in the model output. We will include this in the revised manuscript._

….What are results of other investigations concerning areal rainfall uncertainties?

_We think this is something that should be included in the Discussion rather than the Conclusion. We already discussed some other related studies (Ciach and Krajewski, 2006; Krajewski et al., 2003; Villarini et al., 2008) when comparing our results with those of other studies._

….Is the uncertainty greater/smaller than other uncertainties in urban hydrological modeling? Is it useful to take this uncertainty into account, if others are higher? Is it assumed, that the uncertainty increases with increasing area sizes in the lumped model?

_Individual uncertainties will be catchment specific, but It is still useful to take this uncertainty into account because only by quantifying it will you know if it is larger or smaller than others. We have not assumed that uncertainty in rainfall increases with increasing area size and since our scope does not cover this we cannot draw any conclusion on this issue._

…What is the recommendation for rain gauges number per square kilometer from this investigation? The measurement set-up is quite dense. Can general conclusions be drawn to less dense networks (and how)?

_We think such a hard and fast rule on number of data points cannot be derived for this methodology. Because, like any other geostatistical interpolation method, the efficiency of this method also heavily depends on reliable estimation of the geostatistical model (variogram). Hence it basically comes_

*down to the question of whether a given rain gauge network can produce a meaningful variogram? As we mentioned in the manuscript, Webster and Oliver (2007) suggested around 100 measurement points to calculate a geostatistical model. But there is no hard and fast rule to define minimum number of bins and the number of samples for each bin to produce a reliable variogram. Further, since pooling of repeated measurements would produce a multiplication of spatial lags, the length of the available data would also play a role in deciding the number of measurement locations.*

….How sensitive are the results, if the station density/combination of stations is changed in the investigation?

*Leaving one station out would definitely affect the results. First it will reduce the accuracy of the estimation of the variograms as the number of spatial lags per time instant would come down to 21 from 28. But the further effect of leaving one station out needs to be analysed in detail to see how it effect the uncertainty in the estimation of areal average rainfall intensity. In the manuscript we have not included such sensitivity analyses considering the direct relevance to the main scope of this study and the work load required to perform such an analyses.*

Suggestions:
Title: The title doesn't fit to the content of the manuscript. There are no urban hydrological models applied. Also, if no kriging is applied (not sure about that, see major comment p7 l14), it's not an geostatistical upscaling. The title "Estimation of uncertainties from spatial and temporal upscaling on an urban scale" is therefore misleading.

*Reply: Please refer to our response to comment p7 l14 for the explanation on spatial stochastic simulation, which is a geostatistical method. We believe it is quite clear from the methodology that this study uses geostatistical upscaling (e.g. the derivation of variograms, the use of spatial stochastic simulation). Further, we think the spatial extent (0 – 400 m) and the temporal averaging intervals (2 min -30 min) considered in this study are in interest of urban hydrology. Also we think the uncertainty estimation in areal rainfall would be more useful for the hydrology community working on uncertainty. These are the main reasons why the paper was oriented towards the urban hydrology community. Hence we don't think the tittle is misleading.*

Introduction:

p2 l19-25 There exist other methods for the estimation of uncertainties (bootstrapping,: : :), which should be mentioned in this context. Indeed, a focus should put on these methods, their comparisons and a reasonable decision for the applied method should be given at the end.

*Reply: Thank you for the suggestion. We would like to point out that we did not aim to discuss all uncertainty methods that are available for hydrological applications. Rather we wanted to keep the Introduction mainly focused on the methodology that we adapted for this study. Hence we followed the below order in our introduction*

- *Lumped hydrological models need spatial average rainfall over catchments*
- *Focus is on rainfall observations at points*
- *Thus point observations need to be scaled up*
- *Review of existing methods that do this*
- *Disadvantages of these methods*
- *Solution that does not have these disadvantages is to take geostatistical approach*
- *The main challenges with geostatistical approach and how it can be dealt*

*And we quoted the most relevant studies wherever necessary. The paper did not aim to compare uncertainty methods but to examine the levels of uncertainty in rainfall intensities.*

p2 l8-19 In the introduction a number of interpolation methods are mentioned, which are not used afterwards in the investigation. They could be left out.

*Reply: Please refer to our response to specific comment p2 l19-25*

Methodology:
p5 l11 How have the thresholds for the pooling been chosen?

*Reply: The maximum threshold value was limited to 10mm/hr to have enough time instants for the highest range (i.e. > 10 mm/hr) to produce stable variograms even at 30 min temporal averaging interval. We then decided to divide 0 – 10 mm/hr in to two equal subclasses (i.e. < 5mm/hr and 5-10 mm/hr). This gave us three subclasses which we thought a reasonable number given the length of the data, work load and computational demand.*

p6 l2 Methods and results (Fig. 6) are mixed.

*Reply: We accept that in Fig 6 part of a result is presented, but we believe that this combined figure helps to explain step 3 clearly and consequently makes it easier for the reader to understand. Further this is not a major result, but just an outcome of one of the steps.*

p6 l10 The method of NST could be explained briefly.

*Reply: Thank you for the suggestion. Section 3.3 already briefly explains NST with the basic theory and literature where detailed description of NST including the steps involved can be found. We will add some more explanation in the revised manuscript.*

p6 l17 Step 4 is not a step, only a description, and can be moved to step 5.

*Reply: Step 4 is one of the major steps of this study. It involves the construction of variograms. The reviewer is kindly requested to refer to Section 3.4 in the manuscript for further explanation.*

p7 l4 explanation for q is missing

*Reply: Thank you for pointing out this. q is the total number of measurement points. We will add this in the revised manuscript.*

p8 l9 With the standard deviation and the mean of areal rainfall intensities the restandardisation is carried out. For the former standardisation the standard deviation and the mean of point values have been used (since it is not clear, what has been pooled (see comment p5 l12), we assume time steps). Shouldn't be standard deviation and mean for standardization and re-standardisation be from the same type, so either from area or point values?

*Reply: Time steps, not events, were pooled to increase the spatial pairs. Hence mean and standard deviation from each time step is used for standardisation (ref Equation 1) as well as inverse standardisation.*

Results:

p8 l6-17 The nugget is interpreted as spatial rainfall variability or measurement error.
The network offers the great possibility to have rain gauge pairs with distances of 1 m.
Measurement errors have been excluded before by the paired measured time series.
So the spatial variability can be shown for these small distances, or why should this not
be possible?

Reply: *Although paired gauges are used for efficient quality control, it cannot avoid sampling related error of tipping buckets (Habib et al., 2001). For further explanation on why we associate nugget effect with sampling related error of tipping buckets, please refer to our response to specific comment p8/25.*

p8 l28 Since all errors have been excluded under the usage of the paired time series, the word TB error is somehow misleading. "Sampling error" could be more appropriate.

Reply: *Thank you for the suggestion. But this sampling error is only associated with tipping bucket type rain gauges. That is why we preferred to call it TB error. Further the same term was used in a previous study (Habib et al., 2001)on sampling errors of tipping bucket type rain gauge which was quoted in our discussion. Hence we wish to use the same term to be consistent with previous studies.*

p10 l3 Showing the CV would be more effective then showing the standard deviation in Fig. 8. An increasing of the standard deviation with an increasing intensity is trivial
(which is even stated on p11 l7-10). Also, a logarithmic plot would be useful.

Reply: *Thank you for the suggestion. But since the main reviewers did not think that this figure should be changed, we will wait for the editor's decide to decide if any change is required in this this figure.*

p10 l31-32 Design on peak rainfall intensity: The intensity AND the duration are important and both are used for the dimensioning of e.g. a sewer system.

Reply: *We agree with the reviewer that both peak and duration are important in the design of urban hydraulic structures. We will correct the sentence in the revised manuscript.*

p11 l1 Fig. 10 Maybe it would be useful to use violin plots instead of only the standard deviation to show the uncertainty.

Reply: *Thank you for the suggestion. But based on one of the assigned reviewer's comment we will replace this figure with Figure C2 below. This plot includes labels of CV values instead of error bars to make it easy to read.*

[Figure]

**Figure C2: Predictions of event peaks of AARI (indicated by points) together with labels indicating corresponding COV (%) values**

p11 l12 Fig. 11 The readability of the figure could be increased by colors and/or drawing only contours, not filling them. Showing the means as functions, not as fixed values, would give better conclusions. The high mean for 2 min, <10 mm/h is caused by only one extreme CV (_13 %) and is not representable.

*Reply: Thank you for the suggestion. We wanted to avoid the use of colours in the plots whenever possible. But we agree empty markers would improve the readability and we will use them in the revised manuscript.*

*Although we don't think peak prediction at event 8 (the extreme value) is an outlier, we accept the fact that the sample size is too small to derive a firm conclusion. This will be mentioned clearly in the revised manuscript. We also tested without event 8. and the trend remains the same, the average CV for the range < 10 mm/hr is reduced to 5.3 % from 6.6% at 2 min averaging interval when test 8 is excluded..*

*We also tried to relax the condition of minimum rainfall yield from 10 mm to 5 mm to derive more number of events. This gives us 19 such events. Similar analysis is carried out on these events and the results are presented below,*

[Figure]

**Figure C3: Coefficient of variation plotted against predicted peaks of AARI for the 19 events**

*It shows the similar behaviour to figure 11 where a higher uncertainty (characterised by CV) is seen for lower rainfall intensity (< 10 mm/h) at temporal averaging interval of 2 min where the TB error is at its highest. The trend is clearer this time as the highest three CV values at 2 min averaging interval belongs to lower intensity rainfall. The difference between average CV corresponds to lower (<10 mm/hr) and higher (>10 mm/hr) intensity rainfall is 7.2% and 4.0% respectively. We believe this is mainly due to the dominance of TB error corresponds to lower intensity rainfall at 2 min averaging interval.*

*But when the temporal averaging interval is 30 min where the TB error is minimal this effect is not there anymore. In fact the average CV is slightly higher (by 0.8%) for lower intensity rainfall than higher intensity rainfall. But the difference is too small to derive a conclusion for 30 min averaging interval.*

Conclusion:
p12 l10-13 Decreasing peaks due to aggregation in time is trivial and not a conclusion.

*Reply: We agree with the reviewer that decreasing peaks due to aggregation in time is trivial. But our conclusion gives a quantification of this reduction to show its significance. It helps to subsequently discuss the trade-off between temporal resolution and accuracy in rainfall prediction, which is the main aim of that bullet point.*

p12 l32-33 "This information can help to avoid false calibration and force fitting of model parameters" It remains unclear, how the result of the investigation can be used for the avoidance of the before mentioned issues.

*Reply: Results from this study can be used for uncertainty analyses of hydrologic and hydrodynamic modelling of similar sized urban catchments as it provides information on uncertainty associated with rainfall estimation. This estimate of uncertainty in combination of estimates of uncertainty due to model structure and model parameter will help to indicate the significance of rainfall uncertainty. This estimate of the relative importance of uncertainty sources can help to avoid false calibration and force fitting of model parameters (Vrugt et al., 2008).*

*We will include this in the revised manuscript.*

**Technical notes:**

p7 l5 "locationsxl" to "locations xl" and "locationsx0" to "locations x0"

*Reply: Thank you for pointing it out. It will be corrected in the revised manuscript.*

p8 l5 Eq. (6) "pi" and not "px", also the division by "m" is missing – Since this is a simple equation, it could left out, also Eq. (7)

*Reply: Thank you for pointing it out. It will be corrected in the revised manuscript. Although these equations are well-known in general, given the context of the application, we think it would help the reader to understand the step 8 clearly.*

p8 l6 Eq. (7) The term under the root has to be squared.
*Reply: Thank you for pointing it out. It will be corrected in the revised manuscript.*

p8 l25 "nugget-to-still ratio" to "nugget-to-sill ratio" (several times)
*Reply: Thank you for pointing it out. It will be corrected in the revised manuscript.*

p9 l3 "in their study found" to "found in their study"
*Reply: Thank you for pointing it out. It will be corrected in the revised manuscript.*

p9 l15 "(2003) in their" to "(2003) found in their"
*Reply: Thank you for pointing it out. It will be corrected in the revised manuscript.*

p10 l11-13 In Fig. 9 event 10 is shown, not event 11 (regarding to Fig. 10).
*Reply: Thank you for pointing it out this error. It will be corrected in the revised manuscript.*

p12 l22 "methods to a certain extent" – fuzzy phrase
*Reply: //The pooling procedure used in this study makes use of the continuous measurement of rainfall and helps provide a solution to meet the data requirements for geostatistical interpolation methods to a certain extent.//*
*By the term 'certain extent' we wanted say pooling can only partially solve the problem of scarcity in measurement points as it does not produce any new spatial lags, but only extends the information for existing lags. We agree it was not very clear and will reformulate the sentence in the revised manuscript.*

**Reference**

Ciach, G. J. and Krajewski, W. F.: Analysis and modeling of spatial correlation structure in small-scale rainfall in Central Oklahoma, Adv. Water Resour., 29, 1450–1463, doi:10.1016/j.advwatres.2005.11.003, 2006.

Fiener, P. and Auerswald, K.: Spatial variability of rainfall on a sub-kilometre scale, , 859(February), 848–859, doi:10.1002/esp, 2008.

Habib, E., Krajewski, W. F. and Kruger, A.: Sampling Errors of Tipping-Bucket Rain Gauge Measurements, J. Hydrol. Eng., 6(2), 159–166, doi:10.1061/(ASCE)1084-0699(2001)6:2(159), 2001.

Jaffrain, J. and Berne, A.: Quantification of the small-scale spatial structure of the raindrop size distribution from a network of disdrometers, J. Appl. Meteorol. Climatol., 51(5), 941–953, doi:10.1175/JAMC-D-11-0136.1, 2012.

Krajewski, W. F., Ciach, G. J. and Habib, E.: An analysis of small-scale rainfall variability in different climatic regimes, Hydrol. Sci. J., 48(2), 151–162, doi:10.1623/hysj.48.2.151.44694, 2003.

Villarini, G., Mandapaka, P. V., Krajewski, W. F. and Moore, R. J.: Rainfall and sampling uncertainties: A rain gauge perspective, J. Geophys. Res. Atmos., 113(11), 1–12, doi:10.1029/2007JD009214, 2008.

Webster, R. and Oliver, M. a: Geostatistics for environmental scientists, Second edi., John Wiley & Sons, Ltd, West Sussex, England., 2007.

---

## Author Response (AR2)

**Comments on manuscript- Version 2 (Revised)**

Anonymous referee #1

General comments
The intention of the paper has already been stated in the previous reviews of its first version. As it has not been changed, the focus of this review is on the improvements based on the former critics.
The structure of the paper has not been changed too much, but with the changes the paper focuses more on the methodology which supports the understanding. Furthermore, the single steps of the methodology are explained in more detail such that they could be reconstructed easily.
A major critic was the small sample size, which of course has not changed, but their impact is comprehensively analyzed. The results indicate that the influence seems to be negligible for the target of the paper. The differences between tipping bucket errors as well as spatial uncertainty is also differentiated, explained and their influences on the variograms are illustrated. Overall, the paper has been highly improved within the review process and could be published after the following comments have been considered.
Authors: *We thank the reviewer for the professional and thorough review which helped us to improve the manuscript.*

Specific comments
Remark: [Page: Line]
[8: 5]: "spatial aggregation" This term simply means a spatial average, does it?

*Author's response:*
Yes. Now it has been changed to 'spatial average' to be consistent throughout the manuscript.

Technical corrections
some comments on grammar, spelling or style. Although I did not focus on these aspects, I could find several errors and strange/complicated sentences. Here are a few examples:
[12: 28]: "The reason …" Long and complicated sentence
[14: 29]: "As a result …" A comma after "that" would highly help to understand the structure of the sentence
[15: 2] "…they reduced…" should be present tense
[15: 6] "… averaging interval are…" should be plural
[15:26] "… catchments, in similar climates…" I can't see a reason for the comma here
[15: 27] "This estimate of uncertainty…" three times the same construction in two sentences
Consequently, I would recommend proofreading from a native speaker.

*Author's response:*
Thank you for pointing out the above technical corrections. They all have been addressed now and the manuscript has been proofread again to avoid any grammar, spelling or style mistakes

**Anonymous referee #2**

Dear authors,
I have enjoyed reading this version of the paper, the text is much clearer and the method and the discussion related to the use of it are well explained and interesting.

I have some minor comments/suggestion for you to consider prior for the publication of the paper:

*Authors: We thank the reviewer for the professional and thorough review which helped us to improve the manuscript.*

Introduction section (and also in the conclusion)

1. "Despite recent advances in radar technologies rain gauge measurements are still considered to be the most accurate way of measuring rainfall, especially at short temporal averaging intervals (< 1 hour), which are of most interest in urban hydrology studies (Seo and Krajewski, 2010; Villarini et al., 2008). " The references are at the wrong location in the sentence. Should be:
"Despite recent advances in radar technologies rain gauge measurements are still considered to be the most accurate way of measuring rainfall, especially at short temporal averaging intervals (< 1 hour, e.g. Seo and Krajewski, 2010; Villarini et al., 2008), which are of most interest in urban hydrology studies (REF)". Where REF could be the paper by OchoaRodriguez et al. (2015) for example. Seo and Krajewsky and Villarini did not discuss urban applications in their papers.

*Author's response:*
*Thanks for pointing out this error. It has been corrected now*

Section 2.2
2. "Figure 3 shows time series of daily rainfall averaged over the network for 2012 and 2103." – should be 2013.

*Author's response:*
*Thanks for pointing out this error. It has been corrected now*

3. "For a catchment of this size (400 × 200 m2) it is very unlikely to have a response time of more than 30 min." – I would claim that even 30 min is too long for a response time for an urban catchment in this scale. Probably rainfall data of <10 min are required, especially for rainfall intensity larger than 10 mm/h. See for example: OchoaRodriguez et al. (2015), which you cite, and: Schilling, W. (1991). "Rainfall data for urban hydrology: what do we need?" Atmospheric Research, 27(13), 521. Zhu, H.j., and Schilling, W. (1996). "Simulation errors due to insufficient temporal rainfall resolution Annual combined sewer overflow." Atmospheric Research, 42(14),19.

*Author's response:*
*We agree with reviewer's comment that the resolution of rainfall data needed for this catchment would be less than 10 min as suggested by the references mentioned by the reviewer. But when it comes to response time of this catchment (time of concentration) which is what we wanted to discuss here, we think 30 min is reasonable and safe upper limit as there is no hard and fast rule for catchment response time against the area.*

Section 3.5
4. "is known as block kriging" – please add a reference to a paper or a textbook explaining what block kriging is.

*Author's response:*
*Thank you for the suggestion. The following reference has been added.*
*(Isaaks and Srivastava, 1989)*

Section 3.6
5. back transformation and linear extrapolation. Let assume that at a given time t the highest rain intensity was 10 mm/h at a given gauge and that the highest rain intensity ever observed was 20 mm/h (somewhere in the domain). Can it be that the highest stochastically simulated value at time t will exceed 20 mm/h? In other words – using your method what will be the highest rainfall intensity that can be simulated? Is it limited by the observed data? Is there any other physical limitation?

*Author's response:*
*Stochastic simulation was carried out for each and every event separately. So if a stochastically simulated value at a given time instant is higher than the highest observed value or smaller than the smallest observed value during that particular event, then linear extrapolation was used. These linear models were derived using a selected number of head and tail portion of normal Q-Q plot of observed rainfall intensities of that particular event. But in all cases, only a few stochastically simulated values were outside the range and the difference were very small as these values were basically derived from observed data. Hence regardless of how big the observed intensity during an event, few stochastically simulated values during that event can be marginally outside the range and they need to be extrapolated.*

Section 4.1
6. I strongly encourage the authors to have a look at the paper by Peleg at el. (2013) which is very relevant to your discussion. They are discussing in details the changes of rainfall spatial correlation with aggregation time and the large errors of weather radar for the <5 min time scales (relevant to comment 1 above).
Peleg, N., BenAsher, M., and Morin, E. (2013): Radar subpixelscale rainfall variability and uncertainty: lessons learned from observations of a dense raingauge network, Hydrol. Earth Syst. Sci., 17, 21952208.

*Author's response:*
*Thank you for the suggestion. We agree that some of the finding of the above paper is relevant to the current study especially the change of spatial correlation against temporal averaging time. This paper has been added to the list of paper mentioned where spatial correlation against temporal averaging time is discussed.*

General
7. It would be nice to have some examples of the stochastic fields that you are generating. For example, showing one time step with the gauges marked (and labels indicating the observed rainfall intensity) and the simulated rainfall intensity around the field.

*Author's response:*
*Thank you for the suggestion. But since stochastic fields are generated for the standardised rainfall intensity, such a figure can only be produced for standardised rainfall intensity and it will not be a representation of actual rainfall intensity (observed and simulated). Hence it might not be very useful.*

[revised manuscript text omitted]

**Anonymous referee #1**

**General comments**
The goal of the paper is the analysis of the uncertainty of the areal interpolation of precipitation data from a dense gauge network with a high temporal resolution. The topic itself is relevant and currently discussed, in particular, in the context of radar measurements.
From this point of view it is very interesting to analyse such a very dense gauge network with respect to the influence of the spatial variability/measurement errors on the areal interpolation. The paper is well structured and the methods used are (in most cases) clearly presented using a step wise explanation. However, a major problem of the study is the sample size of only 13 events (of 2 years). As these events are highly variable as shown in table 1, the uncertainty of the entire study is rather high. Although there is a pooling method presented that tries to overcome this issue, the underlying assumption might still be problematic (see comment [5: 5]). Furthermore, many descriptive results are presented, but their consequences are rarely discussed. There are also a lot of trivial results in the paper. It should become much clearer, what is a logical consequence of the precipitation structure and what is an actually new result of this study. For example, temporal averaging will always reduce precipitation peaks. Finally, the language could be more precise because several sentences are too fuzzy.

*Authors: We thank the reviewer for the professional and thorough review of this paper. We have attempted to address all the comments and our response is listed below.*

….However, a major problem of the study is the sample size of only 13 events (of 2 years). As these events are highly variable as shown in table 1, the uncertainty of the entire study is rather high.

*Reply: The major technical comment from the reviewer is that our analyses are based on a small sample size of only 13 events. We would like to clarify that the entire ten months of rainfall data from 8 locations were used for the development and calibration of the geostatistical model. We apologise if it was not clear in the manuscript that the entire 10 months of data was used and we have included additional text in the revised manuscript to clarify this. (Revised manuscript [4: 27-29] and [5:9])*

*Further a discussion on the size of the data used in this study has been added to the revised manuscript (Revised manuscript [10:30 – 11:11])*

Although there is a pooling method presented that tries to overcome this issue, the underlying assumption might still be problematic (see comment [5: 5]).

*Reply: The underlying assumption of pooling is discussed in detail in our reply to specific comment 5: 5.*

Furthermore, many descriptive results are presented, but their consequences are rarely discussed. There are also a lot of trivial results in the paper. It should become much clearer, what is a logical consequence of the precipitation structure and what is an actually new result of this study. For example, temporal averaging will always reduce precipitation peaks. Finally, the language could be more precise because several sentences are too fuzzy.

*Reply: We have addressed the concerns stated by the referee. Superfluous sentences have been removed and some rewritten to reduce the attention paid to trivial results improve clarity and discuss the consequence of the analysis carried out in the paper. Details of these modifications are described below.*

*Trivial results removed, or reduced:*

*Several sentences have been removed and rewritten to try and reduce the level of trivial results, including the ones mentioned by the reviewer. For more details please refer to specific comments [5: 11], [5: 12-13] and [10: 16].*

*Improvement of clarity and language:*
*The following parts have been modified to give a clearer explanation, as indicated in specific comments:*
*Quality control of rainfall data: See our reply to comment [3:32]*
*Event separation: See our reply to comment [4: 14]*
*Stochastic simulation: See our reply to comment [6:29-7:22]*
*Interpretation of the variograms: See our reply to comment [8: 16]*

*New results/novelty:*
*Because of the nature of this study more emphasis is given to the methodology. We believe that this methodology is novel for the following reasons:*

- *In literature, geostatistics has been used to analyse the spatial correlation structure of rainfall at various spatial scales, but its application to estimate the level of uncertainty in rainfall upscaling has not been fully explored, mainly due to its inherent complexity and demanding data requirements. In this study we addressed these challenges which include the use of repetitive rainfall measurements (pooling) to increase the number of observations used for variogram estimation.*

- *We used spatial stochastic simulation to address the combination of change of support (from point to catchment) and non-normality for prediction of rainfall and associated uncertainty. To the best of our knowledge this has not been done previously.*

- *We defined intensity classes and derived different geostatistical models (variograms) for each class. On top of that we also used different temporal averaging intervals. To the best of our knowledge no previous study attempted to assign geostatistical models for a combination of intensity class and temporal averaging interval.*

*In addition we presented and discussed the following new results:*

- *Spatial correlation structure of rainfall at a spatial extent of 200m × 400m has been presented. To the best of our knowledge no previous studies have analysed spatial correlation structure of rainfall at such a small spatial extent.*

- *The current study shows that for this spatial scale the use of a single geostatistical model based on a single variogram is not appropriate. Instead, different variograms for different rainfall intensity classes should be used. This is a key finding of the study.*

- *In addition to visual interpretation we also presented quantification of uncertainty in upscaled rainfall predictions in terms of CV for peaks of rainfall intensity. Previous studies (e.g. Villarini et al. 2008) have provided such quantification of uncertainty only using simpler error measures (normalized root mean squared error and normalized mean absolute error) while we use geostatistical approaches that take the effect of spatial correlation into account.*

**Specific comments**
**[Page: Line]**

[2: 19] "Since rainfall can vary over space significantly, any method for scaling up the point rainfall measurements adds uncertainty on top of existing measurement error."

The measurement error is not explained so far. It should be introduced as it is a major aspect in the rest of the paper.

*Reply: A short introduction on measurement error mainly focusing on tipping bucket errors has been added to our revised manuscript (Revised manuscript [2:22-26]).*

[Fig 5] For a better comparison, the classes of the histogram should be the same as the classes used later for the variogram.

*Reply: The rainfall intensity class intervals were the same for histograms (Fig. 5) and variograms (Fig. 7).*

[2: 32] ": : :not always: : :" Rainfall intensity values are almost never normally distributed

*Reply: Thank you for pointing this out. This has been corrected in the revised manuscript.*

[3: 32] "During the dynamic calibration: : :" How did you identify the volume error per tip using a dynamic calibration? The entire section is not clear: "Every long set of data: : :if the differences: : :" What is the long set and what kind of differences?

*Reply: This section has been modified in the revised manuscript to give more precise information on how the quality control was carried out using a paired setup (Revised manuscript [4: 8-18]).*

[3: 8] ": : : to obtain a more normally : : :" After the transformation they are perfectly normally distributed.

*Reply: Thank you for pointing out this out. This has been corrected in the revised manuscript*

[4: 14]: How are the events defined? What are the criteria of the end of the event; if all stations show zero precipitation? Is there a minimum separation time of two events?
Or will a few minutes without rainfall already separate the events? Why does the event need to be at least 20 min?

*Reply: There are two conditions for defining an event: the yield should be more than 10 mm and the event duration should be larger than 20 minutes. If all stations show zero precipitation based on 5 min averaging interval it is considered as the end of an event and there is no minimum separation time between events. Fig. 3 and Table 1 show that except for events 6 and 7 there are no events close to each other.*

*We chose a 20 min window in order to get a sufficient number of data values when temporal averaging intervals of more than 2 min are used in the analysis. For example, a 10 min event gives two data values when using a 5 min averaging interval and gives only 1 data value for 15 min and 30 min averaging intervals. Hence in order to have at least two data values for all temporal averaging intervals examined, a minimum event duration of 20 minutes is needed. Table 1 shows that the lowest event duration in the collected data was 1.5 hours. Hence all events had at least 45 data values for a 2 minute averaging interval and at least 3 data values for a 30 minutes averaging interval.*

*Also note that, as we mentioned already, the event separation (ref Table 1) is used only for the analyses presented in Sections 4.2.3 and 4.2.4. Hence these criteria don't leave out any data in the development and calibration of the geostatistical models. We have included additional text in the revised manuscript to clarify this. (Revised manuscript [4: 27-29] and [5:9])*

[5: 5] "The underlying assumption: : :" This is your major assumption for the pooling based on the trade-off between independence and number of available time instants for the three defined classes.

However, you did not show any analysis to validate the independence assumption. Instead, literature is cited, which shows the dependence of the spatial correlation on the intensity. How can you be sure that the influence is small enough to be disregarded, and that the pooling procedure does not mess up your further analysis?

*Reply:* We agree with the concerns expressed by the reviewer. This is the reason why we introduced step 2 to treat each and every time instant within a subset, by individually calculating their mean and standard deviation. Although variograms are derived only for the whole subset, step 2 (before geostatistical upscaling) and step 9 (after geostatistical upscaling) ensure that the probabilistic model is adjusted for each time instant separately, based on the mean and standard deviation for that particular time instant. Effectively, we assume the same correlogram for time instants of the same subclass, not the same variogram. Although this does not justify the assumption of similar spatial correlation structure with in the pooled classes, it at least relaxes the assumption of the same variogram within subclasses.

*The similarity between Fig. C1 and Fig. 7 also shows that the characteristics of the data from subsets are consistent with those of the entire pooled class. Moreover, several studies (e.g. Dirks et al., 1998; Ly et al., 2011; Tao et al., 2009) used only a single geostatistical model in the form of single variograms/correlograms for the entire range of rainfall intensity. The current study shows that for small time and space scales the use of a single geostatistical model based on a single variogram is not appropriate. This is a key finding of this study. We agree that with narrower intervals the assumption of consistency in spatial variability would be more realistic. But with the available data we had to find a compromise with the number of time instants. We believe that using three intensity subclasses is a workable compromise. Based on Fig. C1 (in our reply to comment 8:16), where variograms are produced for narrower subclasses, we conclude that the variograms shown in Fig. 7 are good representations of the average spatial variability conditions for corresponding intensity classes.*

*We discussed this issue in the revised manuscript (Revised manuscript [11:12 − 26])*

[5: 11] "As expected, with increasing temporal averaging: : :" This is not only expected, this is obvious for an aggregation process of rainfall.

*Reply: We agree with this comment and we revised this sentence in the revised manuscript (Revised manuscript [5:30 -6:3]).*

[5: 12-13] This is also not surprising due to the skewed intensity distribution of precipitation. It should become clearer in this part [5:11-13], that the results are just natural characteristics of precipitation.

*Reply: We agree with this comment and we revised this sentence in the revised manuscript (Revised manuscript [5:30 -6:3])*

[5: 14] ": : :only eight: : :" Where does this number come from? In Fig. 5 these are way more than eight for the 30 min average.

*Reply: Thank you for pointing out this. We apologise for the mismatch between text and figure. This histogram shows the number of time instants (t) × number of stations (eight). It has now been corrected in the revised manuscript to show the number of time instants (t) to be consistent with the text. Actually there are seven (not eight) time instants where intensity exceeds 10 mm/hr at 30 min temporal averaging interval. This figure has now been revised (Revised manuscript: Fig. 5).*

[6: 26] "It is negligible small: : :" Isn't that contradicting to the later parts where the uncertainty of the tipping bucket error is analysed (for example in [12:15]).

*Reply: We wanted to argue that any physical micro-scale spatial variation between rain gauges, which theoretically may be one of the causes of the nugget effect, is negligibly small for this case. We did not mean to claim that the nugget itself is negligibly small. We accept that the particular sentence was poorly constructed. We have now modified the text (Revised manuscript [7:10-11])*

[6:29-7:22] In the entire chapter it becomes not clear what kind of stochastic simulation was performed (conditional/unconditional) and which method was used to obtain the 500 simulation results.

*Reply: We accept that this section requires more detail. We have modified this section in the revised manuscript (Revised manuscript Section 3.5).*

[7: 10] Kriging would be possible, if the back-transformation of the individual points was performed before the averaging. (No block kriging, but ordinary kriging of single points (25 x 25m grid)
*Reply: This might work for the prediction but not for the prediction uncertainty of spatially aggregated rainfall, which we also needed to quantify in this study. That is why we turned to spatial stochastic simulation. We agree it was not clear from the text. Now this section has been revised (Revised manuscript [7:22-8:6])*

[8: 1]": : :is spatial aggregation of each and every simulation: : :" Should it be of "each time step"?

*Reply: It should be 'spatial aggregation of each and every simulation (realisation)' (i.e. 500 per time instant). This yields 500 values of the areal mean per time instant. We have modified Section 3.5 to give a clearer explanation of spatial stochastic simulation (refer comment [6:29-7:22]).*

[8: 16] How can you be sure that this effect is caused by measurement errors? Couldn't the nugget effect be also caused by the pooling technique, that is, by mixing different time steps; or by a high variability in the natural precipitation process? (see Remark [9: 7-21])

*Reply: The two other possible reasons for the nugget effect as mentioned by the reviewer are discussed in detail below.*

1. *Pooling of time instants:*
   *We do not think pooling is the reason for the nugget effect. Note that we did not pair rainfall measurements from one time instant with rainfall measurements from another time instant. We only pooled sample variograms from different time instants. But that cannot cause an additional nugget effect; it just gives back the average nugget of those for all time instants. If none of the variograms from different time instants have a large nugget, then the pooled one also will not have a large nugget.*

[Figure]

**Fig. C1: Calculated variograms for 5 min averaging interval and for a narrower range of intensity.**

*Furthermore, Fig. C1 shows the behaviour of variogram models for narrower intensity classes ranging from 0 to 14 mm/hr for the 5 min averaging interval. The highest intensity class is limited to ≥12 - <14 mm/hr as for further narrower ranges (i.e ≥14 - <16 mm/hr and so on) there are not enough sample points to produce a meaningful variogram. The variograms for the narrower classes show that the assumption of similar spatial variability within a pooled subset is stronger. Hence the effect of variation caused by pooling is reduced. Comparing this figure with Fig. 7 not only shows the behaviour of the nugget effect against the intensity as seen from Fig. 7, it also makes clear that there is a clear decreasing trend in the nugget as intensity increases. This also indicates that pooling is not responsible for the pattern of the nugget effect.*

*We have added the Fig. C1 in the revised manuscript and discussed the effect of pooling (Revised manuscript [11:12-26)*

2. *High variability in the natural precipitation process:*
   *Please refer to our response to the comment [9: 7-21]*

[9: 2] Is there a reason why the content of Habib et al. is mentioned explicitly and not of Villarini et al?

*Reply: The main aim of Habib et al. (2001) was to investigate the sampling error of tipping bucket measurements, whereas Villarini et al. (2008) was a later study and derived a similar conclusion as a part of their findings. Among the two only Habib et al. (2001) discuss the sampling errors of tipping bucket rain gauges extensively. Hence their work is explicitly mentioned.*

[9: 7-21] The important point of the interpretation of the variograms is not clearly stated. The variograms actually show, that for short time periods < 5 min (except for high intensities), there is almost no spatial correlation, that is the field is just random. If the nugget (explained here as tipping bucket error) is almost as high as the sill, there are two options. First, there is just no spatial correlation at the regarded distance, or the spatial correlation of the field cannot be detected by the tipping buckets because of the measurement error. There is also a very weak correlation (even for high aggregations) for the intensities smaller 5.0 mm/h. How can you be sure, that the nugget comes

from the tipping bucket error, and does not represent a very high spatial variability of the natural precipitation at very short distances?

*Reply: We accept that the nugget effect could be due to a combination of micro-scale spatial variability and measurement error. We modified this section to address this and to interpret the variograms better (Revised manuscript section 4.1).*

[10: 16] "Here it can be noted: : :" As the precipitation intensities are never uniformly distributed, this effect is a logic consequence.
*Reply: Thank you pointing this out. We agree that this is a logical consequence, but this result leads to the discussion of trade-off between timescale temporal resolution and accuracy in rainfall prediction. Hence we decided to keep this, but we have now reformulated this discussion in the revised manuscript to emphasise on the fact that this is a logical consequence of temporal aggregation (Revised manuscript [13:2-6]).*

[10: 30] This chapter should be rewritten, could be shortened and included in the next one. As explained in the last sentence of the paragraph, it is difficult to compare the standard deviation of different absolute values. Fig. 10 is rather useless, as the intervals (uncertainty) cannot be read. A table including the standard deviations in addition with the CVs for the single events could help.

*Reply: Sections 4.2.3 and 4.2.4 have been combined into one section in the revised manuscript as suggested by the reviewer. We agree that the standard deviation in Fig. 10 is too small to be read. Fig. 10 has been modified to include CV instead of standard deviation.( Revised manuscript Fig. 11, Section 4.2.3)*

*Please refer to our response to comment 11: 12-26 regarding the sample size.*

[11: 12-26] What is the actual goal of that chapter? Has there any kind of significance testing be performed? As there is one very large CV value <10 mm (2 min) out of six, the comparison between the means might be biased. If this value was an outlier, would the result still be that considerable? There seem to be a tendency, but the sample size is very small and thus, there is a lot of uncertainty in this result.

*Reply: We removed Fig. 11 from the manuscript and discussions are now based on revised Fig. 10 (please refer to our response to comment 10: 30).*

*We accept the fact that the sample size is too small to derive a firm conclusion. To comment on prediction uncertainty against intensity with a larger sample size, we modified the Fig. 8 (Revised manuscript Fig. 9) to show CV instead of standard deviation. Now it can be seen from modified Fig. 8 that there is a clear trend of increasing CV with increasing AARI. We also modified the discussions in the sections 4.2.1 and 4.2.3 accordingly.*

*Further although we do not think peak prediction at event 8 is an outlier, we also performed the test without event 8, which results in the same trend. But the average CV for the range < 10 mm/hr is reduced to 5.3 % from 6.6% at 2 min averaging interval.*

[11: 22] "This is fairly high: : :" How did you judge that? 25% to -15 % of the peak runoff is your reference. But what is the expected influence of the uncertainty of the rainfall when estimating the runoff from the precipitation? This is an important question for the judgement as one should know the necessary accuracy of the input precipitation and not the total uncertainty of the runoff.

*Reply: A 13% uncertainty in rainfall will result in a similar level of uncertainty in runoff prediction for a completely impervious surface according to the well-established rational formula (Viessman and Lewis, 1995) which is still widely used for estimating design discharge in small urban catchments. .*

*This uncertainty is high given the allowed limit of -15% to 25%, also not to forget other sources of uncertainty due to parameter and model structure. Furthermore, in several recent studies (Gires et al., 2012; Schellart et al., 2012) the effect of this small scale (< 1km) variability of rainfall on urban runoff peak has been proven to be significant.*

*We have added some more text in the revised manuscript to clarify this (Revised manuscript [13:32-14:2])*

[11: 26] "Hence a better trade-off : : :" What does this actually mean? How could this be achieved? There is a link missing between the shown problems and the actual application. Later, in the conclusion, it becomes clearer, but in this chapter this sentence
is kind of fuzzy.

*Reply: We accept that this sentence is not clear. This has now been reformulated in the revised manuscript (Revised manuscript 13:2-6)*

[12: 6-10] It should be mentioned, that the result of "peak" intensities are explained here.

*Reply: Thank you for pointing this out. We have done so in the revised manuscript.*

[12: 24] ": : : radar measurements : : : would be much higher: : :" Please, give some references here.

*Reply: The following references have been added:*
*(Seo and Krajewski, 2010; Villarini et al., 2008).*

[12: 27] You did not show explicitly the advantage of the paired rain gauges. Unless the improvements are shown in the main chapters, they should not be part of the conclusions.

*Reply: Thank you for this comment. This has been removed from the conclusion in the revised manuscript.*

**Technical corrections**
[2: 7] : : :where time series of areal : : : "are" needed.
[2: 27] : : : [8,9] : : : citations missing
[4: 9] "That" is because: : :
[8:5/8] Index Error: px should be pi
[9: 21] : : : "look" similar: : :

*Reply: Thank you for pointing out these errors. All have been corrected in the revised manuscript.*

**Anonymous referee #2**

**General**
This paper suggests a method to upscale rainfall intensity from point scale to catchment scale. The authors suggest a Kriging-based stochastic method for this upscaling; a method that allows an uncertainty estimation of the areal rainfall. I found the method suggested by the authors very interesting. I think that it will be of interest mainly for the hydro-meteorologist community (dealing with weather radar estimations) rather than for the urban hydrologist community. Main problem in the paper is the short period of observation (two seasons) that is expressed in a low confidence in the presented results.
I found that some key papers dealing with dense rain-gauge networks and rainfall variability in the past were not mentioned and that some trivial aspects discussed in the past are repeated in here. I would suggest the authors to thoroughly revise the paper as follow: make the upscaling method as the main focus of the paper, explain it with much further details and with a much clearer language. Use the data you have from the dense rain-gauge network as a case study to demonstrate how you can upscale rainfall for the catchment / weather radar scale and show the advantages of estimating uncertainties with the method you are suggesting. Please find below my specific comments, following by some general comments.

*Authors: We thank the reviewer for the professional and thorough review of our paper. We have attempted to address all comments listed below.*

…Main problem in the paper is the short period of observation (two seasons) that is expressed in a low confidence in the presented results…

*Reply: Please refer to our detailed response to the reviewer's specific comment on data used in the study [3:19-20]*

…I found that some key papers dealing with dense rain-gauge networks and rainfall variability in the past were not mentioned

*Reply: Please refer to our detailed response to reviewer's specific comment on literature used in the study [2:19-21]*

….and that some trivial aspects discussed in the past are repeated in here.

*Reply: We attempted to deal with the concerns stated by the reviewer. Several sentences have been removed and some rewritten to try and reduce the level of trivial results. They are listed below.*

*[5: 11-13] "As expected, with increasing temporal averaging the number of time instants t reduces. Fig. 5 also shows that the higher the intensity, the smaller the t. There is a large difference between t for lower and higher intensity ranges which shows the dominance of lower intensity (0.1-5.0 mm/h) rainfall over the recording periods" has been reformulated to emphasise on the fact that these observations are either natural or logical consequence. (Revised manuscript [5:30 − 6:1])*

*[10: 16] "It is expected that with increasing temporal averaging interval the local minima and maxima of AARI get smoothed out. Here it can be noted that in this event this effect decreases the event peak AARI from around 50 mm/h to around 20 mm/h as the temporal averaging interval increases from 2 min to 30 min." has been reformulated to emphasise on the fact that this observations is a logical consequence of the aggregation process. (Revised manuscript [13:2-6])*

… I would suggest the authors to thoroughly revise the paper as follow: make the upscaling method as the main focus of the paper, explain it with much further details and with a much clearer language. Use the data you have from the dense rain-gauge network as a case study to demonstrate how you can upscale rainfall for the catchment / weather radar scale and show the advantages of estimating uncertainties with the method you are suggesting.

*Reply:* Thank you for the suggestion. With all due respect to the reviewer's suggestion, we think that the manuscript is already heavily focused on methodology with a dedicated section which covers around 35% of the manuscript (pagewise). Further, to enable the reader to follow the methodology more easily, we explained it with a step by step procedure for a general case of estimating uncertainty in upscaling of point rainfall data. We tried to keep the methodology as general as possible while also providing enough detail on how each step is applied in the case of the Bradford case study. Furthermore, we believe that introducing the data before the methodology enables the reader to follow the methodology more easily as some of the steps require a pre-introduction to the data to explain why such step is needed. Hence we feel that the structure of the manuscript follows a logical work flow.

*Nevertheless we accept that some part of the manuscript needs more explanation, especially the spatial stochastic simulation methodology. We also agree that the language could be clearer throughout the manuscript. Hence we modified several sections/ parts of sections including the following based on reviewer's specific comments:*

*Quality control of rainfall data using paired gauge set up: please refer to our response to reviewer's specific comment [4:1-5]*

*Spatial stochastic simulation: please refer to our response to reviewer's specific comment [7:13-22]*

**Specific comments**

**[Page:Lines]**

[2:4-6] Please support this statement with a reference.

*Reply:* The following references have been added in the revised manuscript:
*(Seo and Krajewski, 2010; Villarini et al., 2008)*

[2:19-21] Since you are dealing with rainfall uncertainty for small domains, taking into consideration the rainfall spatial and temporal variability, I am strongly recommend you to check also the following papers that were published in the recent years, which I think you will find them all relevant to your study:

Peleg N, Ben-Asher M, Morin E. Radar subpixel-scale rainfall variability and uncertainty: lessons learned from observations of a dense rain-gauge network. Hydrol. Earth Syst. Sci. 2013 Jun 14;17(6):2195-208.

Krajewski, W. F., Kruger, A., and Nespor, V.: Experimental and numerical studies of small-scale rainfall measurements and variability, Water Sci. Technol., 37, 131–138, doi: 10.1016/s0273-1223(98)00325-4, 1998.

Pedersen, L., Jensen, N. E., Christensen, L. E., and Madsen, H.: Quantification of the spatial variability of rainfall based on a dense network of rain gauges, Atmos. Res., 95, 441–454, doi:10.1016/j.atmosres.2009.11.007, 2010.

Fiener, P. and Auerswald, K.: Spatial variability of rainfall on a sub-kilometre scale, Earth Surf. roc. Land., 34, 848–859, doi:10.1002/esp.1779, 2009.

Seo, B. C. and Krajewski, W. F.: Investigation of the scale dependent variability of radar-rainfall and rain gauge error covariance, Adv. Water Resour., 34, 152–163, doi:10.1016/j.advwatres.2010.10.006, 2011.

Gebremichael, M. and Krajewski, W. F.: Assessment of the statistical characterization of small-scale rainfall variability from radar: Analysis of TRMM ground validation datasets, J. Appl. Meteorol., 43, 1180–1199, doi:10.1175/1520-0450(2004)043<1180:aotsco>2.0.co;2, 2004.

Ciach, G. J. and Krajewski, W. F.: On the estimation of radar rainfall error variance, Adv. Water Resour., 22, 585–595, doi:10.1016/s0309-1708(98)00043-8, 1999.

*Reply: Thank you for suggesting the above papers. We wanted to keep our introduction mainly focused on the methodology that we adapted for this study. Hence we followed the below order in our introduction*

- *Lumped hydrological models need spatial average rainfall over catchments*
- *Focus is on a case where the input data are rainfall observations at points*
- *Thus point observations need to be scaled up*
- *Review of existing methods that can do this*
- *Disadvantages of these methods*
- *Solution that does not have these disadvantages is to take a geostatistical approach*
- *The main challenges with geostatistical approach and how these can be dealt with*

*We quoted the most relevant studies wherever necessary. We accept that there are many other studies like the ones mentioned by the reviewer which can be relevant because of the similar spatial extent of the rainfall data. But most of these publications are based or partly based on radar data (areal rainfall data) and outside the main scope of our study, which is upscaling of point rainfall data and estimating associated uncertainty. Discussion of the literature where both radar data and point rainfall are used together to compare and/or analyse spatial correlation is slightly out of the context and might lead to a very lengthy Introduction and might also confuse the readers even if the spatial extent of interest is the same. Therefore to keep the Introduction to the point and concise and focused on our objectives (upscaling of point rainfall data and associated uncertainty) we have not included literature which are completely/partly based on radar data.*

*However, from the reviewer's suggestion we found the following papers directly relevant to our study. We thank the reviewer for suggesting these papers. We have discussed these papers in appropriate sections of the revised manuscript. We summarise how these papers are related to our study below.*

*Pedersen, L., Jensen, N. E., Christensen, L. E., and Madsen, H.: Quantification of the spatial variability of rainfall based on a dense network of rain gauges, Atmos. Res., 95, 441–454, doi:10.1016/j.atmosres.2009.11.007, 2010.*

*The aim of this paper is to quantify the uncertainties of using a single rain gauge to represent the rainfall over a 500 × 500 m area. A field experiment placing nine 0.2 mm tipping bucket type rain gauges within an area of 500 × 500 m, each representing one-ninth of the area, was used to address the issue. The variability of rainfall is studied and uncertainty in areal rainfall is estimated for different time scales. Although this study uses a simpler approach to estimate the uncertainty, results are still comparable to our study.*
(Revised manuscript [12:7-11])

*Fiener, P. and Auerswald, K.: Spatial variability of rainfall on a sub-kilometre scale, Earth Surf. roc. Land., 34, 848–859, doi:10.1002/esp.1779, 2009*

*In this study, a network of 13 tipping-bucket rain gauges was operated on a 1·4 km2 test site in southern Germany for four years to quantify spatial trends in rainfall depth, intensity, erosivity, and predicted runoff. Their data is comparable to ours as they also used summer half-year data for their analyses. Although they did not calculate any uncertainty in areal rainfall estimation, their analyses on spatial trend against temporal averaging interval could be of interest to our study. One of their conclusions suggests that in the longer term there is no difference in rainfall depth within the test site, but in short-time periods or for single events the assumption of spatially uniform rainfall is invalid on the sub-kilometre scale, This complements one of the findings from our study.*
(Revised manuscript [10:4-6], [10:34-11:1])

[2:27] Reference typo.

*Reply: The following references have been added in the revised manuscript.*
*(Ly et al., 2013; Mair and Fares, 2011)*

[2:32] I would even claim that it is rare to find locations where the rainfall is normally distributed.

*Reply: Thank you for pointing this error out. This has been corrected in the revised manuscript.*

[3:3] word "often" can be deleted.

*Reply: Thank you for pointing this out. This has been corrected in the revised manuscript.*

[3:19-20] This is a very short period of observation, and winter rainfall is not represented at all. How does it affect your results? Moreover, in [4:7-12] you mention the large difference between the two years of observation. This imply that the climatology was different between the two years and therefore I would expect that it will somehow influence on the rainfall spatial correlation. With only two years, the variability expected for the spatial rainfall structure cannot be represented and this should be discussed.

*Reply: We acknowledge that the data cover only 10 months, i.e. two summer periods in 2012 and 2013, but however our geo statistical models (based on which further results are produced) are stable. This has now been discussed in detail in our revised manuscript. (Revised manuscript [10: 30-11:11])*

[Fig. 1] The recommendation is to mount rain-gauges elevated at 1.2 m above ground, where here the gauges seem to be placed directly on ground level (roof top). I wonder how this affects rainfall intensity estimations.

*Reply: We understand that different guidelines suggest different elevations when it comes to height of a rain gauge from the surrounding ground level. In our case we followed the standard UK practice (http://www.metoffice.gov.uk/guide/weather/observations-guide/how-we-measure-rainfall) which suggests the rim of the tipping bucket to be around 0.5 m above the surrounding ground level. This clarification has been added to the manuscript (Revised manuscript [3:30-33])*

[3:29] what about time drift? Did you reset the loggers every 4-5 weeks to avoid this problem?

*Reply: Thank you for pointing this out. The data loggers were reset every 4-5 weeks during data collection to avoid any significant time drift. We have included this information in the revised manuscript (Revised manuscript [4:5-6])*

[4:1-5] This is not clear to me. If I got it right, you are comparing paired gauges for each rain event by accumulating the rainfall over the gauges and comparing them and if the difference exceed the 4%

Peleg, N., Marra, F., Fatichi, S., Paschalis, A., Molnar, P. and Burlando, P., 2016. Spatial variability of extreme rainfall at radar subpixel scale.
Journal of Hydrology.

*Reply: This section has been modified in the revised manuscript to give more precise information on how the quality control was carried out using paired setup (Revised manuscript [4:8-18]).*

 [5:10] three rainfall intensity classes were SUBJECTIVELY selected? What was the criterion?

*Reply: The maximum threshold value was limited to 10mm/hr to have enough time instants for the highest range (i.e. > 10 mm/hr) in order to produce stable variograms even at 30 min temporal averaging interval. It was then decided to divide the 0 – 10 mm/hr class to two equal subclasses (i.e. < 5mm/hr and 5-10 mm/hr). This resulted in three subclasses, which is a reasonable number given the size of the data set and work load and computational demand.*

*We added the above clarification in the revised manuscript (Revised manuscript [5:25-29])*

*Furthermore, we performed the following test to see if these three classes represent the entire intensity range. We produced variogram models for narrower intensity classes ranging from 0 to 14 mm/hr for the 5 min averaging interval. The highest intensity class is limited to ≥12 - <14 mm/hr as for further narrower ranges (i.e ≥14 - <16 mm/hr and so on) there are not enough sample points to produce a meaningful variogram. Looking at these variogram at Fig. C1, we believe the variograms in Fig. 7 are good representations of the average conditions for corresponding intensity classes.*

[Figure]

**Fig. C1: Calculated variograms for 5 min averaging interval and for a narrower range of intensity**

*We have added the Fig. C1 in the revised manuscript and included the above discussion (Fig. 9, Revised manuscript [11:12-26])*

[6:8] n.d.?

*Reply: Thank you for pointing out this. It should be Van der Waerden (1953). We have corrected this in the revised manuscript*

[6:26] "It is negligibly small in the case of rainfall intensity data". Not necessarily, Peleg et al. (2013) reported a 0.92 nugget for 1-min time resolution. I am not sure that this can be neglected.

*Reply: We wanted to argue that any physical micro-scale spatial variation between rain gauges, which theoretically may be one of the causes of the nugget effect, is negligibly small for this case. We did not mean to claim that the nugget itself is negligibly small. We accept that the particular sentence was poorly constructed. We have now modified the text (Revised manuscript 7:10-11)*

[7:13-22] spatial stochastic simulation – Please provide more information about how the actual stochastic engine works. How was the variogram reproduced?

*Reply: We accept that this section requires more detail. We have modified this section in the revised manuscript (Revised manuscript Section 3.5).*

[7:22] I would except that a finer grid would improve your predictions. Especially when very high rainfall intensity is recorded over the domain, as a rapid (exponential) decrease in rainfall intensity from the centre away is expected (for convective rainfall at least).

*Reply: We agree that convective rainfall would vary rapidly and therefore having a higher resolution grid might improve the results. But we had only 8 measurement points over the area of 200m × 400m which gives a measurement resolution of 10000 $m^2$/measurement point. Hence prediction at every 25 m × 25 m (625 $m^2$) is fine enough for prediction of areal rainfall, also for convective rainfall. Increasing the resolution to 10 m × 10 m only reduces the standard deviation of the prediction by less than 5% in most cases while making the computational time six times higher (a summary on computation power is presented as supplementary material together with the revised manuscript).*

[Equation 6 and 7] Equation 6 – doesn't it also need to be divided by m? I think the readers are aware to the statistics of mean and standard deviation thus you can probably delete these equations.

*Reply: Thank you for pointing out this error in Equation 6. As per reviewer's suggestion we decided to remove these equations and modified the section accordingly. (Revised manuscript - Section 3.6)*

[8:17] "nugget effect . . . at zero lag distance due to measurement error" – are you sure it is just because of a measurement error? Rainfall variability exists between pair gauges, even for a 1 m distance, at least for temporal resolution of 1-5 min. Please check over the paper I have mentioned at comment [2:19-21] above. Your statement is repeated several times again during the text. I would at least discuss the possibility of having the nugget effect as more than a simple representation of rain-gauges measurement error.

*Reply: We accept that the nugget effect could be due to a combination of micro-scale spatial variability and measurement error. We modified this section to address this and to interpret the variograms better (Revised manuscript section 4.1).*

[9:18-21] I would argue that reason why "the behaviour of spatial correlation against rainfall intensity class is not very distinctive" in your study is due to the short period of data you have used.

*Reply: As stated in our response to comment [3:19-20], we had enough data points to develop meaningful and stable variograms based on which the above statement ("the behaviour of spatial*

*correlation against rainfall intensity class is not very distinctive")* is made. *Webster and Oliver (2007) suggested around 100 samples to reliably estimate a variogram model. Even in the case of 30 min temporal averaging interval and > 10 mm/hr (where we had the least observations) we had 196 sampling to calculate the variogram which is substantially larger than 100. Hence, we do not think that the absence of a clear trend in the behaviour of spatial correlation against rainfall intensity class is due to lack of data. Moreover in a previous similar study (Ciach and Krajewski, 2006), where the behaviour of spatial correlation against rainfall intensity was analysed, they also could not find a consistent trend and concluded that such trends are not consistent.*

[Equation 8] CV equation is also commonly known, I suggest you to delete this equation as well.

*Reply: Thank you for your suggestion. We agree that CV is a very common measure, but given the context and for completeness we decided to keep it in.*

[12:23] X-band radar can reach 250 m and 3 min resolution. I think it is good enough for small urban catchments.

*Reply: We agree with the reviewer. What we wanted to argue was that the resolution of most commonly available radar data (1000 m) is not enough for an urban catchment of this spatial extent (< 1000m). In addition the level of uncertainty in radar measurements would be much higher than that of point measurements, especially at a fine averaging interval (< 5 min) which is often of interest in urban hydrology (Seo and Krajewski, 2010; Villarini et al., 2008). We have modified this sentence in the revised manuscript. (Revised manuscript 15:20-23)*

[12:29-31] for a similar climate.

*Reply: Thank you for pointing out this. We have included this in the revised manuscript. (Revised manuscript 15:25-27)*

[General comment 1] I think your method suggested for rainfall upscaling is really interesting and can be very useful to some of the reader. However I, as a reader, would like to have more information, such as: What is the minimum number of rain-gauges required for a given catchment is order to apply your suggested upscaling (e.g. are 3 gauges over 1 km2 are enough?)? What should be the spatial configuration of these rain-gauges over the domain? Another question- if you would leave one of the gauges out of your analysis, how it would affect the results (what is the sensitivity of the network design?)?

*Reply: Thank you for the suggestion. We agree that this additional information would be useful to some readers, but to answer some of these questions we would need a more extensive analyses on the sampling design which we think is a research topic in its own right. Please find below our detailed response:*

What is the minimum number of rain-gauges required for a given catchment is order to apply your suggested upscaling (e.g. are 3 gauges over 1 km2 are enough?)? What should be the spatial configuration of these rain-gauges over the domain?

*We think that a simple and generic rule on number of data points cannot be derived for this methodology. Because, like any other geostatistical interpolation method, the efficiency of this method also heavily depends on reliable estimation of the geostatistical model (variogram). Hence it basically comes down to the question of whether a given rain gauge network can produce a meaningful variogram? As we mentioned in the manuscript, Webster and Oliver (2007) suggested around 100 measurement points to calculate a geostatistical model. But there is no obvious rule to define minimum number of bins and the number of samples for each bin to produce a reliable variogram.*

*Further, since pooling of repeated measurements would produce a multiplication of spatial lags, the length of the available data would also play a role in deciding the number of measurement locations.*

*We have included the above discussion in the revise manuscript (Revised manuscript 15:12-19)*

Another question- if you would leave one of the gauges out of your analysis, how it would affect the results (what is the sensitivity of the network design?)?

*Leaving one station out would affect the results. First it will reduce the accuracy of the estimation of the variograms as the number of spatial lags per time instant would reduce to 21 from 28. But the further effect of leaving one station out needs to be analysed in detail to see how it affects the uncertainty in the estimation of areal average rainfall intensity. In the manuscript we have not included such sensitivity analysis considering the direct relevance to the main scope of this study, length of the manuscript and the work load required to perform such analysis.*

[General comment 2] You stated that the stochastic model require some "computational demands". Can you give some details? How much time is needed to run the stochastic model per time step? What kind of machine do you need to use? It can be given as supplementary information but some readers might be interested to know.

*Reply:* *Thank you for this comment. We have provided this information as supplementary material together with revised manuscript.*

[General comment 3] The paper is oriented for the urban hydrology community, but if fact who will benefits the most from your method are hydro-meteorologists that are often looking for different methods to upscale rainfall observation from point scale to weather radar scale. Consider changing the title and addressing this as well. I think that due to the lack of sufficient length of observation, you should focus more on the method and who can benefit from it and less on your results.

*Reply:* *Thank you for the suggestion. We think the spatial extent ($0 - 400\ m$) and the temporal averaging intervals (2 min - 30 min) considered in this study are in the interest of the urban hydrology community. Also we think the uncertainty estimation in areal rainfall would be more useful for the hydrology community working on uncertainty. These are the main reasons why the paper was oriented towards the urban hydrology community.*

*Regarding the comment on lack of data and focusing more on the methodology, we request the reviewer to refer to our responses to the general comment and specific comment [3:19-20].*

**H. Müller (**mueller@iww.uni-hannover.de**)**

Complete author list: Müller, Hannes; Callau Poduje, Ana; Fangmann, Anne; Plötner,
Stefan; Shehu, Bora; Uniyal, Bhumika
(all from the Institute of Water Resources Management, Hydrology and Agricultural
Hydraulic Engineering, Leibniz Universität Hannover, Hanover, 30167, Germany)

This review results from six reviewers, all interested in the topic of the manuscript. Due to the number
of six additional reviewers, the review is organized in major comments, suggestions and technical
notes.

Brief summary: The manuscript deals with uncertainties resulting from upscaling of rainfall data.
Upscaling hereby includes temporal aggregation as well as the determination of areal rainfall from
point measurements. The topic is highly interesting and the investigation can be a good contribution
to this field. However, we think that the manuscript can be improved significantly in the methods and
the results part.

*Authors: We thank the reviewers for the professional and thorough review of this paper. We have
attempted to address all the comments listed below.*

**Major Comments:**
Data:

p4 l7-21 The measuring period is quite short with two summer periods in 2012 and
2013. However, for such a dense network this is often the case. The two periods differ clearly and
hence it is difficult to draw general conclusions from results.

*Reply: We acknowledge that the data cover only 10 months, i.e. two summer periods in 2012 and
2013, but however our geo statistical models (based on which further results are produced) are
stable. This has now been discussed in detail in our revised manuscript. (Revised manuscript [10: 30-
11:11])*

- From the two periods events are selected using a certain threshold. Why are events selected and why
is the investigation not carried out for the whole observation periods?
The results shown later are not based on/related to events. Is there a need for the event separation?

*Reply: The event separation (ref Table 1) is used only for the analyses presented in Section 4.2. We
apologise if this was not clear in the manuscript and we clarified this in the revised manuscript.
(Revised manuscript [4: 27-29])*

- A threshold of 10 mm network average rainfall depth and a minimum of 20 min rainfall duration
were chosen for the event selection. How have these thresholds been chosen? The chosen thresholds
can lead to exclusion of convective events with high rainfall amounts at one station, but no rainfall at
the other stations. This is also indicated by the durations of the resulting events, ranging from 1.5 h to
11.4 h, which are more typical for stratiform events and not convective ones. Indeed these convective
ones are crucial for urban hydrology and the resulting uncertainty in spatial upscaling is very high.
Have convective events be excluded from the investigation by the chosen thresholds?

*Reply: We chose a 20 min window in order to get sufficient data when temporal averaging intervals of
more than 2 min are used in the analysis. For example, a 10 min event could give two data values for
a 5 min averaging interval and would give only 1 data value for 15 min and also for 30 min averaging
intervals. Hence in order to have at least two data values for all temporal averaging intervals
examined a minimum event duration of 20 minutes was needed. Table 1 shows that the lowest event*

*duration in the collected data was 1.5 hours. Hence all events had at least 45 data values for 2 minute averaging interval and at least 3 data values for 30 minutes averaging interval.*

*As we mentioned in the above response the entire ten months of rainfall data from 8 locations were used for the development of the geostatistical model in the form of variograms. Hence no data are excluded from the investigation. The event separation (ref Table 1) is used only for the analyses presented in Sections 4.2.*

- How is the network average rainfall depth calculated for the event selection? In the introduction several methods are discussed. Is ordinary kriging applied here?

*Reply: The network average rainfall depth is calculated using arithmetic mean of the rainfall depths of 8 stations over the network. Ordinary kriging is not applied here.*

Methodology:
p5 l12 What is pooled – events or single time steps? In the text before, time steps (p5 l1) and events (p5 l6-7) are mentioned. If time steps are pooled (and not events), later for one event different variograms may be used due to different intensities of the single time steps in the event, right?

*Reply: Time instants (i.e. sample variograms for time instants with similar rainfall characteristics), not events, were pooled to increase the number of spatial pairs. Please refer to p4 | 26 – p5 | 3 in the manuscript for a detailed explanation (In revised manuscript 5:13-18). P5 l 6-7 is just a part of an explanation on how the intensity classes were chosen.*

*Different variograms corresponding to different intensity classes can be used for a single event as a single event can contain a range of intensity values which fall into different intensity classes.*

p7 l14 What is spatial stochastic simulation? All results are based on this method, so an explanation in the text is necessary (not only a reference). Is it applied as a subsequent step to the ordinary kriging or instead of the ordinary kriging? What is the stochastic simulation based on?
*Reply: We accept that this section requires more detail. We have modified this section in the revised manuscript to answer the reviewer's comment (Revised manuscript Section 3.5).*

Results:
p8 l25 The nugget-to-sill ratio is interpreted as measurement error, decreasing with an increasing temporal aggregation. The movement of events is ignored, which could significantly contribute to this ratio. With 2 min time steps, the event has reached one (pair) of the gauges, after 30 min all gauges are influenced by the event. This explanation should be implemented. Is it possible with other measurements (wind velocity,: : :) to exclude / quantify this effect? Also, can the whole nugget effect be described as measurement error from the author's point of view?

*Reply: We accept that the nugget effect could be due to a combination of micro-scale spatial variability and measurement error. We modified this section to address this and to interpret the variograms better (Revised manuscript section 4.2).*
*Since TB error is sampling related, other measurements (wind velocity, etc) cannot help quantifying or reducing this error.*

Conclusions:
General comment: Some conclusions are trivial (e.g. the intensity becomes less with increasing averaging interval), and there could be more conclusions out of the investigation.
What is the message to the urban hydrologic modelers? How can this uncertainty be involved in the calibration process/result discussion? Is the uncertainty greater/smaller than other uncertainties in urban hydrological modeling? Is it useful to take this uncertainty into account, if others are higher?

What are results of other investigations concerning areal rainfall uncertainties? Is it assumed, that the uncertainty increases with increasing area sizes in the lumped model? What is the recommendation for rain gauges number per square kilometer from this investigation? How sensitive are the results, if the station density/combination of stations is changed in the investigation? The measurement set-up is quite dense. Can general conclusions be drawn to less dense networks (and how)? Can the results be validated with an urban hydrologic model?

_Reply:_ *Please refer to our response on specific comment about trivial conclusion p12 l10-13.*

Thank you for suggesting more conclusions. Please find below our response.

….What is the message to the urban hydrologic modelers? How can this uncertainty be involved in the calibration process/result discussion? Can the results be validated with an urban hydrologic model?

*We believe the summary of our finding (p12|1-18) are all of interest to urban hydrologic modellers. In addition, results from this study can be used for uncertainty analyses of hydrologic and hydrodynamic modelling of similar sized urban catchments as it provides information on uncertainty associated with rainfall estimation. This estimate of uncertainty in combination of estimates of uncertainty due to model structure and model parameter will help to indicate the significance of rainfall uncertainty. The estimate of the relative importance of uncertainty sources can help to avoid false calibration and force fitting of model parameters (Vrugt et al., 2008). We included this discussion in the revised manuscript (Revised manuscript [15:25-31])*

*It is a challenging task to validate these results using hydrological modelling as such validation also needs estimation of other sources of uncertainty (structural and parameter) as well as overall uncertainty in the model output.*

….What are results of other investigations concerning areal rainfall uncertainties?

*We think this is something that should be included in the Discussion rather than the Conclusion. We already discussed some other related studies* (Ciach and Krajewski, 2006; Fiener and Auerswald, 2009; Krajewski et al., 2003; Pedersen et al., 2010; Villarini et al., 2008) *when comparing our results with those of other studies.*

….Is the uncertainty greater/smaller than other uncertainties in urban hydrological modeling? Is it useful to take this uncertainty into account, if others are higher? Is it assumed, that the uncertainty increases with increasing area sizes in the lumped model?

*Individual uncertainties will be catchment specific, but it is still useful to take this uncertainty into account because only by quantifying it will be known if it is larger or smaller than other uncertainty sources. We have not assumed that uncertainty in rainfall increases with increasing area size and since our scope does not cover this we cannot draw any conclusion on this issue.*

…What is the recommendation for rain gauges number per square kilometer from this investigation? The measurement set-up is quite dense. Can general conclusions be drawn to less dense networks (and how)?

*We think that a simple and generic rule on the number of data points cannot be derived from this methodology. It will be case-dependent but our methodology can help solve this problem in any particular case. Because, like any other geostatistical interpolation method, the efficiency of this method also heavily depends on reliable estimation of the geostatistical model (variogram). Hence it basically comes down to the question of whether a given rain gauge network can produce a meaningful variogram. As we mentioned in the manuscript, Webster and Oliver (2007) suggested*

*around 100 measurement points to calculate a geostatistical model. But there is no hard and fast rule to define minimum number of bins and the number of samples for each bin to produce a reliable variogram. Further, since pooling of repeated measurements would produce a multiplication of spatial lags, the size of the available data set would also play a role in deciding the number of measurement locations.*

*We have included the above discussion in the revise manuscript (Revised manuscript [15:12-19])*

….How sensitive are the results, if the station density/combination of stations is changed in the investigation?

*We did not investigate the sensitivity of results to changes in station density/combination so we cannot answer this question. Since our paper is already quite long we decided not to add this specific issue as well. It could be addressed in subsequent research.*

Suggestions:
Title: The title doesn't fit to the content of the manuscript. There are no urban hydrological models applied. Also, if no kriging is applied (not sure about that, see major comment p7 l14), it's not an geostatistical upscaling. The title "Estimation of uncertainties from spatial and temporal upscaling on an urban scale" is therefore misleading.

Reply: *Please refer to our response to comment p7 l14 for the explanation on spatial stochastic simulation, which is a geostatistical method. We believe it is quite clear from the methodology that this study uses geostatistical upscaling (e.g. the derivation of variograms, the use of spatial stochastic simulation, aggregation of simulations at points to spatial averages). Further, we think the spatial extent ($0 - 400\ m$) and the temporal averaging intervals (2 min -30 min) considered in this study are of interest to urban hydrology. Also, we think uncertainty estimation in areal rainfall would be more useful for the (urban) hydrology community working on uncertainty. These are the main reasons why the paper was oriented towards the urban hydrology community. Hence we don't think the title is misleading.*

Introduction:

p2 l19-25 There exist other methods for the estimation of uncertainties (bootstrapping,: : :), which should be mentioned in this context. Indeed, a focus should put on these methods, their comparisons and a reasonable decision for the applied method should be given at the end.

Reply: *Thank you for the suggestion. We would like to point out that we did not aim to discuss all uncertainty methods that are available for hydrological applications. Rather we wanted to keep the Introduction mainly focused on the methodology that we adapted for this study. Hence we followed the below order in our introduction*

- *Lumped hydrological models need spatial average rainfall over catchments*
- *Focus is on a case where the input data are rainfall observations at points*
- *Thus point observations need to be scaled up*
- *Review of existing methods that can do this*
- *Disadvantages of these methods*
- *Solution that does not have these disadvantages is to take a geostatistical approach*
- *The main challenges with geostatistical approach and how these can be dealt with*

*We quoted the most relevant studies wherever necessary. The paper did not aim to compare uncertainty methods but to examine the levels of uncertainty in rainfall intensities.*

p2 l8-19 In the introduction a number of interpolation methods are mentioned, which are not used afterwards in the investigation. They could be left out.

*Reply: Please refer to our response to specific comment p2 l19-25.*

Methodology:
p5 l11 How have the thresholds for the pooling been chosen?

*Reply: The maximum threshold value was limited to 10mm/hr to have enough time instants for the highest range (i.e. > 10 mm/hr) in order to produce stable variograms even at 30 min temporal averaging interval. It was then decided to divide the 0 − 10 mm/hr class to two equal subclasses (i.e. < 5mm/hr and 5-10 mm/hr). This resulted in three subclasses, which is a reasonable number given the size of the data set and computational demand.*

*We have included the above text in the revised manuscript. (Revised manuscript [5:25-29])*

p6 l2 Methods and results (Fig. 6) are mixed.

*Reply: We accept that in Fig 6 part of a result is presented, but we believe that this combined figure helps to explain step 3 clearly and consequently makes it easier for the reader to understand. Further this is not a major result, but just an outcome of one of the steps.*

p6 l10 The method of NST could be explained briefly.

*Reply: Thank you for the suggestion. Section 3.3 already briefly explains NST with the basic theory and literature where detailed description of NST including the steps involved can be found. Considering the length of the paper and the length of methodology section itself we decided to keep this section as it is.*

p6 l17 Step 4 is not a step, only a description, and can be moved to step 5.

*Reply: Step 4 is one of the major steps of this study. It involves the construction of variograms. The reviewer is kindly requested to refer to Section 3.4 in the manuscript for further explanation.*

p7 l4 explanation for q is missing

*Reply: Thank you for pointing this out. We have replaced q with n, which is the total number of observation points.*

p8 l9 With the standard deviation and the mean of areal rainfall intensities the restandardisation is carried out. For the former standardisation the standard deviation and the mean of point values have been used (since it is not clear, what has been pooled (see comment p5 l12), we assume time steps). Shouldn't be standard deviation and mean for standardization and re-standardisation be from the same type, so either from area or point values?

*Reply: Time instants (i.e. sample variograms for time instants with similar rainfall characteristics), not events, were pooled to increase the spatial pairs. Hence mean and standard deviation from each time instant is used for standardisation (ref Equation 1) as well as inverse standardisation. Clearly, the mean and standard deviation used for standardisation should also be used for re-standardisation.*

Results:

p8 l6-17 The nugget is interpreted as spatial rainfall variability or measurement error. The network offers the great possibility to have rain gauge pairs with distances of 1 m. Measurement errors have been excluded before by the paired measured time series.

So the spatial variability can be shown for these small distances, or why should this not be possible?

Reply: *Although paired gauges are used for efficient quality control, it cannot avoid sampling related error of tipping buckets (Habib et al., 2001). Regarding the interpretation of variograms, please refer to our response to specific comment p8|25.*

p8 l28 Since all errors have been excluded under the usage of the paired time series, the word TB error is somehow misleading. "Sampling error" could be more appropriate.

Reply: *Thank you for the suggestion. But this sampling error is only associated with tipping bucket type rain gauges. That is why we preferred to call it TB error. Further, the same term was used in a previous study (Habib et al., 2001) on sampling errors of tipping bucket type rain gauges, which was quoted in our discussion. Hence we wish to use the same term to be consistent with previous studies.*

p10 l3 Showing the CV would be more effective then showing the standard deviation in Fig. 8. An increasing of the standard deviation with an increasing intensity is trivial (which is even stated on p11 l7-10). Also, a logarithmic plot would be useful.

Reply: *Thank you for the suggestion. We modified the Fig. 8 (Revised manuscript Fig. 9) to show CV instead of standard deviation and modified the section 4.2.1 accordingly. We also made the x-axis logarithmic for easier interpretation.*

p10 l31-32 Design on peak rainfall intensity: The intensity AND the duration are important and both are used for the dimensioning of e.g. a sewer system.

Reply: *We agree with the reviewer that both peak and duration are important in the design of urban hydraulic structures. We have corrected the sentence in the revised manuscript (Revised manuscript 13:12-14)*

p11 l1 Fig. 10 Maybe it would be useful to use violin plots instead of only the standard deviation to show the uncertainty.

Reply: *Thank you for the suggestion. But based on one of the reviewer 1's comment we have replaced this figure with Fig. C1 below and modified the discussion accordingly. This plot includes labels of CV values instead of error bars.*

[Figure]

**Fig. C1: Predictions of event peaks of AARI (indicated by points) together with labels indicating corresponding CV (%) values**

p11 l12 Fig. 11 The readability of the figure could be increased by colors and/or drawing only contours, not filling them. Showing the means as functions, not as fixed values, would give better conclusions. The high mean for 2 min, <10 mm/h is caused by only one extreme CV (_13 %) and is not representable.

*Reply: We removed Fig. 11 from the manuscript and discussions are now based on revised Fig. 10 (please refer to our response to comment p11 l1).*

*We accept the fact that the sample size is too small to derive a firm conclusion. To comment on prediction uncertainty against intensity with a larger sample size, we modified the Fig. 8 (Revised manuscript Fig. 9)to show CV instead of standard deviation (refer to our response to comment p10 l3). Now it can be seen from modified Fig. 9 that there is a clear trend of increasing CV with increasing AARI. We also modified the discussions in the sections 4.2.1 and 4.2.3 accordingly.*

*Further, although we do not think peak prediction at event 8 is an outlier, we also performed the test without event 8, which results in the same trend. But the average CV for the range < 10 mm/hr is reduced to 5.3 % from 6.6% at 2 min averaging interval.*

Conclusion:
p12 l10-13 Decreasing peaks due to aggregation in time is trivial and not a conclusion.

*Reply: We agree with the reviewer that decreasing peaks due to aggregation in time is trivial. But our conclusion gives a quantification of this reduction to show its significance. It helps to subsequently discuss the trade-off between temporal resolution and accuracy in rainfall prediction, which is the main aim of that bullet point. In any case, we moved this conclusion to the discussion and emphasize more on the fact that reduction in the peaks is obvious. (Revised manuscript 13:3-6)*

p12 l32-33 "This information can help to avoid false calibration and force fitting of model parameters" It remains unclear, how the result of the investigation can be used for the avoidance of the before mentioned issues.

*Reply: Results from this study can be used for uncertainty analyses of hydrologic and hydrodynamic modelling of similar sized urban catchments, in similar climates, as it provides information on uncertainty associated with rainfall estimation which is arguably the most important input in these models. This estimate of uncertainty in combination with estimates of uncertainty due to model structure and model parameter will help to indicate the significance of rainfall uncertainty. This estimate of the relative importance of uncertainty sources can help to avoid false calibration and force fitting of model parameters (Vrugt et al., 2008).*

*We have included this explanation in the revised manuscript (Revised manuscript [15:25-31]).*

**Technical notes:**

p7 l5 "locationsxl" to "locations xl" and "locationsx0" to "locations x0"

*Reply: Thank you for pointing this out. It has been corrected in the revised manuscript. We also revised the symbols to be consistent throughout the manuscript.*

p8 l5 Eq. (6) "pi" and not "px", also the division by "m" is missing – Since this is a simple equation, it could left out, also Eq. (7)

*Reply: Thank you for pointing out this error in Equation 6. As per reviewer's suggestion we decided to remove these equations and modified the section accordingly. (Revised manuscript - Section 3.6)*

p8 l6 Eq. (7) The term under the root has to be squared.
*Reply: Thank you for pointing this out.*

p8 l25 "nugget-to-still ratio" to "nugget-to-sill ratio" (several times)
*Reply: Corrected in the revised manuscript.*

p9 l3 "in their study found" to "found in their study"
*Reply: Corrected in the revised manuscript.*

p9 l15 "(2003) in their" to "(2003) found in their"
*Reply: Corrected in the revised manuscript.*

p10 l11-13 In Fig. 9 event 10 is shown, not event 11 (regarding to Fig. 10).
*Reply: Thank you. Corrected in the revised manuscript.*

p12 l22 "methods to a certain extent" – fuzzy phrase
*Reply: //The pooling procedure used in this study makes use of the continuous measurement of rainfall and helps provide a solution to meet the data requirements for geostatistical interpolation methods to a certain extent.//*
*By the term 'certain extent' we meant that pooling can only partially solve the problem of scarcity in measurement points as it does not produce any new spatial lags, but only extends the information for*

*existing lags. We agree it was not very clear and have reformulated the sentence in the revised manuscript (Revised manuscript 15:10-12)*

**Reference**

[revised manuscript text omitted]

**Table 1A1: Summary of computational power required for spatial stochastic simulation.**

| Computer used | Area (m²) | Grid size (m²) | number of simulations | Computational time per time instant (s) |
|---|---|---|---|---|
| Core i5, 1.7 GHz, 4 | 200 × 400* | 25 × 25* | 500* | 10* |
| processers , 8 GB | 200 × 400 | 25 × 25 | 1000 | 20 |
| RAM | 200 × 400 | 10 × 10 | 500 | 60 |

*This configuration is used in this study.